# Surprising Effectiveness of pretraining Ternary Language Models at Scale

**Ayush Kaushal**[1][2][*], **Tejas Vaidhya** [1][4][†][*], **Arnab Kumar Mondal** [4], **Tejas Pandey**[1][3],
**Aaryan Bhagat** [5], **Irina Rish** [1][2][4]

[1]Nolano AI  [2]University of Montreal  [3]IIT Kharagpur  [4]Mila - Quebec AI Institute  [5]UC Riverside

## Abstract

Rapid advancements in GPU computational power has outpaced memory capacity and bandwidth growth, creating bottlenecks in Large Language Model (LLM) inference. Post-training quantization is the leading method for addressing memory-related bottlenecks in LLM inference, but it suffers from significant performance degradation below 4-bit precision. This paper addresses these challenges by investigating the pretraining of low-bitwidth models specifically *Ternary Language Models (TriLMs)* as an alternative to traditional floating-point models (FloatLMs) and their post-training quantized versions (QuantLMs). We present *Spectra LLM suite*, the first open suite of LLMs spanning multiple bit-widths, including FloatLMs, QuantLMs, and TriLMs, ranging from 99M to 3.9B parameters trained on 300B tokens. Our comprehensive evaluation demonstrates that TriLMs offer superior scaling behavior in terms of model size (in bits). Surprisingly, at scales exceeding one billion parameters, TriLMs consistently outperform their QuantLM and FloatLM counterparts for a given bit size across various benchmarks. Notably, the 3.9B parameter TriLM matches the performance of the FloatLM 3.9B across all benchmarks, despite having fewer bits than FloatLM 830M. Overall, this research provides valuable insights into the feasibility and scalability of low-bitwidth language models, paving the way for the development of more efficient LLMs.

## 1 Introduction

The computational power of GPUs, measured in FLOPs, is increasing exponentially, doubling approximately every 1.26 years. In contrast, memory capacity and bandwidth are growing at a slower pace, doubling every 2 and 2.9 years, respectively (Gholami et al., 2024). This disparity highlights that computing capabilities are outpacing advances in memory technology. In Large Language Models (LLMs) inference, the primary bottlenecks are caused by model size (bits), which affects memory usage (memory capacity) and data transfer to processors (memory bandwidth). These issues are becoming more critical than the growing number of model parameters which affect the computational limits (FLOPs). For instance, state-of-the-art LLMs such as 340B Nemotron 4 (Nvidia et al., 2024) have sizes (in bits) exceeding the memory capacity of data center GPUs, such as 8xH100s. Token generation speed, or latency, is now limited by memory bandwidth (Kim et al., 2024). Addressing these bottlenecks requires more expensive training, exceeding Chinchilla's compute-optimal regime (Hoffmann et al., 2022), with less than 10B parameter models being trained on up to 15 trillion tokens (Touvron et al., 2023b; AI@Meta, 2024; Team et al., 2024). Another widely used technique is post-training quantization during deployment (Zhu et al., 2023); however, we demonstrate in Section 5 that this approach is sub-optimal.

In post-training quantization, LLMs initially trained using 16-bit floating point precision (referred to as FloatLMs) undergo parameter quantization, and the resulting models are termed QuantLMs. These models use optimized kernels for deployment, offering speedups nearly proportional to the compression factor (Frantar & Alistarh, 2024). However, quantizing to very low bitwidths creates a significant mismatch between the representations of pretrained FloatLM and the deployable QuantLM, resulting in undesired behavior and quality degradation (Li et al., 2024; Huang et al., 2024). Some state-of-the-art methods (Frantar et al., 2022; Egiazarian et al., 2024) mitigate this issue by using

---

[*]Equal contribution, listed in alphabetical order.
[†]Correspondence: tejas@nolano.ai

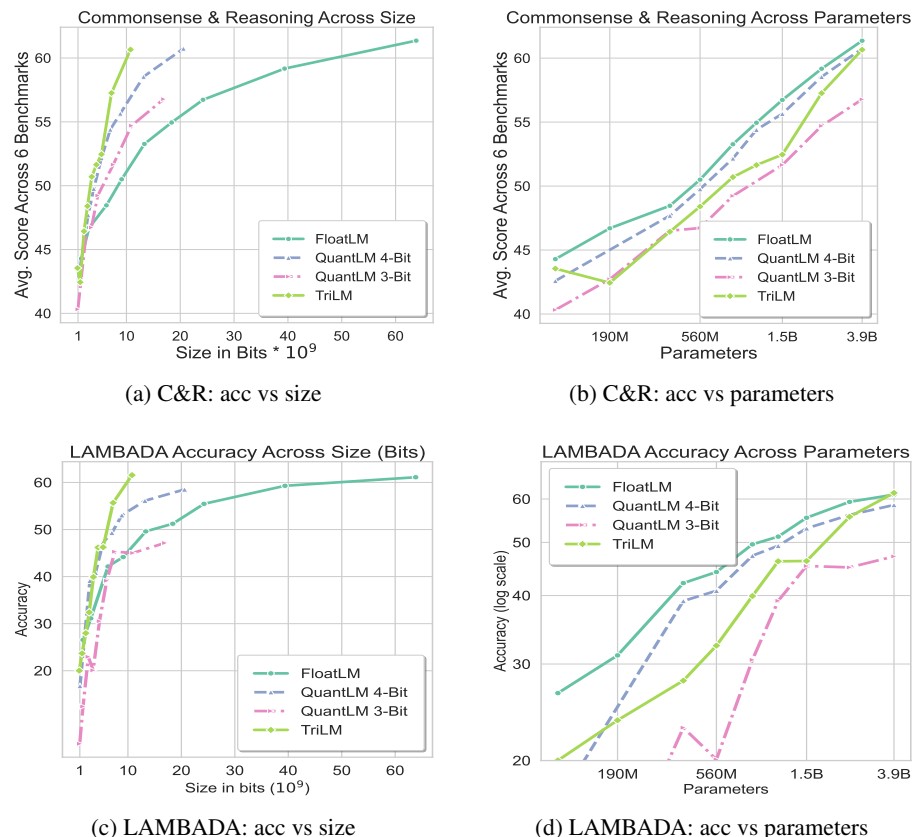

Figure 1: Common Sense and Reasoning (C&R) & LAMBADA Accuracy for ternary TriLM, FP16 FloatLM and quantized QuantLM models across different model sizes, in bits and number of parameters. C&R scores are averaged across 6 benchmarks. At 3B+ scales, TriLMs demonstrate better performance for their size than QuantLM and competitive performance to FloatLM of the same parameters. See Tables 7, 8 and 10 for details.

calibration and re-training data from target domains; however, this approach increases sensitivity to the calibration data. For instance, seemingly simple choices, like whether to length-normalize the calibration data, can significantly impact QuantLM's performance (Malinovskii et al., 2024). Other works have observed that QuantLM at 4 bits (4-bit QuantLMs) have about 65% lower knowledge capacity per parameter compared to trained and aligned FloatLMs (Allen-Zhu & Li, 2024).

An alternate approach to reducing model bitsize while maintaining parameter count involves training neural networks with low effective bitwidths (Zhou et al., 2018). This approach offers compression benefits beyond post-training quantization without its associated drawbacks. While low-bitwidth approaches typically employ binary (1-bit) or ternary quantization (1.58-bit), binary quantization generally underperforms compared to regular FP16 models (Liu et al., 2023a) (see Appendix B). In contrast, ternary modeling can match performance while significantly reducing memory requirements (as we demonstrate in section 5). Nevertheless, *the relative performance of pretrained low-bitwidth language models compared to QuantLMs across similar sizes (in bits) and similar parameter counts remains unclear*. This is a crucial unanswered question, given the high inference costs during the deployment of very large-scale LLMs. Moreover, *the training dynamics and scaling law of these low-bitwidth models are also poorly understood, partly due to the lack of publicly available systematic suites and associated comparative studies*.

Motivated by these challenges, we make the following contributions in this paper:

**Feasibility and Scalability of Training Ternary Language Models (TriLMs)** We discuss the deployment advantages (in section 2.1) and theoretical feasibility (in section 2.2) of training low-bitwidth models at scale. We then introduce ternary language models (TriLMs) and systematically study their scaling laws compared to FloatLMs. Our analysis reveals that TriLMs offer better scaling behavior in terms of model size, measured in bits (refer to Section 4.3). Moreover, as the number

of parameters increases, the difference in validation loss between TriLMs and FloatLMs becomes insignificant, indicating TriLM's competitive performance at scale.

**Spectra LLM suite.** We present **Spectra**, the first open suite of LLMs spanning many bit-widths. This suite includes FloatLMs, the corresponding QuantLMs at 3, 4, 6, and 8 bits, and ternary LLMs (TriLMs). The latter uses ternary weights {-1, 0, +1}. The suite features 9 models ranging from 99M to 3.9B parameters, all trained on the same 300B token dataset, totalling 54 models. We believe that the Spectra LLM suite makes a valuable contribution to the LLM research community by facilitating comparative studies, exploring the scalability and efficiency of ternary modeling, and improving interpretability from neuronal to connection levels.

**Evaluation and comparative analysis of TriLMs, FloatLMs, and QuantLMs at different scales.** We evaluate TriLMs, FloatLMs, and QuantLMs across multiple benchmarks, spanning commonsense, reasoning, knowledge capacity and toxicity. At the billion parameter scale, TriLMs consistently outperform their QuantLM and FloatLM counterparts of the same bit-size across all the aforementioned benchmarks (see Figure 1). Surprisingly, TriLM 3.9B matches the performance of FloatLM 3.9B across all benchmarks, despite getting a higher perplexity and being 5.9 times smaller in bitsize.

However, we also note that certain challenges remain in TriLMs. For instance, TriLM 3.9B exhibits the same level of toxicity and stereotyping as FloatLM 3.9B, significantly higher than a similarly sized FloatLM 830M when measured in bits. While TriLM 3.9B and FloatLM 3.9B show similar validation perplexity on some datasets, such as Penn Tree Bank and Lambada, a gap persists at this scale on web corpora, both in-domain (i.e., on a test subset of SlimPajama, the same domain used to train the models) and out-of-domain (e.g., on Dolma, C4 and RefinedWeb datasets). We provide detailed perplexity results in the appendix D.4.

## 2 MOTIVATION FOR LOW-BITWIDT MODELS

### 2.1 MEMORY BOTTLENECKS AND LANGUAGE MODEL DEPLOYMENT

**Memory Capacity of GPGPUs and Model Size (in Bits):** Figure 24a in Appendix G reveal that memory capacity has consistently lagged behind computational power across various accelerators, including recent hardware like Blackwell (Nvidia Team, 2024), MI325X (AMD Team, 2024), and Gaudi3 (Intel Gaudi Team, 2024). This gap is further exacerbated by advanced computational techniques like Ampere sparse or FP8. As shown in Figure 25a of Appendix G, the model sizes (in GB) for TriLM, QuantLM 4-Bit, and FloatLM scale differently with parameter counts. For simplicity, the figure excludes overhead from KV Cache, activations, and compilation during model deployment. FloatLM reaches the memory capacity of a single H100 at 34B parameters, with larger models exceeding the capacity of multiple GPUs. In contrast, QuantLM 4-Bit supports up to 300B parameters on a single MI300X. TriLMs, with over 300B parameters and appropriate packing, can fit on a single H100, making them not only efficient for GPU deployment but also ideal for edge devices.

**Memory bandwidth of GPGPUs and model inference speedup:** Figure 24b in Appendix G demonstrate the trends of Memory Bandwidth over FLOPs for accelerators over the years, highlighting the slower growth of memory bandwidth compared to computation. This trend, indicating a downward slope, aligns with Kim et al. (2024)'s establishment of the memory wall in autoregressive LLM computation. They found that the speed of token generation is bottlenecked by the rate at which data is fed from memory to processors, rather than the processing speed of the hardware. Consequently, the autoregressive decoding of LLM inference can theoretically achieve speedup proportional to its compression factor. Figure 25b in Appendix G illustrates the maximum possible speedup for QuantLM 4-Bit and TriLM compared to FP16 at different parameter counts. Even at 7 billion parameters, TriLMs can be more than 4 times faster at autoregressive decoding than FloatLM and 2 times faster than QuantLM 4-bit. While QuantLM 4-Bit plateaus at a maximum possible speedup factor of 4x, TriLMs plateau much higher at 10x for FloatLM.

## 2.2 LOW BITS CAN CAPTURE WEIGHT VARIANCE EFFECTIVELY AT SCALE

In this section, we use information theory to support our hypothesis: *as the number of parameters increases, training language models with low-bitwidth can effectively capture the necessary weight variance without significant information loss*. Assuming a fixed training dataset, we base this hypothesis on analyzing weight distributions in FloatLMs ranging from 99M to 3.9B parameters.

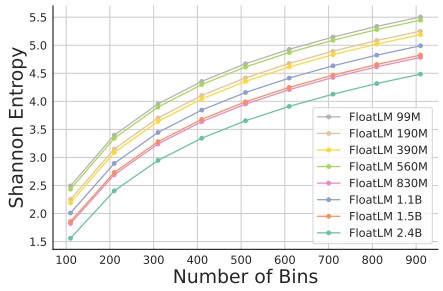
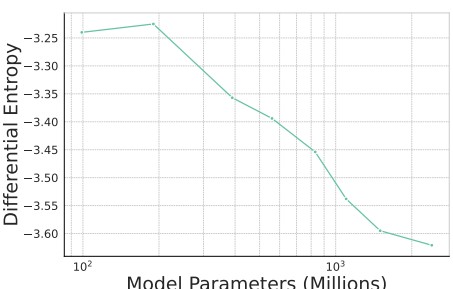

Figure 2: Shannon entropy (in bits) of discretized weight distribution with increasing number of bins.

Figure 3: Differential entropy of Gaussian fits on weight distributions across different scales.

Assuming that the weights of a trained model follow a Gaussian distribution (see Appendix E), we fit a Gaussian to these weights to understand their statistical behavior across model scales. The differential entropy is calculated using the expression: $H(W) = \frac{1}{2}\log_2(2\pi e \sigma_W^2)$ where $\sigma_W$ is the standard deviation of the weights (Papoulis & Pillai, 2002). As shown in Figure 3, differential entropy decreases with an increase in the number of parameters. This decrease indicates that the weights become more concentrated around the mean as the model size increases, suggesting higher predictability and less variability (MacKay, 2003). This reduced variability is due to overparameterization, which leads to redundancy in the weights Zhang et al. (2017); Neyshabur et al. (2018).

Additionally, we also use Shannon entropy, calculated by discretizing the weight distribution into ($N$) bins and computing $H_{Shannon} = -\sum_{i=1}^{N} p_i \log_2 p_i$, where $p_i$ is the probability of the weights falling into the $i$-th bin. The probability is given by the normalized frequency of weights within each bin. Discretization allows us to apply discrete entropy measures to continuous data, with the number of bins determining the representation granularity. Shannon entropy measures the average minimum number of bits needed to encode the weight values based on their distribution, providing a quantifiable measure of their "information content" (Shannon, 1948). Lower Shannon entropy indicates a more predictable, less diverse distribution that can be effectively encoded using fewer bits. Examining Shannon entropy across scales and bin sizes, we validate our finding that larger models require fewer bits for effective representation. As plotted in Figure 2, there is a general trend of decreasing Shannon entropy with increasing parameter count for a given number of bins.

These analyses support the potential of low-bitwidth models to match full-precision performance, especially as model sizes grow. In Section 4.3, we further substantiate this by analyzing the scaling behavior of our low-bitwidth TriLMs compared to FloatLMs.

## 2.3 SELECTING THE APPROPRIATE LOW-BITWIDTH MODEL

Selecting the appropriate quantization level is crucial for balancing computational efficiency and performance in low-bitwidth models. Binary models quantize weights to $\{-1, 1\}$, simplifying multiplications to efficient XOR operations (Courbariaux et al., 2016a), making them suitable for resource-constrained environments (Hubara et al., 2017). Ternary models introduce a zero state $\{-1, 0, +1\}$), replacing multiplications with additions and subtractions (Li et al., 2016), and improve efficiency by skipping calculations when weights are zero (Zhu et al., 2017). Conversely, models quantized to 2, 3, or 4 bits require more complex operations, including full multiplications, which are computationally more demanding than those required for binary and ternary models (Zhou et al., 2016). Therefore, this work only considers binary and ternary language models for further analysis.

The reduction in computational complexity in binary models often comes at the cost of significant performance degradation (Rastegari et al., 2016). In contrast, ternary quantization adds a zero state,

better approximating weight distributions without significantly increasing computational demands (Li et al., 2016), making them a sweet spot for balancing performance with efficiency gains (Zhu et al., 2017). Our observations align with these, as the scaling trends for given data size for our TriLMs consistently outperform those of BiLMs (Binary LLMs) across all parameter counts and bit sizes considered in this work, as shown in Appendix B and Figure 12. As a result, this work primarily focuses on the TriLM model, which we describe in the next section.

# 3  TRILM: TERNARY LANGUAGE MODEL

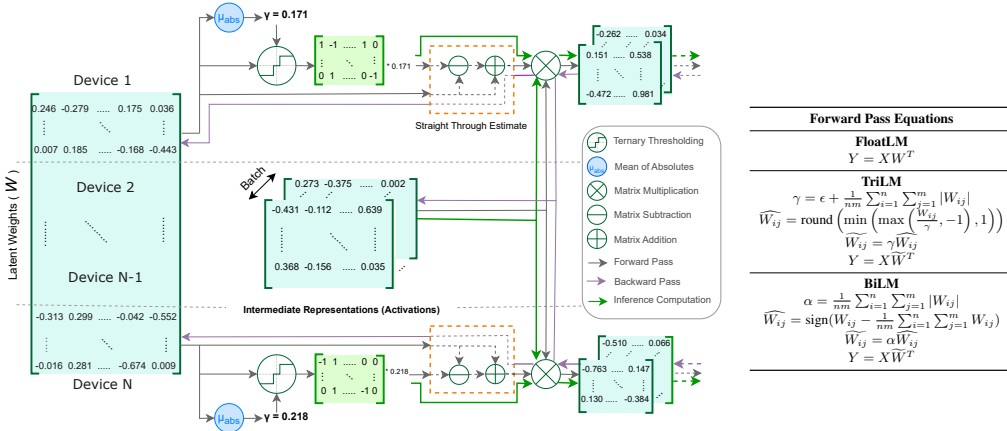

Figure 4: The computational flow of the forward, backward, and inference processes in TriLM's linear layer with N-Way model parallelism is shown on the left. Additionally, we provide the equations (on left) for the forward pass during the training of our FloatLM, TriLM, and BiLM (for details, see 1).

TriLM is a LLaMa-style (Touvron et al., 2023a) autoregressive transformers (Vaswani et al., 2017) model with RMSNorm (Zhang & Sennrich, 2019), SwiGLU Gated MLP (Shazeer, 2020), Rotary Position Embedding (RoPE) (Su et al., 2021), Multi-Headed Attention and no bias terms. In TriLMs, we represent the weights of all the linear layers in one of three possible ternary states $\{-1, 0, 1\}$, along with an additional floating-point number called 'scale value' shared across the matrix. During training, we maintain the *latent* (or master) weights in floating point precision, allowing for the accumulation of small updates over iterations that eventually contribute to a transition in the estimated ternary state of a parameter. As shown in Figure 4, during the forward pass, we ternarize the floating point latent weights on-the-fly. This process involves calculating the scale value to the absolute mean of the latent weights. After scaling, we estimate the ternary state of each parameter by rounding off to the nearest ternary state. In the backward pass, we use a straight-through estimator to compute gradients of the floating point latent weights (Bengio et al., 2013). Since we only need the scale values and the ternarized states during inference, we achieve a significant reduction in both model size and inference time at larger scales compared to FloatLMs. We provide a formal description of these forward pass, backward pass, and inference time equations in the Appendix (§A.1). We represent the embedding and language model head in half-precision floating point across all our experiments.

Since, training of TriLMs requires on-the-fly computation of scale values, synchronizing for a single scalar across devices in model parallel training (Shoeybi et al., 2019) can cause significant communication overhead. To address this, we allow each device to independently compute these scale values over its own matrix shard. This approach introduces additional artifacts, where the number of scalar values for each matrix is the same as the degree of model parallelism used during training (see appendix A.6). However, the impact on modelsize is negligible, for matrices with millions of parameters, we only add 6 scalar values each.

Concurrent research on low-bit LLMs, like BitNet 1.58 (Ma et al., 2024), also demonstrates the feasibility of training LLMs with ternary weights. Our experiments demonstrate that TriLM's architecture not only outperforms BitNet b1.58 but is also simpler and more stable. Moreover, both the larger BitNet 1.3B model presented in their paper and our replication of the BitNet 1.1B model

underperform compared to our TriLM 1.1B on commonsense and reasoning benchmarks. We provide further details on these comparisons in Appendix A.7.

# 4 SPECTRA SUITE: SPANNING PARAMETERS AND BITWIDTHS

The Spectra suite includes comprehensive families of large language models designed to span different parameter counts and bit-widths. This suite includes three main model families: *TriLMs*, *FloatLMs*, and *QuantLMs* (3, 4, 6, and 8 bits). Drawing inspiration from established model suites such as those by Biderman et al. (2023), Liu et al. (2023c), and Groeneveld et al. (2024), Spectra aims to facilitate scientific research on low-bitwidth LLMs, interpretability, and safety.

## 4.1 OVERVIEW OF SPECTRA SUITE

The spectra suite of LLMs is distinguished by several key attributes listed below.

1. **Scale.** The open suite spans a broad spectrum of scales across parameter count (99M to 3.9B), sizes ($9 * 10^8$ to $6.4 * 10^{10}$ bits) and bitwidths (1.58 bits to 16 bits).

2. **Uniform Training.** All the TriLMs and FloatLMs are trained on identical data sequences, specifically a 300B subset of Slim Pajama (Soboleva et al., 2023) dataset (see Appendix A.2), ensuring training consistency. Data ordering and batch sizes are also kept consistent within each model family. All QuantLMs undergo quantization using the same calibration data on identical data sequences, maintaining uniformity in model quantization procedures.

3. **Public Accessibility.** Training data as well as intermediate training checkpoints of TriLMs and FloatLms are publicly available for future research.

4. **Consistent Model Parameter Mapping.** All the model families maintain a consistent one-to-one mapping in terms of parameter count for studying scaling phenomena.

Figure 11 demonstrates the Spectra LM suite spanning across two dimensions - size (bits) and parameters. For each parameter count, we have 6 models across different bitwidths. Due to the availability of FloatLM, Spectra can easily be extended with new QuantLMs by using different Post Training Quantization methods. We provide details on the dataset and tokenizer, pretraining setup, hyperparameters, and optimization schedule across the families of models in Appendix A. Moreover, we describe our FloatLMs and their quantized versions in the next section.

## 4.2 FLOATLMS AND QUANTLMS

**Family of FloatLMs.** We utilize LLaMa-style (Touvron et al., 2023a) architecture akin to TriLM. In FloatLMs, we represent the parameters in the weight matrices of linear layers as floating-point numbers (FP16/BF16). The optimization schedule for FloatLM follows a cosine decay scheduling with weight decay and includes a learning rate warmup. This is consistent with the practices established in models such as Pythia, OLMo, LLM360. For more details, refer to the Appendix (A.4).

**Family of QuantLMs.** Recently, data-aware quantization techniques like GPTQ (Frantar et al., 2022) have emerged as efficient solutions for near-lossless weight quantization down to 4-bit precision (Dettmers & Zettlemoyer, 2023). In this work, we implement GPTQ post-training quantization to FloatLM, creating the QuantLM family of models across 3, 4, 6, and 8 bits. We quantize all transformer layer weights. For 3-bit and 4-bit quantization, we employ a group size of 128, which results in effective bit rates of 3.25 and 4.25 bits per parameter, respectively. We refine our approach by incorporating best practices from recent research by Malinovskii et al. (2024), particularly in terms of calibration data and scaling it to a million tokens for improved reconstruction. To ensure a fair comparison with TriLM, we maintain certain components in their original precision. Specifically, we do not quantize the embedding, language model head, or activations. Additionally, we employ symmetric quantization (without zero offset) for its simplicity, fast inference kernels support (Frantar & Alistarh, 2024), and comparable performance to asymmetric quantization, which uses separate zero offsets and scale values for each group. This approach also ensures consistency and facilitates a fair comparison with TriLMs. Notably, our Spectra suite is designed with flexibility in mind, allowing for easy extension to other quantization methods as needed.

## 4.3 TRAINING DYNAMICS

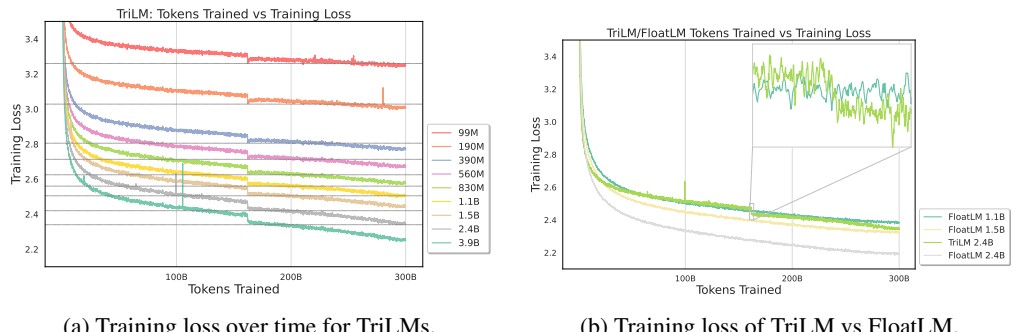

(a) Training loss over time for TriLMs.

(b) Training loss of TriLM vs FloatLM.

Figure 5: Training Cross Entropy Loss across steps for the TriLM family of models. At the halfway point (150B tokens) when we lower the peak learning rate, we observe a sudden drop in training loss. In the two-third way, removing weight decay leads to faster convergence.

Optimizing low bitwidth neural networks, such as in Quantization Aware Training (Liu et al., 2023b; Yuan et al., 2024; Bethge et al., 2018; Le & Li, 2023), requires specific considerations like higher initial learning rates and reduced weight decay. Our optimization schedule for TriLM incorporates two key interventions within a standard linear decay learning rate schedule with warmup and weight decay (L2 regularization). First, we reduce the peak learning rate at approximately the halfway point of training. Second, we remove the weight decay regularization about two-thirds into the training, as ternarization provides sufficient regularization (Courbariaux et al., 2016b). Figure 9 in Appendix A.5 shows an ablation study for a 1.1B parameter model trained on 100B tokens, demonstrating that both interventions are necessary to achieve the lowest validation perplexity.

Figure 5a illustrates the training loss curves for all the TriLMs trained and Figure 5b shows the relative training loss of a TriLM in comparison to two smaller FloatLMs. The loss curves demonstrate a continuous and consistent improvement in TriLMs with an increase in parameter count. During training, we make several key observations. First, minor spikes and drops in training loss occurred consistently across different TriLM scales at the same token counts, as all models were trained on identical data with the same ordering. Notably, the two largest models—TriLM 2.4B and TriLM 3.9B—each experienced a large loss spike in the first half of training. Second, adjusting the learning rate at the midpoint led to a sharp decline in training loss over a few hundred million tokens, though its impact varied by model size: for the larger models (2.4B and 3.9B), the rate of loss reduction returned to the prior pace after the initial sharp drop, while for smaller models (1.1B and 1.5B), the loss reduction plateaued, and models below 1B parameters saw an increase in training loss. Lastly, the removal of weight decay at the two-thirds mark accelerated convergence for all models, with the effect being most pronounced in the largest TriLM models.

## 4.4 SCALING LAWS

In this section, we derive the scaling law for TriLMs and compare it to that of FloatLMs. All models, including our largest 3.9B parameter model, are trained on 300B tokens, placing them in the overtrained regime as determined by Chinchilla's Hoffmann et al. (2022) compute-optimal token calculation. Figures 6a and 6b illustrate the final validation loss across different model size in terms of bits and number of parameters ($N$) respectively. In terms of effective model size in bits (6a), which is crucial during inference, TriLMs exhibit significantly better scaling than FloatLMs. Notably, TriLM 3.9B validation error matches with FloatLMs 1.1B, which is nearly 1.7 times larger in terms of effective bit size. To investigate the scaling behaviour in terms of $N$ with fixed data, we fit the validation loss to a power-law with offset[1] Hoffmann et al. (2022) (see Figure 6b and Appendix C):

$$\mathcal{L}_{\text{type}}(N) = \frac{A_{\text{type}}}{N^{\alpha_{\text{type}}}} + \epsilon_{\text{type}}, \text{ where } \begin{cases} A_{\text{TriLM}} = 185, & \alpha_{\text{TriLM}} = 0.26, & \epsilon_{\text{TriLM}} = 1.76 \\ A_{\text{FloatLM}} = 159, & \alpha_{\text{FloatLM}} = 0.26, & \epsilon_{\text{FloatLM}} = 1.67 \end{cases} \quad (1)$$

---

[1]derived using a fixed data regime of 300B tokens

We employ the Levenberg-Marquardt algorithm (Levenberg (1944); Marquardt (1963)) for efficient non-linear least squares fitting. Both FloatLM and TriLM share the scaling exponent $\alpha = 0.26$, indicating similar scaling behavior with the number of parameters $N$. However, the offset terms $\epsilon_{\text{TriLM}} = 1.76$ and $\epsilon_{\text{FloatLM}} = 1.67$ reveal that their validation losses converge as $N$ increases. Although TriLM starts with a higher coefficient $A = 185$, suggesting greater initial validation loss than FloatLM ($A = 159$), this difference becomes insignificant with larger $N$, aligning their performance at asymptotic scales as shown in Figure 6b and 17.

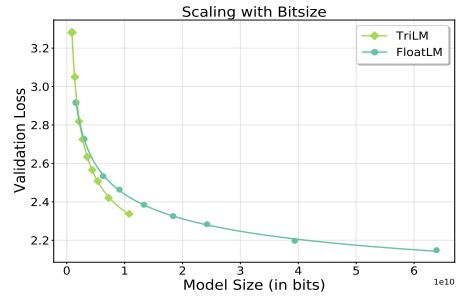
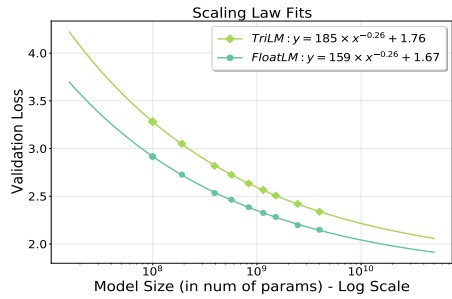

(a) Scaling laws - perplexity across size (bits).     (b) Scaling laws - perplexity across parameters.

Figure 6: Final validation loss across sizes (in bits) and parameters. TriLMs with increasing size offer better performance than FloatLMs of the same number of bits; and the gap in validation perplexity closes at scale.

Despite the observed differences in validation loss at the scale of models considered in this work, we demonstrate in Section 5 that at 3.9B parameters, TriLM offers competitive downstream performance compared to a FloatLM of the same parameter count across a variety of benchmarks in commonsense reasoning and knowledge-based tasks. Moreover, as discussed in Appendix §D.4, both models show similar perplexity on clean datasets such as Penn Tree Bank and OpenAI's Lambda. However, the gap in perplexity is observed in overlapping web-based datasets like Dolma and RefinedWeb.

# 5 RESULTS

We evaluate the families of LLMs on three aspects - commonsense & reasoning tasks, knowledge-based tasks, and toxicity, all of which are crucial measures of their downstream performance. Readers may refer to the appendix for more details regarding the benchmarks (§D).

**Commonsense and Reasoning.** We evaluate Spectra Suite models using eight distinct commonsense and reasoning benchmarks consisting of tasks from logical and reasoning questions to grounded and physical commonsense tasks: Arc Easy, Arc Challenge (Clark et al., 2018), BoolQ (Clark et al., 2019), HellaSWAG (Zellers et al., 2019), WinoGrande (Sakaguchi et al., 2021), PIQA (Bisk et al., 2019), LAMBADA (Paperno et al., 2016), LogiQA (Liu et al., 2021), all under zero-shot settings. Figures 1a and 1b display the average performance of the LLMs on the first six benchmarks across size in bits and number of params. Figures 1c and 1d present the performance for the LAMBADA dataset. The results show that TriLMs consistently demonstrate superior performance for their size across all benchmarks at the 2.4B and 3.9B parameter scales. At the largest scale of 3.9B, TriLM surpasses FloatLM on LAMBADA and achieves competitive average scores across six benchmarks. Additionally, TriLMs at the largest scales consistently outperform 4-bit QuantLMs of equivalent parameter count. However, across the considered scales, all LLMs show poor performance on LogiQA, making it difficult to identify a clear performance trend. For detailed benchmarking across all datasets –see Tables 7, 8 and 10.

**Knowledge.** Several downstream practical uses of LLMs require them to have knowledge about common subjects like science or politics. To evaluate the performance of LLMs on these subjects, we use SciQ (Welbl et al., 2017), TriviaQA (Joshi et al., 2017) and MMLU (Hendrycks et al., 2021) benchmarks in zero-shot settings. Figures 7a and 7b show the accuracy of the Spectra suite LLMs on SciQ across size in bits and parameter counts. Figures 21c and 21d depict the accuracy for TriviaQA, while 8a and 8b do the same for MMLU. Across both benchmarks, at large 2.4B+ scales, TriLMs offer

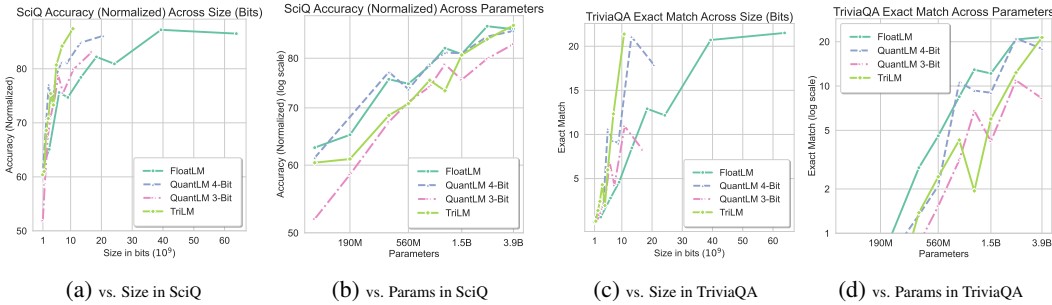

Figure 7: Performance of ternary TriLM, FloatLM and quantized QuantLM (3-bit & 4-bit) models on SciQ and TriviaQA tasks across Size (Bits) and Parameters. Refer to Tables 10 and 15 for details.

the best performance at a given size (bits). Surprisingly, despite having fewer bits, the knowledge capacity of TriLM does not have any significant degradation as observed in the case of QuantLMs (Allen-Zhu & Li, 2024). This indicates that low-bitwidth LLMs like TriLMs have similar knowledge capacity to FloatLMs, indicating that knowledge capacity is parameterized via the presence and nature of a connection (+1 or -1), rather than its strength. Tables 10 and 15 expand on these results.

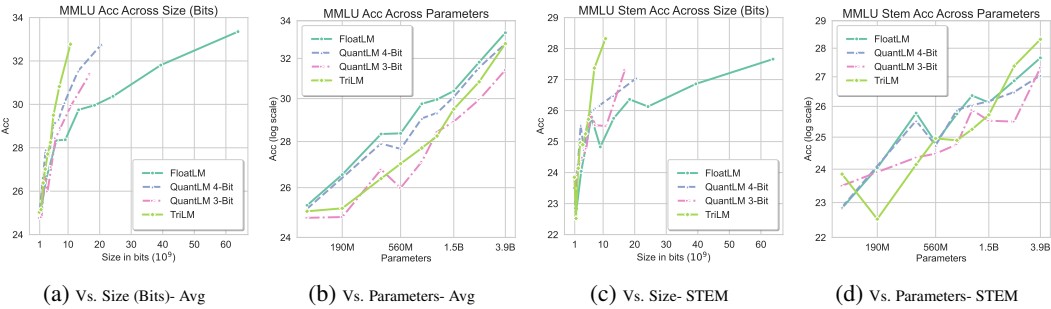

Figure 8: MMLU Accuracy for ternary TriLM, FloatLM and quantized QuantLM (3-bit & 4-bit) models across Size and Parameters. Please refer to Table 16 for details.

**Toxicity.** We evaluate the Spectra suite across various safety and toxicity benchmarks of TruthfulQA (Lin et al., 2021), Big Bench BBQ Lite (Parrish et al., 2022) and CrowsPairs (Nangia et al., 2020). These scores are listed in the Appendix in Table 15. We observe that none of the LLMs, even those with up to 3.9 billion parameters and trained on 300 billion tokens, perform significantly better than random guessing on the TruthfulQA benchmark. For the other two datasets, there is a noticeable correlation between the occurrence of toxicity and stereotypes and the LLMs' performance on various tasks. In particular, TriLMs with fewer than one billion parameters exhibit less stereotyping than FloatLMs with a similar parameter count. However, this difference diminishes as the scale increases, with TriLM 2.4B and TriLM 3.9B exhibiting biases comparable to those of FloatLM 2.4B and FloatLM 3.9B, respectively, on these benchmarks. This suggests that, although TriLMs initially show reduced bias compared to similarly sized FloatLMs, their performance aligns with FloatLMs of equivalent parameter counts at larger scales. This also highlights that TriLMs exhibit considerably more stereotyping than FloatLMs of comparable size (measured in bits), yet perform comparably to FloatLMs with similar parameter counts.

## 6    RELATED WORK

**Quantization of Large Language Models after Training.**    Post-training quantization (PTQ) algorithms convert a pretrained high-precision model (FP32 / FP16 / BF16) into a lower precision format without requiring the original training process (Cai et al., 2020; Hubara et al., 2020; Choukroun et al., 2019). These methods can be either data-independent or need a small calibration dataset. Additionally, PTQ for LLMs presents unique challenges due to numerical outliers in both weights and activations (Bondarenko et al., 2021). GPTQ (Frantar et al., 2022) is a state-of-the-art one-shot

weight quantization method aimed at finding a matrix of quantized weights (say $\hat{W}$) that minimizes the squared error relative to the full precision layer output. By leveraging second-order information, GPTQ derives a closed-form solution to this optimization problem, making it scalable to large LLMs. Other methods (Dettmers et al., 2023; Lin et al., 2024; Lee et al., 2024) emphasize the importance of outlier weights that correspond to high-magnitude activations. Some recent methods also quantized activation along with the weights (Xiao et al., 2024; Yao et al., 2022; 2023). Ahmadian et al. (2023) demonstrate that large activation outliers can be effectively mitigated at scale by making appropriate optimization decisions during the pretraining phase.

**Training Language Models At Lower Precision.** Several prominent language models such as GPT (Brown et al., 2020a), NeoX (Black et al., 2022), Llama and Pythia families have been traditionally trained using mixed precision (FP32/FP16 or FP32/BF16) (Micikevicius et al., 2018) or half-precision (FP16/BF16) (Kalamkar et al., 2019). Recently, Tao et al. (2022) introduced QuantGPT, a model that incorporates contrastive and logit distillation from a full-precision teacher to a quantized student model during pretraining. Further developments, such as BitNet (Wang et al., 2023) and BitNet b1.58 (Ma et al., 2024), have specifically focused on quantization-aware training for extremely low-bitwidth networks in transformer-based models. In their work, models are trained at low "effective" precision of binary and ternary respectively - where the latent (or master) weights during training are maintained in higher precision like FP16. The model weights are binarized or ternarized on the fly during the forward pass and gradients are backpropagated for the latent weights using the straight-through estimator (Courbariaux et al., 2016b). Prior works emphasize the importance of maintaining latent (or master) weights at high precision to allow accumulation of small updates during training - for example, Peng et al. (2023) observed a significant performance drop on the language model when the latent (or master) model weights were switch from 16-bits (FP16/BF16) to 8-bits (FP8) during training. Concurrent architectural improvements such as Flash Attention (Dao et al., 2022; Dao, 2023), the mixture of experts (Zoph et al., 2022), xLSTM Beck et al. (2024) and state space models (Gu & Dao, 2024; Dao & Gu, 2024; Gu et al., 2022) complement these advancements in lower precision modeling.

## 7 CONCLUSION AND FUTURE WORK

In this work, we address memory limitations in large language model (LLM) deployment by exploring both post-training quantization and direct low-bitwidth training. We introduce the Spectra LLM suite, featuring 54 models ranging from 99 million to 3.9 billion parameters, trained on 300 billion tokens. This suite includes Float16 LLMs (FloatLMs), quantized QuantLMs (3–8 bits), and our proposed ternary LLMs (TriLMs). Our findings reveal that TriLMs scale better than their half-precision Float16 counterparts in terms of effective model bit size, and they can achieve comparable validation loss when scaled to a large number of parameters. Additionally, our results demonstrate that TriLMs surpass other models in bit-size efficiency and achieve performance comparable to FloatLMs at 3 billion+ parameters across multiple benchmarks.

Future work should address the remaining challenges of toxicity, stereotyping, and performance gaps on web corpora associated with low-bitwidth models. Investigating scaling laws across diverse data regimes and bit sizes in low-bitwidth models trained with quantization-aware training can provide deeper insights into their behavior and limitations. Additionally, combining these models with state-space architectures like Mamba Gu & Dao (2024) could further enhance efficiency and performance without sacrificing accuracy. These research directions hold promise in advancing efficient language modeling further.

## ACKNOWLEDGEMENT

We acknowledge the support from the Mozilla Responsible AI Grant, the Canada CIFAR AI Chair Program and the Canada Excellence Research Chairs Program. This research was enabled by the computational resources provided by the Summit supercomputer, awarded through the Frontier DD allocation and INCITE 2023 program for the project "Scalable Foundation Models for Transferable Generalist AI" and SummitPlus allocation in 2024. These resources were supplied by the Oak Ridge Leadership Computing Facility at the Oak Ridge National Laboratory, with support from the Office of Science of the U.S. Department of Energy.

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

# Appendix

## Table of Contents

## A  ARCHITECTURE AND PRETRAINING DETAILS

This section provides a comprehensive overview of the architectural design and pretraining for TriLM (Ternary Language Model) and FloatLM (Floating Point Language Model). We outline the forward and backward pass equations specific to their linear layers, highlighting the contrast between the FP16 matrices in FloatLM and the ternary matrices with scalar scaling in TriLM. Additionally, it covers dataset selection, tokenizer usage, and preprocessing methods employed for training data preparation. These discussions provide information on pretraining setups, implementation nuances, and key hyperparameters critical to the models' development.

## A.1 FORWARD PASS, BACKWARD PASS AND INFERENCE EQUATIONS

Table 1 shows the equations across TriLM vs FloatLM for forward pass, backward pass and inference.

| Type | Forward Pass | Backward Pass | Inference |
|------|-------------|---------------|-----------|
| FloatLM | $Y = XW^T$ | $\frac{\partial L}{\partial X} = \frac{\partial L}{\partial Y} W$ 
 $\frac{\partial L}{\partial W} = \frac{\partial L}{\partial Y}^T X$ | $Y = XW^T$ |
| TriLM | $\gamma = \epsilon + \frac{1}{nm}\sum_{i=1}^{n}\sum_{j=1}^{m} \lvert W_{ij} \rvert$ 
 $\widehat{W_{ij}} = \text{round}\left(\min\left(\max\left(\frac{W_{ij}}{\gamma}, -1\right), 1\right)\right)$ 
 $\widetilde{W_{ij}} = \gamma \widehat{W_{ij}}$ 
 $Y = X\widetilde{W}^T$ | $\frac{\partial L}{\partial X} = \frac{\partial L}{\partial Y}\widetilde{W}$ 
 $\frac{\partial L}{\partial W} = \frac{\partial L}{\partial Y}^T X$ | Compute $\widehat{W}$ and $\gamma$ once and cache 
 $\widetilde{W_{ij}} = \gamma \widehat{W_{ij}}$ 
 $Y = X\widetilde{W}^T$ |
| BiLM | $\alpha = \frac{1}{nm}\sum_{i=1}^{n}\sum_{j=1}^{m} \lvert W_{ij} \rvert$ 
 $\widehat{W_{ij}} = \text{sign}(W_{ij} - \frac{1}{nm}\sum_{i=1}^{n}\sum_{j=1}^{m} W_{ij})$ 
 $\widetilde{W_{ij}} = \alpha \widehat{W_{ij}}$ 
 $Y = X\widetilde{W}^T$ | $\frac{\partial L}{\partial X} = \frac{\partial L}{\partial Y}\widetilde{W}$ 
 $\frac{\partial L}{\partial W} = \frac{\partial L}{\partial Y}^T X$ | Compute $\widehat{W}$ and $\alpha$ once and cache 
 $\widetilde{W_{ij}} = \alpha \widehat{W_{ij}}$ 
 $Y = X\widetilde{W}^T$ |

Table 1: Equations in the Linear Layer of TriLMs and FloatLMs.

**Reason for restricting the quantization approach to linear weights in TriLMs.** In developing extremely large language models like TriLMs, a key architectural strategy is to quantize only the linear layer weights while keeping the embedding layers and language model head in higher precision. This is driven by the need to reduce model size while maintaining performance. Linear layers (dense layers) hold the bulk of the parameters in transformer models (Vaswani et al., 2017). Quantizing these weights to ternary states significantly reduces the model size, facilitating deployment on memory-constrained hardware. However, the embedding layers and language model head remain in higher precision (e.g., half-precision floating point) to preserve critical functions in language understanding and generation. Embedding layers encode important semantic and syntactic information, and quantizing them would degrade performance (Mikolov et al., 2013). Similarly, the language model head, which maps internal representations to the vocabulary space, requires high precision to maintain prediction quality (Press & Wolf, 2017)

## A.2 DATA AND TOKENIZER

**Dataset Selection:** Let input be $X \in R_{b \times n}$ for a linear layer with FP16 weight matrix $W \in R_{m \times n}$ and $Y \in R_{b \times m}$ be the output. The same matrix $W$ is also used to denote latent weights in TriLMs during training.

For ternarized layers in TriLMs, we also have a scalar scale $\gamma \in R$, matrix with ternarized states $\widehat{W} \in \{-1, 0, 1\}_{n \times m}$ and ternarized matrix $\widetilde{W} \in R_{n \times m}$. We set $\epsilon = 1e - 5$.

Due to lack of availability of Pile 300B (Gao et al., 2020) used in Pythia, we opted to use a 300B token sample of deduplicated Slim Pajama dataset[2]. We sample from each subset with the probability proportional to its size.

**Training Data Preparation:**

- **Main experiments (Spectra suite)**: We used the full 300B token sample.

- **Ablation studies**: Training runs with 100B tokens, we sample from these 300B tokens with equal probability weight to each data-point.

- **Fine-Web Edu experiments**: We tokenized one-third of a 350B token sample, from which we then sampled 100B tokens for our experiments.

---

[2]We also make this subset public

**QuantLM:** For the creation of QuantLM, we utilized a subset of the Slimpajama-627B dataset, consisting of 512 samples with a sequence length of 2048. These samples were normalized for length. Our approach closely follows the methodology outlined in (Malinovskii et al., 2024).

**Tokenizer and Optimization Techniques:** We use the GPT-NeoX 20B tokenizer following Pythia. For speeding up training, we round embedding rounding of to the nearest multiple of 128 times the model parallel size.

| Dataset | Size (Tokens) |
|---|---|
| Arxiv | 13B |
| Book | 13B |
| C4 | 80B |
| Common Crawl | 156B |
| GitHub | 16B |
| Stack Exchange | 10B |
| Wikipedia | 12B |
| **Total** | 300B |

Table 2: 300B Subset of Slim Pajama

## A.3 PreTraining Setup

We scale using 2D-parallelism with Megatron-style sharding (Shoeybi et al., 2019) and use ZeRO stage 2 Deepspeed (Rasley et al., 2020) for ZeRO (Rajbhandari et al., 2020). Our implementation was based on GPT NeoX Codebase (Andonian et al., 2023). We use AdamW (Kingma & Ba, 2017) for optimization. We train on nodes with IBM Power9 PC CPUs and 6x16GB V100. Due to the lack of BFloat16 support in V100, we train both TriLM and FloatLM in FP16 using Mixed Precision Training and Dynamic Loss Scaling. Please refer to §A.6 for more implementation specific details. We extensively use Huggingface (Wolf et al., 2020) and Wandb (Biewald, 2020) for handling the checkpoints and experiment tracking.

## A.4 Hyperparameters

Table 3 shows the hyperparameters for TriLM and FloatLM's transformer architecture and their learning rate. We set Adam $\beta$ are set to (0.9, 0.95) for both families of models and all the reported runs are trained to 2048 sequence length. FloatLM and TriLM are respectively trained with batch sizes of $2M$ and $1M$ tokens respectively.

| Params | Hidden | GLU | Heads | Layers | MP | FloatLM LR | TriLM LR |
|---|---|---|---|---|---|---|---|
| 99.74M (99M) | 512 | 1280 | 8 | 16 | 1 | $4.0 * 10^{-4}$ | $2.4 * 10^{-3} \rightarrow 1.5 * 10^{-3}$ |
| 190.0M (190M) | 768 | 2048 | 12 | 16 | 1 | $4.0 * 10^{-4}$ | $2.4 * 10^{-3} \rightarrow 1.5 * 10^{-3}$ |
| 392.4M (390M) | 1024 | 2560 | 16 | 24 | 1 | $3.0 * 10^{-4}$ | $1.8 * 10^{-3} \rightarrow 1.2 * 10^{-3}$ |
| 569.2M (560M) | 1280 | 3072 | 20 | 24 | 1 | $2.8 * 10^{-4}$ | $1.6 * 10^{-3} \rightarrow 1.1 * 10^{-3}$ |
| 834.0M (830M) | 1536 | 4096 | 24 | 24 | 1 | $2.5 * 10^{-4}$ | $1.5 * 10^{-3} \rightarrow 1.0 * 10^{-3}$ |
| 1.149B (1.1B) | 1792 | 5120 | 28 | 24 | 2 | $2.2 * 10^{-4}$ | $1.3 * 10^{-3} \rightarrow 9.0 * 10^{-4}$ |
| 1.515B (1.5B) | 2048 | 6144 | 32 | 24 | 2 | $2.0 * 10^{-4}$ | $1.2 * 10^{-3} \rightarrow 8.0 * 10^{-4}$ |
| 2.461B (2.4B) | 2304 | 7680 | 36 | 30 | 3 | $2.0 * 10^{-4}$ | $1.2 * 10^{-3} \rightarrow 8.0 * 10^{-4}$ |
| 3.989B (3.9B) | 3072 | 9216 | 24 | 30 | 6 | $1.5 * 10^{-4}$ | $1.2 * 10^{-3} \rightarrow 8.0 * 10^{-4}$ |

Table 3: Hyperparameters across model sizes for TriLM and FloatLM.

| Params | 99M | 190M | 390M | 560M | 830M | 1.1B | 1.5B | 2.4B | 3.9B |
|---|---|---|---|---|---|---|---|---|---|
| FloatLM | 1.60 | 3.05 | 6.28 | 9.11 | 13.34 | 18.39 | 24.23 | 39.38 | 63.83 |
| QuantLM 8-Bit | 1.21 | 2.14 | 3.96 | 5.58 | 7.91 | 10.64 | 13.77 | 21.55 | 34.39 |
| QuantLM 6-Bit | 1.11 | 1.92 | 3.38 | 4.70 | 6.55 | 8.70 | 11.15 | 17.09 | 27.03 |
| QuantLM 4-Bit | 1.03 | 1.72 | 2.88 | 3.93 | 5.36 | 7.00 | 8.86 | 13.18 | 20.59 |
| QuantLM 3-Bit | 0.98 | 1.60 | 2.59 | 3.49 | 4.68 | 6.03 | 7.55 | 10.95 | 16.91 |
| TriLM | 0.90 | 1.42 | 2.11 | 2.76 | 3.55 | 4.42 | 5.36 | 7.23 | 10.76 |

Table 4: Sizes in bits ($*10^9$) for Spectra suite of LLMs across varying parameter counts.

## A.5 Optimization Schedule

In this section, we ablate the two interventions in a vanilla linear decay learning rate scheduling with warmup and weight decay (L2 Regularization). (1) *Peak LR* - at roughly the halfway point, we reduce

the peak learning rate. (2) *L2 Reg.* - at roughly two-thirds of the training, we remove the weight decay regularization as ternarization provides sufficient regularization (Courbariaux et al., 2016b). Figure 9 demonstrates the ablation run performed for a 1.1B parameter model on 100B tokens with both, only one and neither of these interventions.

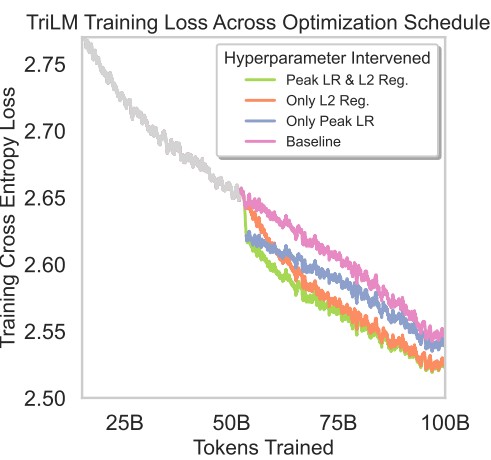

Figure 9: Training loss for a 1.1B parameter TriLM, across different optimization schedules. We intervene for combinations of two hyperparameters peak learning rate and L2 regularization. Intervention for both hyperparameters given best training loss.

Among these four runs, we notice the lowest final training loss when both, the L2 Regularization and Peak LR are intervened, closely followed only L2 Regularization (at 1/2 and 2/3) being intervened and then only Peak LR being intervened. Dropping the peak LR at the halfway point leads to a quick sharp drop in training loss. Similar phenomena have also been observed in schedules with small episodes of fast learning rate decaying like MiniCPM (Hu et al., 2024). On the other hand, removing L2 regularization, or weight decay, leads to accelerated convergence, which can even mostly have the same effect as lowering peak LR leading to a quick drop in loss. These relative training loss observation at 100B tokens also go hand in hand with relative downstream performance across commonsense and reasoning tasks, which are listed in Table 14 and 13. Thus, we fix the TriLM optimization schedule where we drop in the peak learning rate at the halfway mark and the weight decay is removed at the two-thirds mark.

### A.6 KNOWN IMPLEMENTATION ARTIFACTS

Similar to BitNet (Wang et al., 2023), our models exhibit artifacts resulting from model parallelism. A key issue arises when computing the scale, $\gamma$, across the entire weight matrix, which is sharded across multiple devices. This process introduces a significant communication overhead due to the all-reduce operations. In our implementation, we address this by computing the scales over the portion of the weight matrix local to each device. Consequently, during inference with TriLM models, scales are computed independently within each model parallel group. Importantly, this modification has a negligible impact on the bits per parameter, amounting to less than $10^{-5}$, even at the highest model parallelism level of 6 for our largest model.

Given that we train in FP16, some artifacts are expected as a result of this training method. However, we do not anticipate significant performance differences when comparing mixed precision training with BF16 or even FP32. This expectation is based on the observation that the lowest loss scales recorded during our runs were consistently at or above the recommended value of 128 (Micikevicius et al., 2018) (refer to Table 5).

### A.7 DIFFERENCES FROM BITNET ARCHITECTURE

TriLM differs from BitNet b1.58 in several ways for better performance as well as for fairer comparison with FloatLMs. Adopting the GPT-3's Pre-Normalization approach as outlined by (Brown et al., 2020b), normalization is applied prior to each linear layer. This method has proven essential for maintaining stable training under FP16 precision (Wang et al., 2023). Consequently, normalization occurs twice within each transformer layer: once at the input representations to the attention sub-layer and again at the input representations to the Gated MLP sub-layer. This approach contrasts with BitNet, where activation or intermediate representations are normalized, scaled, and quantized to 8 bits before each linear layer, which occurs between 4 to 7 times per transformer layer depending on

| Model | Min. Loss-Scale | # Skipped Batches | # Skipped Tokens |
|---|---|---|---|
| FloatLM 99M | 256.0 | 181 | 0.37B |
| TriLM 99M | 1024.0 | 303 | 0.33B |
| FloatLM 190M | 512.0 | 168 | 0.35B |
| TriLM 190M | 512.0 | 305 | 0.33B |
| FloatLM 390M | 1024.0 | 170 | 0.35B |
| TriLM 390M | 512.0 | 312 | 0.34B |
| FloatLM 560M | 256.0 | 164 | 0.33B |
| TriLM 560M | 512.0 | 294 | 0.32B |
| FloatLM 830M | 2048.0 | 175 | 0.36B |
| TriLM 830M | 128.0 | 307 | 0.33B |
| FloatLM 1.1B | 2048.0 | 158 | 0.32B |
| TriLM 1.1B | 512.0 | 306 | 0.33B |
| FloatLM 1.5B | 256.0 | 170 | 0.35B |
| TriLM 1.5B | 512.0 | 318 | 0.34B |
| FloatLM 2.4B | 1024.0 | 165 | 0.34B |
| TriLM 2.4B | 256.0 | 294 | 0.32B |
| FloatLM 3.9B | 256.0 | 164 | 0.34B |
| TriLM 3.9B | 128.0 | 309 | 0.33B |

Table 5: Final loss-scale and number of batches skipped across TriLM and FloatLM training runs - We are able to maintain above the recommended loss scales of 128 for mixed precision training (Micikevicius et al., 2018).

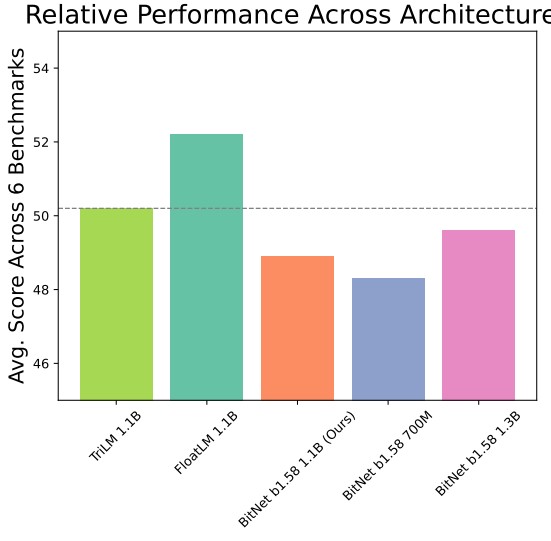

Figure 10: Performance across various architectures - TriLM 1.1B, FloatLM 1.1B, BitNet b1.58 1.1B (our replication) along with reported scores of BitNet b1.58 at 700M and 1.3B params. Scores are averaged across 6 common sense and reasoning benchmarks, mentioned in Table 14 and 13.

the specific implementation. Furthermore, TriLM employs RMSNorm with a scale parameter over the parameterless RMSNorm.

Figure 10 shows the commonsense and reasoning performance of TriLM 1.1B, FloatLM 1.1B and our replication of BitNet b1.58's architecture at 1.1B scale, along with the reported performance for BitNet b1.58 700M and 1.3B. All these models have been trained for 100B tokens. Our BitNet

replication achieves performance between the 700M and 1.3B models. However, all the BitNet models, including the larger 1.3B parameter model perform worse than TriLM 1.1B. It should be noted that at this 1.1B scale, TriLMs do not achieve parity with FloatLMs of the same parameter count. Table 14 and 13lists the detailed performance of these models across common sense benchmarks.

## A.8  Spectra Suite of LLMs: Parameter Scaling and Bitwidth Variations

Figure 11 illustrates the Spectra LM suite spanning two key dimensions: bitwidth and the number of parameters. The suite includes the TriLM, FloatLM, and QuantLM model families (in 3, 4, 6, and 8 bits). The suite features 9 different parameter scales ranging from 99M to 3.9B parameters. In total, the suite comprises 54 models.

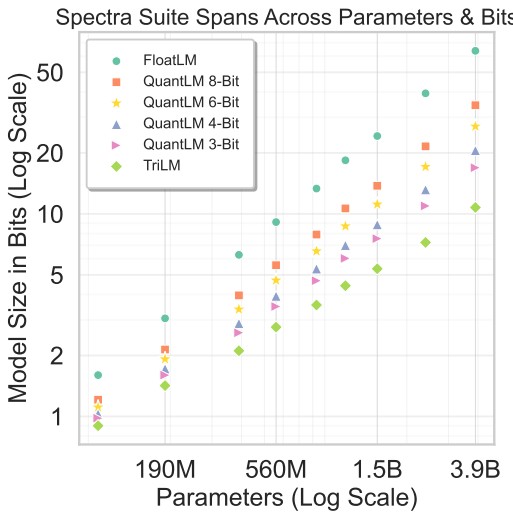

Figure 11: The Spectra Suite spans across two dimensions: parameters and bitwidth. Each point corresponds to a language model in the suite.

## A.9  Details on the Selection of Hyperparameters

**Model architecture hyperparameters.** To ensure stable training and mitigate risks of loss spikes or slow convergence, we carefully selected key architectural hyperparameters such as layer normalization techniques and other training parameters. To optimize hardware efficiency, we padded the vocabulary size and rounded the hidden dimensions to the nearest power of two, facilitating alignment with accelerator constraints. Given computational limitations, our scaling and coefficient analyses were conducted with nine data points within the 4-billion-parameter range, each trained on datasets containing 300 billion tokens. These constraints directly influenced the design choices for the number of layers, hidden dimensions, embedding size, feedforward network dimensions, and attention heads across model variants.

**Training hyperparameters.** We adopted a rigorous evaluation-driven approach to refine training hyperparameters, encompassing model initialization, optimizers, and learning rate schedules. This methodology mirrors the training-time evaluations utilized in other model suites such as those by Biderman et al. (2023) and Groeneveld et al. (2024). To systematically explore the hyperparameter space, we employed a grid search over configurations, including learning rate, weight decay, batch size, and optimizer variants. Evaluations are conducted at different stages of training, from 4 billion up to 20 billion tokens, providing continuous and early-stage feedback on model performance. The key metrics evaluated include validation loss, commonsense reasoning, knowledge-based task performance, and toxicity assessments, as outlined in the paper.

## A.10 PERPLEXITY FOR INCREASING TRAINING TOKENS IN TRILM MODELS

**Impact of Training Tokens on Perplexity in TriLM Models.** The performance of language models generally improves with an increased number of training tokens, as demonstrated in prior work (Hoffmann et al., 2022; Kaplan et al., 2020). This trend is clearly reflected in the validation perplexity of our TriLM models across varying training token counts. The results, summarized in Table 6, highlight the critical role of scaling up training data to reduce perplexity. Notably, the larger model—TriLM 560M—consistently achieves lower perplexity at all token scales. This reinforces the scaling law relationship among larger model sizes, more extensive training data, and reduced validation perplexity, mirroring the scaling laws observed for FloatLMs.

| Training Tokens | Perplexity of TriLM 99M | Perplexity of TriLM 560M |
|---|---|---|
| 50B | 27.11 | 15.48 |
| 150B | 26.30 | 14.50 |
| 300B | 25.70 | 14.01 |

Table 6: Perplexity comparison for TriLM models with 99M and 560M parameters across different training token counts.

**Impact of Token-to-Parameter.** The reduction in validation perplexity, even after training on over 3000 times the number of tokens relative to the number of parameters, is particularly evident in the smallest model (99M) of our suite. As shown in Table 6, the 99M model continues to demonstrate improvement with up to 300B tokens, suggesting that TriLM remains effective even at higher training token-to-parameter ratios. Extrapolating from this trend, our largest model (3.9B) could theoretically train on up to 11.8 trillion tokens without reaching convergence. This scale is comparable to the magnitude of training tokens used for LLaMA 3, a state-of-the-art open-source language model AI@Meta (2024). However, due to resource constraints, training at such a scale remains a direction for future work.

# B BINARY VS TERNARY LARGE LANGUAGE MODELS

In this section, we will comprehensively compare Binary Large Language Models (BiLMs) with Ternary Large Language Models (TriLMs). We will start by describing BiLMs, followed by studying scaling laws and presenting results on various benchmarks, as well as comparisons with TriLMs across parameter count and model size (in bits).

## B.1 BILM: BINARY LARGE LANGUAGE MODEL

In Binary Large Language Models (BiLMs), the weights of the linear layers are represented by binary values of -1 or 1, with an accompanying floating-point scaling factor, similar to the method employed in TriLMs. Comprehensive formal descriptions of the forward pass, backward pass, and inference time calculations are provided in Appendix (§A). We have trained three BiLM models of distinct sizes: 99M, 560M, and 1.1B parameters. These models were trained on the same dataset and in the same sequence as the TriLMs, adhering to the identical optimization schedule detailed in Appendix A.5.

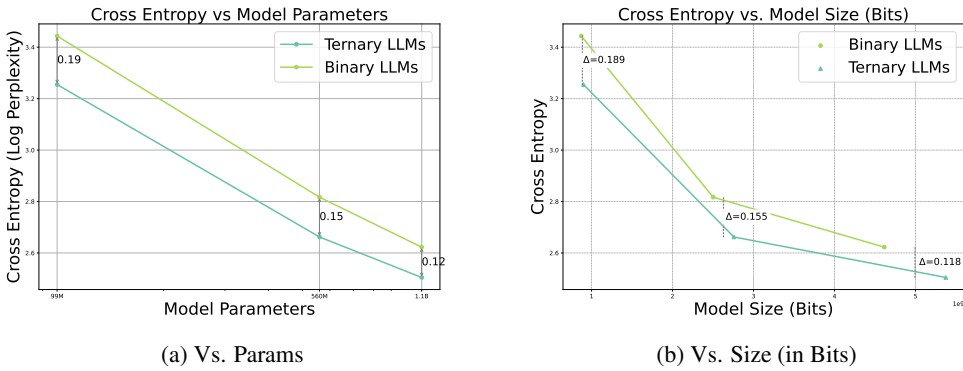

(a) Vs. Params                    (b) Vs. Size (in Bits)

Figure 12: Final Validation loss across size (measure in bits) and parameters.

## B.2 SCALING BINARY VS TERNARY MODEL

Figures 12a and 12b show the final validation loss across model sizes (in bits) and parameter counts, respectively for the three distinct sizes previously mentioned. At the Billion+ model scale, Ternary Models appear to be preferable in terms of both the number of parameters and model size (in bits). However, the gap seems to be narrowing, which suggests that BiLMs have the potential to be competitive with TriLMs at higher parameter counts (100B+). Therefore, we decide to only scale TriLMs further to study the scaling laws of FloatLMs vs TriLMs up to 3.9B parameters.

## B.3 RESULTS

We conducted a comprehensive benchmark analysis of Binary Large Language Models (BILMs) across three key dimensions: commonsense and reasoning tasks, knowledge-based tasks, and toxicity evaluation, as detailed in Tables 7, 810, and 15

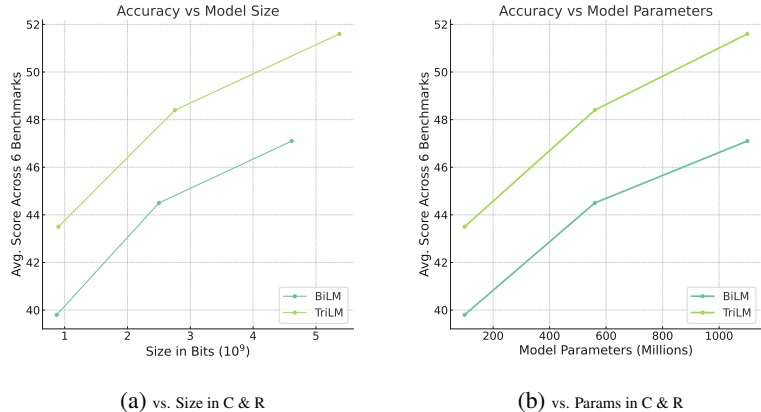

(a) vs. Size in C & R

(b) vs. Params in C & R

Figure 13: Performance of ternary TriLMs and BiLMs models on commonsense and Reasoning and MMLUs tasks across Size (Bits) and Parameters. Refer to Tables 7 and 8 for details.

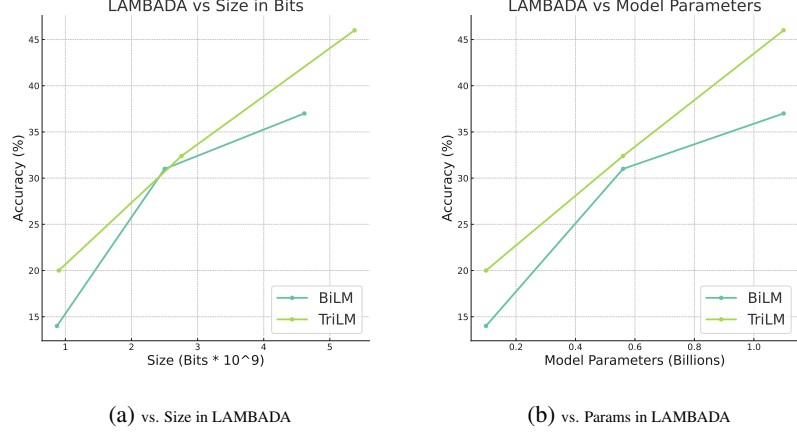

(a) vs. Size in LAMBADA

(b) vs. Params in LAMBADA

Figure 14: Performance of ternary TriLMs and BiLMs models on LAMBADA tasks across Size (Bits) and Parameters. Refer to Tables 7, and 8 for details.

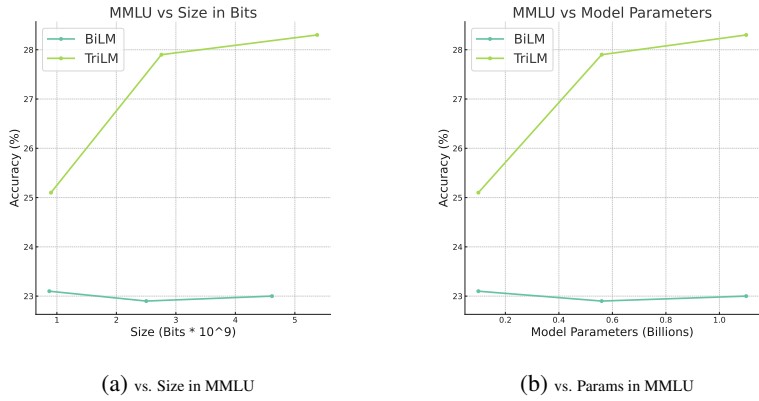

(a) vs. Size in MMLU

(b) vs. Params in MMLU

Figure 15: Performance of ternary TriLMs and BiLMs models on MMLU across Size (Bits) and Parameters. Refer to Tables 15 for details.

## C  SCALING LAW

In this section, we provide additional insights into the scaling fits discussed in Section 4.3. In addition to fitting a power law with an offset, we also explore a standard power law following Kaplan et al. (2020). Our findings suggest that the standard power law fits are slightly less precise than those incorporating an offset term. However, both models indicate a decreasing difference in validation loss as the number of parameters ($N$) increases.

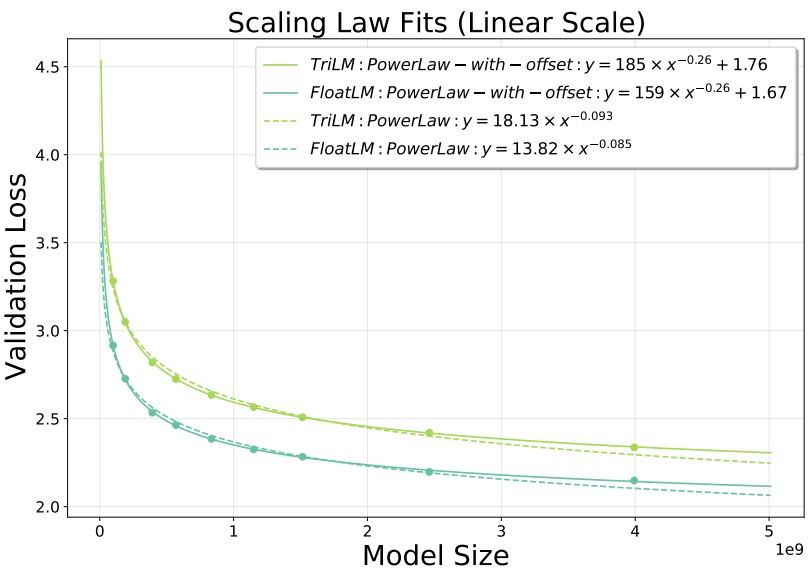

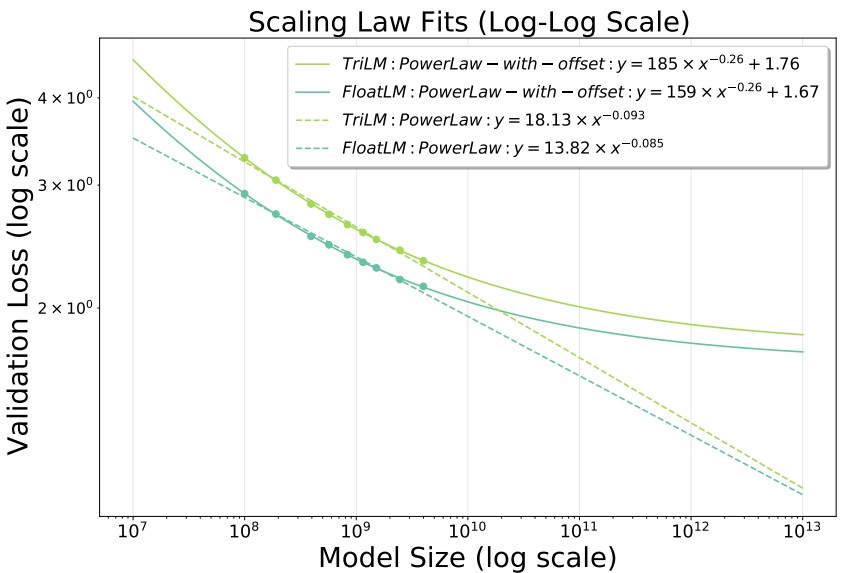

Figure 16: Comparison of Power Law and Power Law-with-offset Fits for TriLM and FloatLM.

As shown in Figure 17, using the scaling equations for TriLMs and FloatLMs, we derive the relationship between parameter count and the percentage difference in validation loss relative to FloatLMs. We observe that at 330B and 15.6B parameters, the validation losses for TriLMs are

within 6% and 7% of FloatLMs' validation losses, respectively. This indicates that TriLMs are likely to closely match the performance of FloatLMs at larger scales.

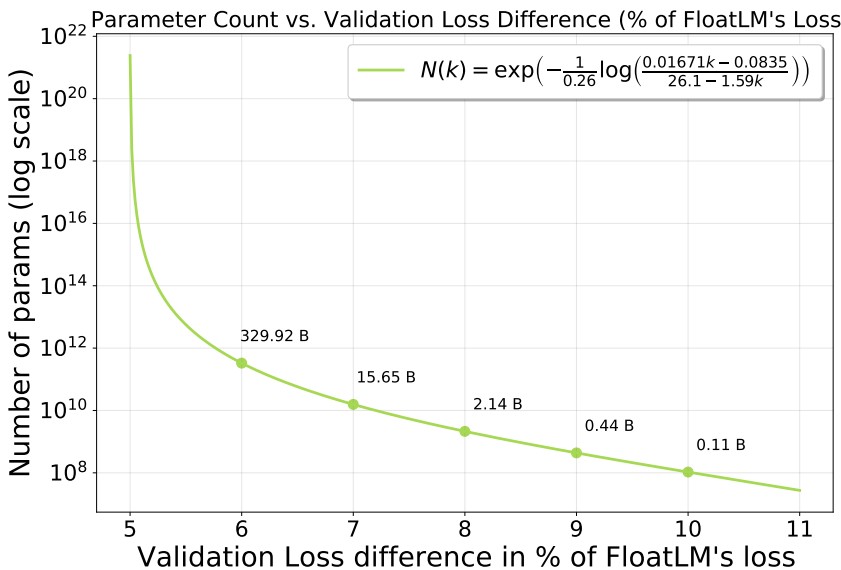

Figure 17: Comparison of Power Law and Power Law-with-offset Fits for TriLM and FloatLM.

## D    BENCHMARK DETAILS

We benchmark TriLM, FloatLM and QuantLM across Knowledge, Commonsense, Reasoning and Toxicity benchmarks. We average our scores across 3 different 'seeds' by preparing three different QuantLM models quantized using different calibration sets. We also add Pythia (deduplicated with consistent $2M$ batch size across families) suite of models (70M to 2.8B params) and BitNet b.158 performance scores from their paper for comparison. We use the LM Evaluation Harness (Gao et al., 2023) to benchmark.

### D.1    COMMONSENSE AND REASONING

We report commonsense and reasoning benchmark scores across 6 benchmarks previously considered by BitNet b.158 in Table 7, 8 and rest in Table 10. Each is considered in a zero-shot setting. Following are the details of each of the benchmarks considered:

- **ARC Challenge and Easy**: (Clark et al., 2018) ARC dataset comprises 7787 multiple-choice science questions divided into two sets: Challenge and Easy. We calculate accuracy and normalised accuracy across both of these sets.

- **BoolQ**: (Clark et al., 2019) BoolQ is a reading comprehension dataset consisting of naturally occurring yes/no questions. We calculate the accuracy of this task.

- **HellaSwag**: (Zellers et al., 2019) HellaSWAG is a dataset of multiple choice questions for testing grounded commonsense. The incorrect options are generated through Adversarial Filtering (AF) to fool machines but not humans. We calculate the accuracy and normalized accuracy on this task.

- **WinoGrande**: (Sakaguchi et al., 2021) WinoGrande is a collection of 44k problems for testing commonsense reasoning formulated as a fill-in-a-blank task with binary options. We report the accuracy on this task.

- **PIQA**: (Bisk et al., 2019) Physical Interaction Question Answering (PIQA) is a physical commonsense reasoning benchmark dataset to test the physical knowledge of language models. We calculate the accuracy and normalized accuracy on this task.

- **LAMBADA OpenAI**: (Paperno et al., 2016) LAMBADA is a dataset to evaluate text understanding by next-word prediction. It is a collection of narrative passages BooksCorpus To succeed on LAMBADA, models must integrate broader discourse information, not solely rely on local context. We calculate the perplexity and the accuracy of the model on this task.

- **LogiQA**: (Liu et al., 2021) LogiQA is a dataset for testing human logical reasoning. It contains questions spanning multiple types of deductive reasoning. We calculate the accuracy and normalized accuracy on this task.

### D.2    KNOWLEDGE

We report performance on SciQ, TriviaQA in Tables 10, 15 and 16. Each is considered in a zero-shot setting. Following are the details of each of the benchmarks considered:

The knowledge-based evaluation included the following tasks:

- **SciQ**: (Welbl et al., 2017) The SciQ dataset contains multiple-choice questions with 4 answer options from crowd-sourced science exams. The questions range from Physics, Chemistry and Biology and several other fields. We calculate the accuracy and length normalized accuracy on this task.

- **TriviaQA**: (Joshi et al., 2017) TriviaQA is a reading comprehension dataset containing question-answer-evidence triples. We calculate the exact match accuracy on this task.

- **MMLU** (Hendrycks et al., 2021): The benchmark aims to assess the knowledge gained during pretraining by evaluating models solely in zero-shot and few-shot scenarios. It spans 57 subjects, including STEM fields, humanities, social sciences, and more.

| Models | Arc Challenge | | Arc Easy | | BoolQ |
| --- | --- | --- | --- | --- | --- |
| | Acc Norm. | Acc | Acc Norm. | Acc | Acc |
| Pythia 70M | 22.0± 1.2 | 22.1± 1.2 | 24.8± 0.9 | 24.8± 0.9 | 38.5± 0.9 |
| FloatLM 99M | 23.8± 1.2 | 19.9± 1.2 | 39.1± 1.0 | 45.1± 1.0 | 58.2± 0.9 |
| QuantLM 99M 8-Bit | 23.8± 1.2 | 19.6± 1.2 | 39.4± 1.0 | 45.3± 1.0 | 58.5± 0.9 |
| QuantLM 99M 6-Bit | 23.2± 1.2 | 19.7± 1.2 | 38.8± 1.0 | 44.8± 1.0 | 58.9± 0.9 |
| QuantLM 99M 4-Bit | 22.6± 1.2 | 18.0± 1.1 | 37.1± 1.0 | 41.7± 1.0 | 52.2± 0.9 |
| QuantLM 99M 3-Bit | 23.2± 1.2 | 19.5± 1.2 | 34.8± 1.0 | 36.1± 1.0 | 48.4± 0.9 |
| TriLM 99M | 24.1± 1.3 | 19.1± 1.1 | 36.6± 1.0 | 39.8± 1.0 | 61.3± 0.9 |
| Binary 99M | 20.8± 1.2 | 18.3± 1.1 | 35.8± 0.9 | 40.1± 1.0 | 61.0± 0.8 |
| Pythia 160M | 23.8± 1.2 | 23.1± 1.2 | 26.7± 0.9 | 26.6± 0.9 | 38.3± 0.9 |
| FloatLM 190M | 24.1± 1.3 | 20.5± 1.2 | 43.0± 1.0 | 48.4± 1.0 | 59.1± 0.9 |
| QuantLM 190M 8-Bit | 24.4± 1.3 | 20.3± 1.2 | 43.0± 1.0 | 48.5± 1.0 | 59.3± 0.9 |
| QuantLM 190M 6-Bit | 23.8± 1.2 | 20.0± 1.2 | 42.0± 1.0 | 48.0± 1.0 | 59.1± 0.9 |
| QuantLM 190M 4-Bit | 25.2± 1.3 | 19.9± 1.2 | 26.5± 0.9 | 26.8± 0.9 | 40.9± 0.9 |
| QuantLM 190M 3-Bit | 22.5± 1.2 | 19.4± 1.2 | 37.1± 1.0 | 39.7± 1.0 | 56.5± 0.9 |
| TriLM 190M | 23.0± 1.2 | 19.5± 1.2 | 39.6± 1.0 | 43.9± 1.0 | 46.8± 0.9 |
| FloatLM 390M | 24.7± 1.3 | 21.3± 1.2 | 46.5± 1.0 | 51.0± 1.0 | 54.7± 0.9 |
| QuantLM 390M 8-Bit | 24.6± 1.3 | 21.2± 1.2 | 46.6± 1.0 | 51.0± 1.0 | 54.6± 0.9 |
| QuantLM 390M 6-Bit | 24.8± 1.3 | 21.5± 1.2 | 46.8± 1.0 | 51.8± 1.0 | 55.3± 0.9 |
| QuantLM 390M 4-Bit | 25.1± 1.3 | 21.3± 1.2 | 45.2± 1.0 | 49.6± 1.0 | 50.8± 0.9 |
| QuantLM 390M 3-Bit | 24.9± 1.3 | 21.5± 1.2 | 41.6± 1.0 | 43.6± 1.0 | 56.3± 0.9 |
| TriLM 390M | 24.5± 1.3 | 21.2± 1.2 | 44.1± 1.0 | 48.6± 1.0 | 55.1± 0.9 |
| Pythia 410M | 24.7± 1.3 | 21.2± 1.2 | 45.7± 1.0 | 51.6± 1.0 | 60.0± 0.9 |
| FloatLM 560M | 26.5± 1.3 | 23.9± 1.2 | 48.4± 1.0 | 54.4± 1.0 | 57.9± 0.9 |
| QuantLM 560M 8-Bit | 26.5± 1.3 | 23.6± 1.2 | 48.3± 1.0 | 54.1± 1.0 | 57.6± 0.9 |
| QuantLM 560M 6-Bit | 26.0± 1.3 | 23.5± 1.2 | 47.6± 1.0 | 54.2± 1.0 | 57.3± 0.9 |
| QuantLM 560M 4-Bit | 25.9± 1.3 | 23.0± 1.2 | 46.3± 1.0 | 52.4± 1.0 | 58.8± 0.9 |
| QuantLM 560M 3-Bit | 24.0± 1.2 | 21.2± 1.2 | 42.3± 1.0 | 45.8± 1.0 | 59.0± 0.9 |
| TriLM 560M | 25.7± 1.3 | 21.0± 1.2 | 45.5± 1.0 | 50.2± 1.0 | 57.3± 0.9 |
| Binary 560M | 24.6± 1.2 | 20.2± 1.1 | 41.9± 1.0 | 47.8± 1.0 | 61.5± 0.8 |
| FloatLM 830M | 28.0± 1.3 | 24.5± 1.3 | 51.6± 1.0 | 57.3± 1.0 | 61.0± 0.9 |
| QuantLM 830M 8-Bit | 28.2± 1.3 | 25.1± 1.3 | 51.7± 1.0 | 57.3± 1.0 | 60.9± 0.9 |
| QuantLM 830M 6-Bit | 27.6± 1.3 | 24.7± 1.3 | 51.6± 1.0 | 57.7± 1.0 | 61.3± 0.9 |
| QuantLM 830M 4-Bit | 27.6± 1.3 | 23.3± 1.2 | 50.5± 1.0 | 56.2± 1.0 | 58.1± 0.9 |
| QuantLM 830M 3-Bit | 27.1± 1.3 | 22.7± 1.2 | 46.8± 1.0 | 50.5± 1.0 | 56.3± 0.9 |
| TriLM 830M | 25.3± 1.3 | 22.5± 1.2 | 48.7± 1.0 | 54.2± 1.0 | 60.4± 0.9 |
| Pythia 1B | 27.0± 1.3 | 24.4± 1.3 | 49.0± 1.0 | 57.0± 1.0 | 60.8± 0.9 |
| FloatLM 1.1B | 29.1± 1.3 | 26.1± 1.3 | 54.0± 1.0 | 60.4± 1.0 | 62.9± 0.8 |
| QuantLM 1.1B 8-Bit | 28.9± 1.3 | 26.1± 1.3 | 54.1± 1.0 | 60.2± 1.0 | 62.6± 0.8 |
| QuantLM 1.1B 6-Bit | 29.8± 1.3 | 25.5± 1.3 | 54.3± 1.0 | 60.2± 1.0 | 62.9± 0.8 |
| QuantLM 1.1B 4-Bit | 30.3± 1.3 | 26.0± 1.3 | 53.6± 1.0 | 59.0± 1.0 | 61.3± 0.9 |
| QuantLM 1.1B 3-Bit | 29.2± 1.3 | 27.0± 1.3 | 48.9± 1.0 | 55.0± 1.0 | 62.1± 0.8 |
| TriLM 1.1B | 26.5± 1.3 | 24.6± 1.3 | 49.8± 1.0 | 56.3± 1.0 | 59.1± 0.9 |
| Binary 1.1B | 24.8± 1.3 | 22.3± 1.2 | 46.1± 1.0 | 52.7± 1.0 | 56.3± 0.9 |
| Pythia 1.4B | 28.7± 1.3 | 26.0± 1.3 | 54.0± 1.0 | 60.4± 1.0 | 63.2± 0.8 |
| FloatLM 1.5B | 29.7± 1.3 | 26.2± 1.3 | 56.4± 1.0 | 62.6± 1.0 | 63.2± 0.8 |
| QuantLM 1.5B 8-Bit | 29.8± 1.3 | 26.0± 1.3 | 56.6± 1.0 | 62.4± 1.0 | 63.3± 0.8 |
| QuantLM 1.5B 6-Bit | 30.1± 1.3 | 26.0± 1.3 | 56.8± 1.0 | 62.2± 1.0 | 63.4± 0.8 |
| QuantLM 1.5B 4-Bit | 29.4± 1.3 | 26.9± 1.3 | 55.2± 1.0 | 60.4± 1.0 | 62.5± 0.8 |
| QuantLM 1.5B 3-Bit | 27.8± 1.3 | 25.2± 1.3 | 49.7± 1.0 | 54.8± 1.0 | 53.7± 0.9 |
| TriLM 1.5B | 28.2± 1.3 | 24.7± 1.3 | 53.1± 1.0 | 59.0± 1.0 | 54.1± 0.9 |
| FloatLM 2.4B | 32.7± 1.4 | 30.1± 1.3 | 60.5± 1.0 | 65.5± 1.0 | 62.1± 0.8 |
| QuantLM 2.4B 8-Bit | 32.6± 1.4 | 30.0± 1.3 | 60.3± 1.0 | 65.7± 1.0 | 62.1± 0.8 |
| QuantLM 2.4B 6-Bit | 32.7± 1.4 | 30.6± 1.3 | 60.4± 1.0 | 65.4± 1.0 | 62.0± 0.8 |
| QuantLM 2.4B 4-Bit | 33.3± 1.4 | 30.8± 1.3 | 59.6± 1.0 | 64.1± 1.0 | 59.0± 0.9 |
| QuantLM 2.4B 3-Bit | 29.7± 1.3 | 28.4± 1.3 | 54.2± 1.0 | 58.4± 1.0 | 55.7± 0.9 |
| TriLM 2.4B | 29.9± 1.3 | 29.5± 1.3 | 58.0± 1.0 | 63.8± 1.0 | 64.4± 0.8 |
| FloatLM 3.9B | 34.6± 1.4 | 32.1± 1.4 | 63.0± 1.0 | 68.3± 1.0 | 65.9± 0.8 |
| QuantLM 3.9B 8-Bit | 34.6± 1.4 | 31.9± 1.4 | 63.0± 1.0 | 68.1± 1.0 | 65.4± 0.8 |
| QuantLM 3.9B 6-Bit | 35.1± 1.4 | 32.1± 1.4 | 63.3± 1.0 | 68.0± 1.0 | 65.6± 0.8 |
| QuantLM 3.9B 4-Bit | 34.7± 1.4 | 32.9± 1.4 | 61.2± 1.0 | 68.3± 1.0 | 65.4± 0.8 |
| QuantLM 3.9B 3-Bit | 32.1± 1.4 | 29.3± 1.3 | 55.5± 1.0 | 62.1± 1.0 | 60.0± 0.9 |
| TriLM 3.9B | 35.3± 1.4 | 31.9± 1.4 | 60.8± 1.0 | 66.0± 1.0 | 66.5± 0.8 |

Table 7: Spectra Suite Performance (Part 1): Arc Challenge, Arc Easy, and BoolQ. Additionally, we also include scores on the Pythia LLM suite.

| Models | HellaSwag | | PIQA | | WinoGrande | Avg ( HellaSwag, PIQA, WinoGrande, |
|---|---|---|---|---|---|---|
| | Acc Norm. | Acc | Acc Norm. | Acc | Acc | Arc Easy, Arc Challenge, and BoolQ) |
| Pythia 70M | 25.1± 0.4 | 25.1± 0.4 | 49.8± 1.2 | 49.9± 1.2 | 49.1± 1.4 | 34.9 |
| FloatLM 99M | 31.6± 0.5 | 29.1± 0.5 | 62.8± 1.1 | 63.2± 1.1 | 50.2± 1.4 | 44.3 |
| QuantLM 99M 8-Bit | 31.7± 0.5 | 29.0± 0.5 | 62.6± 1.1 | 63.0± 1.1 | 50.0± 1.4 | 44.3 |
| QuantLM 99M 6-Bit | 31.7± 0.5 | 29.2± 0.5 | 62.8± 1.1 | 63.1± 1.1 | 50.2± 1.4 | 44.3 |
| QuantLM 99M 4-Bit | 31.0± 0.5 | 28.9± 0.5 | 62.2± 1.1 | 60.9± 1.1 | 50.4± 1.4 | 42.6 |
| QuantLM 99M 3-Bit | 29.2± 0.5 | 27.7± 0.4 | 57.2± 1.2 | 58.2± 1.2 | 49.2± 1.4 | 40.3 |
| TriLM 99M | 28.4± 0.5 | 27.6± 0.4 | 60.1± 1.1 | 60.4± 1.1 | 50.7± 1.4 | 43.5 |
| Binary 99M | 27.7± 0.4 | 27.2± 0.4 | 59.2± 1.1 | 59.2± 1.1 | 48.8± 1.4 | 39.8 |
| Pythia 160M | 25.1± 0.4 | 25.0± 0.4 | 53.1± 1.2 | 53.1± 1.2 | 47.3± 1.4 | 35.7 |
| FloatLM 190M | 36.6± 0.5 | 31.4± 0.5 | 65.6± 1.1 | 64.8± 1.1 | 51.9± 1.4 | 46.7 |
| QuantLM 190M 8-Bit | 36.5± 0.5 | 31.4± 0.5 | 65.6± 1.1 | 64.8± 1.1 | 51.7± 1.4 | 46.8 |
| QuantLM 190M 6-Bit | 36.3± 0.5 | 31.5± 0.5 | 65.6± 1.1 | 64.3± 1.1 | 51.9± 1.4 | 46.4 |
| QuantLM 190M 4-Bit | 26.0± 0.4 | 25.7± 0.4 | 49.3± 1.2 | 51.7± 1.2 | 51.0± 1.4 | 36.5 |
| QuantLM 190M 3-Bit | 32.0± 0.5 | 28.8± 0.5 | 58.1± 1.2 | 58.7± 1.1 | 50.1± 1.4 | 42.7 |
| TriLM 190M | 31.6± 0.5 | 29.0± 0.5 | 62.0± 1.1 | 62.3± 1.1 | 51.7± 1.4 | 42.4 |
| FloatLM 390M | 44.4± 0.5 | 35.7± 0.5 | 68.7± 1.1 | 68.4± 1.1 | 51.8± 1.4 | 48.5 |
| QuantLM 390M 8-Bit | 44.5± 0.5 | 35.7± 0.5 | 68.8± 1.1 | 68.6± 1.1 | 52.6± 1.4 | 48.6 |
| QuantLM 390M 6-Bit | 44.2± 0.5 | 35.6± 0.5 | 69.0± 1.1 | 68.4± 1.1 | 53.0± 1.4 | 48.9 |
| QuantLM 390M 4-Bit | 43.4± 0.5 | 35.1± 0.5 | 68.1± 1.1 | 68.3± 1.1 | 53.7± 1.4 | 47.7 |
| QuantLM 390M 3-Bit | 39.5± 0.5 | 32.9± 0.5 | 63.8± 1.1 | 63.2± 1.1 | 53.0± 1.4 | 46.5 |
| TriLM 390M | 37.9± 0.5 | 32.0± 0.5 | 64.7± 1.1 | 65.0± 1.1 | 52.2± 1.4 | 46.4 |
| Pythia 410M | 40.3± 0.5 | 33.8± 0.5 | 67.2± 1.1 | 66.3± 1.1 | 53.5± 1.4 | 48.6 |
| FloatLM 560M | 47.6± 0.5 | 37.7± 0.5 | 68.8± 1.1 | 69.0± 1.1 | 53.7± 1.4 | 50.5 |
| QuantLM 560M 8-Bit | 47.6± 0.5 | 37.7± 0.5 | 68.9± 1.1 | 68.9± 1.1 | 53.8± 1.4 | 50.4 |
| QuantLM 560M 6-Bit | 47.6± 0.5 | 37.7± 0.5 | 68.7± 1.1 | 68.8± 1.1 | 53.5± 1.4 | 50.1 |
| QuantLM 560M 4-Bit | 46.7± 0.5 | 37.0± 0.5 | 67.8± 1.1 | 67.1± 1.1 | 53.1± 1.4 | 49.8 |
| QuantLM 560M 3-Bit | 41.7± 0.5 | 33.4± 0.5 | 63.5± 1.1 | 63.2± 1.1 | 49.7± 1.4 | 46.7 |
| TriLM 560M | 41.5± 0.5 | 33.8± 0.5 | 67.2± 1.1 | 67.5± 1.1 | 53.1± 1.4 | 48.4 |
| Binary 560M | 36.4± 0.4 | 31.2± 0.4 | 64.6± 1.1 | 64.2± 1.1 | 52.8± 1.4 | 44.5 |
| FloatLM 830M | 51.3± 0.5 | 40.1± 0.5 | 71.4± 1.1 | 71.7± 1.1 | 56.4± 1.4 | 53.3 |
| QuantLM 830M 8-Bit | 51.4± 0.5 | 40.1± 0.5 | 71.2± 1.1 | 71.7± 1.1 | 55.9± 1.4 | 53.2 |
| QuantLM 830M 6-Bit | 51.5± 0.5 | 40.2± 0.5 | 71.3± 1.1 | 71.8± 1.0 | 56.2± 1.4 | 53.2 |
| QuantLM 830M 4-Bit | 50.2± 0.5 | 39.2± 0.5 | 70.6± 1.1 | 71.1± 1.1 | 56.0± 1.4 | 52.2 |
| QuantLM 830M 3-Bit | 45.5± 0.5 | 35.9± 0.5 | 66.1± 1.1 | 66.6± 1.1 | 53.5± 1.4 | 49.2 |
| TriLM 830M | 46.0± 0.5 | 36.8± 0.5 | 68.2± 1.1 | 68.4± 1.1 | 55.6± 1.4 | 50.7 |
| Pythia 1B | 47.2± 0.5 | 37.7± 0.5 | 69.3± 1.1 | 70.8± 1.1 | 53.2± 1.4 | 51.1 |
| FloatLM 1.1B | 55.2± 0.5 | 42.6± 0.5 | 72.2± 1.1 | 71.3± 1.1 | 56.3± 1.4 | 54.9 |
| QuantLM 1.1B 8-Bit | 55.2± 0.5 | 42.6± 0.5 | 72.1± 1.0 | 71.2± 1.1 | 56.2± 1.4 | 54.8 |
| QuantLM 1.1B 6-Bit | 54.9± 0.5 | 42.6± 0.5 | 71.9± 1.0 | 71.2± 1.1 | 56.1± 1.4 | 55.0 |
| QuantLM 1.1B 4-Bit | 54.9± 0.5 | 42.0± 0.5 | 71.6± 1.1 | 70.4± 1.1 | 54.8± 1.4 | 54.4 |
| QuantLM 1.1B 3-Bit | 51.3± 0.5 | 39.4± 0.5 | 69.4± 1.1 | 68.4± 1.1 | 54.8± 1.4 | 52.6 |
| TriLM 1.1B | 49.1± 0.5 | 38.8± 0.5 | 69.8± 1.1 | 69.3± 1.1 | 55.5± 1.4 | 51.6 |
| Binary 1.1B | 43.4± 0.5 | 35.1± 0.4 | 66.9± 1.1 | 68.3± 1.1 | 55.3± 1.4 | 47.1 |
| Pythia 1.4B | 52.0± 0.5 | 40.4± 0.5 | 70.8± 1.1 | 70.6± 1.1 | 57.1± 1.4 | 54.3 |
| FloatLM 1.5B | 57.8± 0.5 | 44.3± 0.5 | 73.9± 1.0 | 73.1± 1.0 | 59.4± 1.4 | 56.7 |
| QuantLM 1.5B 8-Bit | 57.8± 0.5 | 44.3± 0.5 | 73.7± 1.0 | 73.1± 1.0 | 59.4± 1.4 | 56.8 |
| QuantLM 1.5B 6-Bit | 57.5± 0.5 | 44.2± 0.5 | 74.0± 1.0 | 73.0± 1.0 | 59.7± 1.4 | 56.9 |
| QuantLM 1.5B 4-Bit | 56.9± 0.5 | 43.2± 0.5 | 72.7± 1.0 | 72.4± 1.0 | 57.1± 1.4 | 55.6 |
| QuantLM 1.5B 3-Bit | 53.7± 0.5 | 41.0± 0.5 | 70.0± 1.1 | 69.4± 1.1 | 55.0± 1.4 | 51.6 |
| TriLM 1.5B | 53.1± 0.5 | 40.9± 0.5 | 70.1± 1.1 | 70.3± 1.1 | 56.1± 1.4 | 52.5 |
| FloatLM 2.4B | 62.7± 0.5 | 47.1± 0.5 | 75.2± 1.0 | 74.9± 1.0 | 61.8± 1.4 | 59.2 |
| QuantLM 2.4B 8-Bit | 62.7± 0.5 | 47.1± 0.5 | 75.4± 1.0 | 74.9± 1.0 | 61.4± 1.4 | 59.1 |
| QuantLM 2.4B 6-Bit | 62.9± 0.5 | 47.0± 0.5 | 75.7± 1.0 | 74.7± 1.0 | 61.1± 1.4 | 59.1 |
| QuantLM 2.4B 4-Bit | 62.2± 0.5 | 46.5± 0.5 | 75.4± 1.0 | 74.5± 1.0 | 61.7± 1.4 | 58.5 |
| QuantLM 2.4B 3-Bit | 58.6± 0.5 | 43.5± 0.5 | 72.7± 1.0 | 70.8± 1.1 | 57.2± 1.4 | 54.7 |
| TriLM 2.4B | 59.0± 0.5 | 45.3± 0.5 | 72.6± 1.0 | 71.4± 1.1 | 59.7± 1.4 | 57.3 |
| FloatLM 3.9B | 66.1± 0.5 | 49.7± 0.5 | 75.8± 1.0 | 75.4± 1.0 | 62.8± 1.4 | 61.4 |
| QuantLM 3.9B 8-Bit | 66.0± 0.5 | 49.7± 0.5 | 75.9± 1.0 | 75.5± 1.0 | 62.9± 1.4 | 61.3 |
| QuantLM 3.9B 6-Bit | 65.9± 0.5 | 49.7± 0.5 | 75.5± 1.0 | 75.6± 1.0 | 62.2± 1.4 | 61.3 |
| QuantLM 3.9B 4-Bit | 65.0± 0.5 | 49.0± 0.5 | 75.5± 1.0 | 75.6± 1.0 | 62.7± 1.4 | 60.7 |
| QuantLM 3.9B 3-Bit | 61.2± 0.5 | 45.9± 0.5 | 72.6± 1.0 | 72.3± 1.0 | 59.3± 1.4 | 56.8 |
| TriLM 3.9B | 64.7± 0.5 | 48.3± 0.5 | 74.6± 1.0 | 74.4± 1.0 | 62.1± 1.4 | 60.7 |

Table 8: Spectra Suite Performance (Part 2): HellaSwag, PIQA, WinoGrande, and Average Scores (including Arc Easy, Arc Challenge, and BoolQ). Additionally, we include scores from the Pythia LLM suite.

| Models | Arc Challenge | | Arc Easy | | BoolQ | HellaSwag | | PIQA | | WinoGrande | Avg |
|--------|---------------|-----|----------|-----|-------|-----------|-----|------|-----|------------|-----|
| | Acc Norm. | Acc | Acc Norm. | Acc | Acc | Acc Norm. | Acc | Acc Norm. | Acc | Acc | |
| BitNet 700M | 21.4 | | 51.8 | | 58.2 | 35.1 | | 68.1 | | 55.2 | 48.3 |
| BitNet 1.3B | 24.2 | | 54.9 | | 56.7 | 37.7 | | 68.8 | | 55.8 | 49.7 |
| BitNet 3B | 28.3 | | 61.4 | | 61.5 | 42.9 | | 71.5 | | 59.3 | 54.2 |
| BitNet 3.9B | 28.7 | | 64.2 | | 63.5 | 44.2 | | 73.2 | | 60.5 | 55.7 |

Table 9: Performance of BitNet b1.58 on ARC Challenge, ARC Easy, BoolQ, HellaSwag, PIQA, and Wino-Grande. The scores are taken from (Ma et al., 2024).

## D.3 TOXICITY

We report toxicity-based evaluation in 15. Each is considered in a zero-shot setting.

The toxicity-based evaluation included the following tasks:

- **BBQ** (Parrish et al., 2022): The Bias Benchmark for QA (BBQ) dataset, comprises sets of questions developed by its authors, focusing on documented social biases directed towards individuals from protected classes across nine distinct social dimensions pertinent to U.S. English-speaking environments.
- **Crows Pairs** (Nangia et al., 2020): proposed a challenging dataset aimed at quantifying stereotypical biases embedded within language models, with a specific emphasis on U.S. contexts. Hosted on GitHub, this dataset serves as a crucial resource for assessing and addressing biases through paired sentences that illuminate societal stereotypes.
- **TruthfulQA** (Lin et al., 2021): A benchmark designed to evaluate the truthfulness of language models in generating responses to questions. This benchmark includes 817 questions across 38 categories, such as health, law, finance, and politics.

## D.4 PERPLEXITY ON OTHER DATASETS

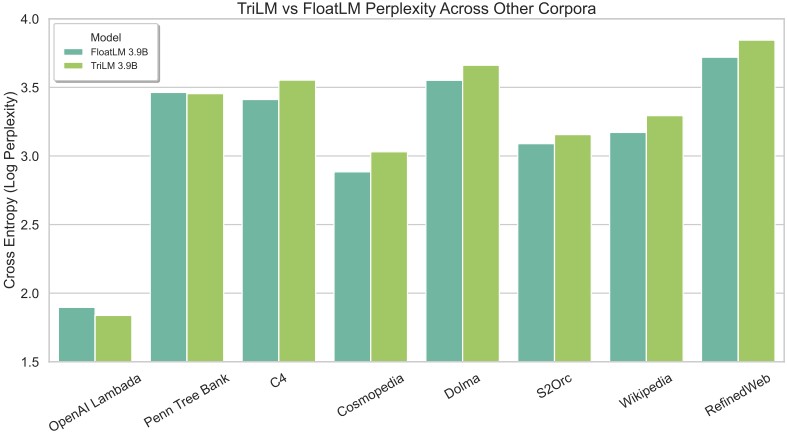

Figure 18: Cross-entropy (log perplexity) comparison between TriLM and FloatLM (both 3.9B parameters) across various datasets apart from SlimPajama.

We measure perplexity using TriLM 3.9B and FloatLM 3.9B across various other corpora than SlimPajama, which was used for training. These corpora include OpenAI Lambada, Penn Tree Bank, C4, Cosmopedia, Dolma, S2Orc, Wikipedia, and RefinedWeb. A portion of Wikipedia, C4 is included in Slim Pajama. Some other corpora like Dolma and RefinedWeb, may also have overlaps from C4, Wikipedia as well as Common Crawl.

Figure 18 demonstrates that while TriLM 3.9B is similar or better than FloatLM 3.9B on PTB and Lambada, across the other datasets, with potential overlaps with SlimPajama, it's performance is consistently worse - indicating lower capability to memorize training data as well as worse in-distribution performance, despite competitive out of distribution performance.

| Models | LAMBADA | | SciQ | | LogiQA | |
|---|---|---|---|---|---|---|
| | Perp. | Acc | Acc Norm. | Acc | Acc Norm. | Acc |
| FloatLM 99M | 85.0± 6.9 | 26.5± 0.6 | 62.9± 1.5 | 73.6± 1.4 | 27.6± 1.8 | 21.2± 1.6 |
| QuantLM 99M 8-Bit | 85.8± 7.0 | 26.6± 0.6 | 62.8± 1.5 | 73.7± 1.4 | 27.8± 1.8 | 21.0± 1.6 |
| QuantLM 99M 6-Bit | 89.9± 7.4 | 26.1± 0.6 | 61.8± 1.5 | 73.9± 1.4 | 28.1± 1.8 | 20.3± 1.6 |
| QuantLM 99M 4-Bit | 211.6± 17.3 | 16.7± 0.5 | 61.2± 1.5 | 70.7± 1.4 | 24.9± 1.7 | 20.7± 1.6 |
| QuantLM 99M 3-Bit | 4765.4± 413.0 | 4.5± 0.3 | 51.9± 1.6 | 57.0± 1.6 | 25.3± 1.7 | 19.8± 1.6 |
| TriLM 99M | 172.0± 8.4 | 20.0± 0.6 | 60.4± 1.5 | 67.6± 1.5 | 25.5± 1.7 | 21.5± 1.6 |
| Binary 99M | 468.3± 24.1 | 14.0± 0.4 | 54.4± 1.6 | 62.5± 1.5 | 27.0± 1.7 | 22.3± 1.6 |
| FloatLM 190M | 50.3± 2.7 | 31.1± 0.6 | 65.1± 1.5 | 77.3± 1.3 | 27.2± 1.7 | 22.1± 1.6 |
| QuantLM 190M 8-Bit | 48.7± 2.6 | 31.5± 0.6 | 65.5± 1.5 | 77.1± 1.3 | 27.0± 1.7 | 22.3± 1.6 |
| QuantLM 190M 6-Bit | 55.3± 3.0 | 30.0± 0.6 | 64.2± 1.5 | 77.0± 1.3 | 26.1± 1.7 | 22.4± 1.6 |
| QuantLM 190M 4-Bit | 72479077.3 | 0.00± 0.0 | 25.6± 1.4 | 22.9± 1.3 | 23.3± 1.7 | 20.7± 1.6 |
| QuantLM 190M 3-Bit | 664.5± 41.1 | 12.4± 0.5 | 58.5± 1.6 | 66.4± 1.5 | 26.3± 1.7 | 21.0± 1.6 |
| TriLM 190M | 130.7± 6.5 | 23.7± 0.6 | 61.0± 1.5 | 72.6± 1.4 | 25.5± 1.7 | 21.5± 1.6 |
| FloatLM 390M | 21.9± 0.9 | 42.2± 0.7 | 75.6± 1.4 | 84.2± 1.2 | 28.1± 1.8 | 23.8± 1.7 |
| QuantLM 390M 8-Bit | 21.7± 0.9 | 42.3± 0.7 | 75.7± 1.4 | 84.1± 1.2 | 28.3± 1.8 | 24.1± 1.7 |
| QuantLM 390M 6-Bit | 24.3± 1.0 | 40.6± 0.7 | 75.5± 1.4 | 83.7± 1.2 | 27.6± 1.8 | 23.2± 1.7 |
| QuantLM 390M 4-Bit | 30.2± 1.3 | 39.1± 0.7 | 77.1± 1.3 | 84.1± 1.2 | 25.8± 1.7 | 23.3± 1.7 |
| QuantLM 390M 3-Bit | 115.0± 5.6 | 23.0± 0.6 | 67.4± 1.5 | 76.7± 1.3 | 25.7± 1.7 | 21.8± 1.6 |
| TriLM 390M | 77.7± 3.8 | 28.0± 0.6 | 68.6± 1.5 | 76.9± 1.3 | 26.4± 1.7 | 21.8± 1.6 |
| FloatLM 560M | 20.8± 0.9 | 44.1± 0.7 | 74.7± 1.4 | 83.5± 1.2 | 27.0± 1.7 | 20.7± 1.6 |
| QuantLM 560M 8-Bit | 20.9± 0.9 | 44.2± 0.7 | 74.7± 1.4 | 83.6± 1.2 | 27.3± 1.7 | 20.7± 1.6 |
| QuantLM 560M 6-Bit | 21.7± 0.9 | 42.8± 0.7 | 74.4± 1.4 | 83.6± 1.2 | 25.8± 1.7 | 20.9± 1.6 |
| QuantLM 560M 4-Bit | 24.9± 1.1 | 40.8± 0.7 | 73.6± 1.4 | 82.0± 1.2 | 27.0± 1.7 | 21.7± 1.6 |
| QuantLM 560M 3-Bit | 146.3± 7.1 | 20.1± 0.6 | 71.1± 1.4 | 75.9± 1.4 | 25.0± 1.7 | 21.8± 1.6 |
| TriLM 560M | 55.6± 2.7 | 32.4± 0.7 | 70.8± 1.4 | 78.7± 1.3 | 26.1± 1.7 | 19.8± 1.6 |
| Binary 560M | 62.8± 3.0 | 31.0± 0.6 | 70.0± 1.4 | 78.8± 1.3 | 26.7± 1.7 | 21.5± 1.6 |
| FloatLM 830M | 13.3± 0.5 | 49.6± 0.7 | 78.4± 1.3 | 85.9± 1.1 | 26.3± 1.7 | 20.1± 1.6 |
| QuantLM 830M 8-Bit | 13.5± 0.5 | 49.4± 0.7 | 78.5± 1.3 | 86.1± 1.1 | 26.6± 1.7 | 20.0± 1.6 |
| QuantLM 830M 6-Bit | 13.3± 0.5 | 49.1± 0.7 | 77.8± 1.3 | 85.4± 1.1 | 26.3± 1.7 | 20.1± 1.6 |
| QuantLM 830M 4-Bit | 15.4± 0.6 | 47.3± 0.7 | 78.8± 1.3 | 85.1± 1.1 | 25.5± 1.7 | 21.2± 1.6 |
| QuantLM 830M 3-Bit | 47.7± 2.0 | 30.5± 0.6 | 74.1± 1.4 | 80.1± 1.3 | 28.1± 1.8 | 21.2± 1.6 |
| TriLM 830M | 26.0± 1.1 | 39.9± 0.7 | 75.4± 1.4 | 82.8± 1.2 | 27.6± 1.8 | 21.4± 1.6 |
| FloatLM 1.1B | 11.7± 0.4 | 51.2± 0.7 | 82.2± 1.2 | 88.1± 1.0 | 27.3± 1.7 | 20.9± 1.6 |
| QuantLM 1.1B 8-Bit | 11.7± 0.4 | 51.2± 0.7 | 82.1± 1.2 | 88.1± 1.0 | 27.8± 1.8 | 21.2± 1.6 |
| QuantLM 1.1B 6-Bit | 11.7± 0.4 | 50.0± 0.7 | 82.3± 1.2 | 88.1± 1.0 | 27.5± 1.8 | 21.5± 1.6 |
| QuantLM 1.1B 4-Bit | 13.9± 0.5 | 49.3± 0.7 | 81.2± 1.2 | 87.6± 1.0 | 28.4± 1.8 | 20.3± 1.6 |
| QuantLM 1.1B 3-Bit | 26.9± 1.1 | 39.1± 0.7 | 78.7± 1.3 | 85.0± 1.1 | 25.8± 1.7 | 20.7± 1.6 |
| TriLM 1.1B | 17.3± 0.7 | 46.2± 0.7 | 73.3± 1.4 | 81.9± 1.2 | 26.9± 1.7 | 22.0± 1.6 |
| Binary 1.1B | 33.4± 1.4 | 37.6± 0.6 | 71.1± 1.4 | 81.2± 1.3 | 28.4± 1.7 | 23.2± 1.6 |
| FloatLM 1.5B | 9.4± 0.3 | 55.5± 0.7 | 80.9± 1.2 | 87.4± 1.0 | 26.1± 1.7 | 20.9± 1.6 |
| QuantLM 1.5B 8-Bit | 9.5± 0.3 | 55.5± 0.7 | 81.3± 1.2 | 87.5± 1.0 | 25.7± 1.7 | 20.6± 1.6 |
| QuantLM 1.5B 6-Bit | 9.5± 0.3 | 55.4± 0.7 | 81.4± 1.2 | 87.6± 1.0 | 25.7± 1.7 | 20.3± 1.6 |
| QuantLM 1.5B 4-Bit | 10.4± 0.4 | 53.0± 0.7 | 81.1± 1.2 | 86.9± 1.1 | 25.7± 1.7 | 20.3± 1.6 |
| QuantLM 1.5B 3-Bit | 17.8± 0.7 | 45.3± 0.7 | 75.5± 1.4 | 82.1± 1.2 | 28.4± 1.8 | 22.7± 1.6 |
| TriLM 1.5B | 16.4± 0.7 | 46.2± 0.7 | 80.7± 1.2 | 87.3± 1.1 | 27.8± 1.8 | 21.5± 1.6 |
| FloatLM 2.4B | 7.7± 0.3 | 59.3± 0.7 | 87.2± 1.1 | 91.0± 0.9 | 29.5± 1.8 | 21.5± 1.6 |
| QuantLM 2.4B 8-Bit | 7.7± 0.3 | 59.2± 0.7 | 87.1± 1.1 | 91.0± 0.9 | 29.5± 1.8 | 21.5± 1.6 |
| QuantLM 2.4B 6-Bit | 7.9± 0.3 | 58.9± 0.7 | 87.3± 1.1 | 90.9± 0.9 | 29.6± 1.8 | 20.9± 1.6 |
| QuantLM 2.4B 4-Bit | 8.9± 0.3 | 56.1± 0.7 | 84.8± 1.1 | 89.7± 1.0 | 29.6± 1.8 | 20.9± 1.6 |
| QuantLM 2.4B 3-Bit | 15.6± 0.6 | 45.0± 0.7 | 79.9± 1.3 | 86.7± 1.1 | 28.6± 1.8 | 21.4± 1.6 |
| TriLM 2.4B | 8.6± 0.3 | 55.7± 0.7 | 84.2± 1.2 | 88.7± 1.0 | 28.6± 1.8 | 24.3± 1.7 |
| FloatLM 3.9B | 6.7± 0.2 | 61.1± 0.7 | 86.5± 1.1 | 90.9± 0.9 | 26.9± 1.7 | 20.9± 1.6 |
| QuantLM 3.9B 8-Bit | 6.7± 0.2 | 61.1± 0.7 | 86.2± 1.1 | 91.0± 0.9 | 26.6± 1.7 | 20.6± 1.6 |
| QuantLM 3.9B 6-Bit | 6.8± 0.2 | 60.8± 0.7 | 86.6± 1.1 | 91.3± 0.9 | 25.8± 1.7 | 20.4± 1.6 |
| QuantLM 3.9B 4-Bit | 7.4± 0.2 | 58.5± 0.7 | 86.1± 1.1 | 90.8± 0.9 | 28.6± 1.8 | 20.1± 1.6 |
| QuantLM 3.9B 3-Bit | 14.0± 0.5 | 47.1± 0.7 | 83.1± 1.2 | 88.6± 1.0 | 27.0± 1.7 | 21.5± 1.6 |
| TriLM 3.9B | 6.3± 0.2 | 61.6± 0.7 | 87.4± 1.0 | 90.8± 0.9 | 27.6± 1.8 | 22.7± 1.6 |

Table 10: Spectra Suite Performance (Part 3): LAMBADA OpenAI, SciQ, LogiQA. We additionally also include Pythia's performance scores.

## D.5   OMNIQUANT: 3-BIT POST-TRAINING QUANTIZED MODELS

We present additional 3-bit quantized (QuantLM 3-bit) models, which were trained using OmniQuant (Shao et al., 2024) for 5 iterations with a group size of one row. It is important to highlight that the performance benchmarks for these models are consistent with those of GPT-Q, as reported in Tables 7, 8, 10, 15, and 16. Therefore, these results do not affect our findings in the main paper.

| Tasks | Metric | 99M | 190M | 390M | 560M | 1.1B |
|---|---|---|---|---|---|---|
| ARC Challenge | Acc. | $0.19 \pm 0.01$ | $0.21 \pm 0.01$ | $0.22 \pm 0.01$ | $0.22 \pm 0.01$ | $0.24 \pm 0.01$ |
| | Acc. (Norm.) | $0.23 \pm 0.01$ | $0.24 \pm 0.01$ | $0.26 \pm 0.01$ | $0.24 \pm 0.01$ | $0.26 \pm 0.01$ |
| ARC Easy | Acc. | $0.37 \pm 0.01$ | $0.41 \pm 0.01$ | $0.45 \pm 0.01$ | $0.48 \pm 0.01$ | $0.55 \pm 0.01$ |
| | Acc. (Norm.) | $0.34 \pm 0.01$ | $0.38 \pm 0.01$ | $0.41 \pm 0.01$ | $0.42 \pm 0.01$ | $0.48 \pm 0.01$ |
| BoolQ | Acc. | $0.55 \pm 0.01$ | $0.54 \pm 0.01$ | $0.62 \pm 0.01$ | $0.57 \pm 0.01$ | $0.63 \pm 0.01$ |
| HellaSwag | Acc. | $0.28 \pm 0.00$ | $0.29 \pm 0.00$ | $0.32 \pm 0.00$ | $0.34 \pm 0.00$ | $0.38 \pm 0.00$ |
| | Acc. (Norm.) | $0.29 \pm 0.00$ | $0.32 \pm 0.00$ | $0.39 \pm 0.00$ | $0.41 \pm 0.00$ | $0.48 \pm 0.01$ |
| LAMBADA | Acc. | $0.05 \pm 0.00$ | $0.07 \pm 0.00$ | $0.23 \pm 0.01$ | $0.25 \pm 0.01$ | $0.31 \pm 0.01$ |
| LogiQA | Acc. | $0.16 \pm 0.01$ | $0.20 \pm 0.02$ | $0.23 \pm 0.02$ | $0.23 \pm 0.02$ | $0.21 \pm 0.02$ |
| | Acc. (Norm.) | $0.23 \pm 0.02$ | $0.25 \pm 0.02$ | $0.29 \pm 0.02$ | $0.28 \pm 0.02$ | $0.27 \pm 0.02$ |
| PIQA | Acc. | $0.58 \pm 0.01$ | $0.59 \pm 0.01$ | $0.63 \pm 0.01$ | $0.65 \pm 0.01$ | $0.69 \pm 0.01$ |
| | Acc. (Norm.) | $0.58 \pm 0.01$ | $0.59 \pm 0.01$ | $0.63 \pm 0.01$ | $0.66 \pm 0.01$ | $0.69 \pm 0.01$ |
| SciQ | Acc. | $0.55 \pm 0.02$ | $0.64 \pm 0.02$ | $0.74 \pm 0.01$ | $0.77 \pm 0.01$ | $0.82 \pm 0.01$ |
| | Acc. (Norm.) | $0.49 \pm 0.02$ | $0.59 \pm 0.02$ | $0.61 \pm 0.02$ | $0.68 \pm 0.01$ | $0.73 \pm 0.01$ |
| TriviaQA | Exact Match | $0.00 \pm 0.00$ | $0.00 \pm 0.00$ | $0.01 \pm 0.00$ | $0.01 \pm 0.00$ | $0.05 \pm 0.00$ |
| Winogrande | Acc. | $0.51 \pm 0.01$ | $0.50 \pm 0.01$ | $0.51 \pm 0.01$ | $0.53 \pm 0.01$ | $0.56 \pm 0.01$ |
| MMLU | Acc. | $0.25 \pm 0.00$ | $0.25 \pm 0.00$ | $0.26 \pm 0.00$ | $0.27 \pm 0.00$ | $0.29 \pm 0.00$ |

Table 11: Model performance of 3-bit post-training quantized models using OmniQuant across various tasks, from 99M to 1.1B parameters, evaluated on different metrics.

| Tasks | Metric | 1.5B | 2.4B |
|---|---|---|---|
| ARC Challenge | Acc. | $0.26 \pm 0.01$ | $0.27 \pm 0.01$ |
| | Acc. (Norm.) | $0.29 \pm 0.01$ | $0.30 \pm 0.01$ |
| ARC Easy | Acc. | $0.56 \pm 0.01$ | $0.60 \pm 0.01$ |
| | Acc. (Norm.) | $0.51 \pm 0.01$ | $0.53 \pm 0.01$ |
| BoolQ | Acc. | $0.59 \pm 0.01$ | $0.63 \pm 0.01$ |
| HellaSwag | Acc. | $0.40 \pm 0.00$ | $0.44 \pm 0.00$ |
| | Acc. (Norm.) | $0.51 \pm 0.01$ | $0.57 \pm 0.00$ |
| LAMBADA | Acc. | $0.35 \pm 0.01$ | $0.44 \pm 0.01$ |
| LogiQA | Acc. | $0.23 \pm 0.02$ | $0.19 \pm 0.02$ |
| | Acc. (Norm.) | $0.26 \pm 0.02$ | $0.30 \pm 0.02$ |
| PIQA | Acc. | $0.69 \pm 0.01$ | $0.71 \pm 0.01$ |
| | Acc. (Norm.) | $0.70 \pm 0.01$ | $0.72 \pm 0.01$ |
| SciQ | Acc. | $0.82 \pm 0.01$ | $0.86 \pm 0.01$ |
| | Acc. (Norm.) | $0.76 \pm 0.01$ | $0.80 \pm 0.01$ |
| TriviaQA | Exact Match | $0.07 \pm 0.00$ | $0.07 \pm 0.00$ |
| Winogrande | Acc. | $0.56 \pm 0.01$ | $0.58 \pm 0.01$ |
| MMLU | Acc. | $0.28 \pm 0.00$ | $0.30 \pm 0.00$ |

Table 12: Performance of 3-bit Post-Training Quantized Models using OmniQuant across Various Tasks, from 1.5B to 2.4B parameters, across different tasks and metrics.

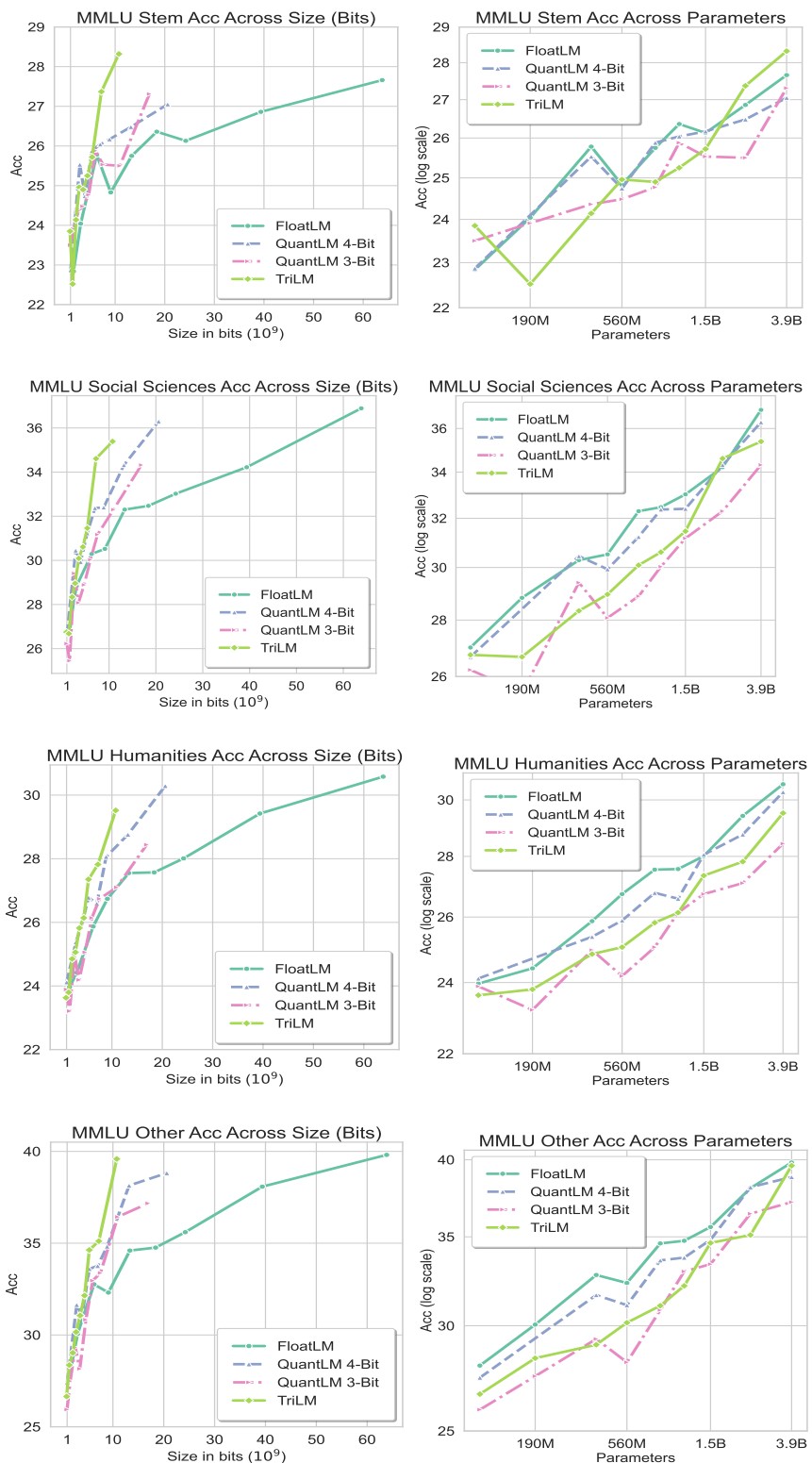

Figure 19: Model performance on MMLU subsets: STEM, Humanities, Social Sciences, and others. Plot accuracy scores against model size in bits (left) and number of parameters (right), ranging from 560M to 3.9B parameters for TriLM (ternary), FloatLM (FP16), and QuantLM (3-bit & 4-bit).

# E    WEIGHT DISTRIBUTION OF LINEAR LAYERS

Figure 20: Weight distribution in the linear layers of FloatLM models across various model sizes, ranging from 99M to 3.9B parameters

The observed Gaussian distribution in the weights of our final trained FloatLM models across various scales is supported by both theoretical foundations and empirical evidence. We outline the rationale for the normality of weight distributions as follows:

**Empirical Consistency:**    Across all model scales, ranging from 99 million to 3.9 billion parameters, the weight distributions consistently exhibit Gaussian characteristics, as illustrated in Figure 20. This consistency is visually evident in distribution plots and quantitatively confirmed by fitting Gaussian functions to the weight histograms, demonstrating a good fit across different model sizes.

**Theoretical Underpinning:**    Neural network weight initialization typically follows a Gaussian distribution to facilitate balanced learning dynamics. As training progresses, despite non-linear transformations and complex interactions within the network, the Central Limit Theorem suggests that the aggregation of numerous independent random variables (such as updates during backpropagation) tends toward a normal distribution. This tendency is particularly pronounced given the high dimensionality and extensive data processing involved in training large language models.

**Stabilization through Regularization and Optimization:**    Techniques such as L2 regularization constrain weight magnitudes, encouraging them towards smaller values and contributing to a peak around zero—a characteristic feature of Gaussian distributions. Additionally, optimization algorithms like Adam, which adjust learning rates based on moving averages of recent gradients, promote smoother updates. This approach maintains the Gaussian form by mitigating the impact of outlier gradients.

Given these, we can assert that the weight distribution of our trained models closely follows a Gaussian distribution which is crucial for understanding the weight variance across different scales.

## F    LEARNING DYNAMICS ANALYSIS OF PEAK LR DROP AND DECAY STAGE

In this section, we present a concise analysis of our learning dynamics, inspired by Appendix D of MiniCPM (Hu et al., 2024). Specifically, we examine the impact of our two key interventions: the peak learning rate reduction at the halfway point of training and the removal of weight decay regularization at two-thirds of the training stage. This analysis explores loss dynamics through the lens of checkpoint updates and gradient behavior.

We analyze both the latent weights and ternary weights, calculating the maximum weight element update, $\max_{ij}(W^{(t+1)}ij - W^{(t)}ij)$, across all weight matrices in the TriLM-1.5B model for each training step. Our findings, consistent with observations from MiniCPM, reveal a strong correlation between weight updates and the learning rate decay, as depicted in Figure 21. Notably, after the weight decay regularization is removed at approximately 160,000 training iterations, the model checkpoints undergo smaller updates. However, surprisingly during this stage, despite smaller weight changes, the loss decreases at an accelerated pace.

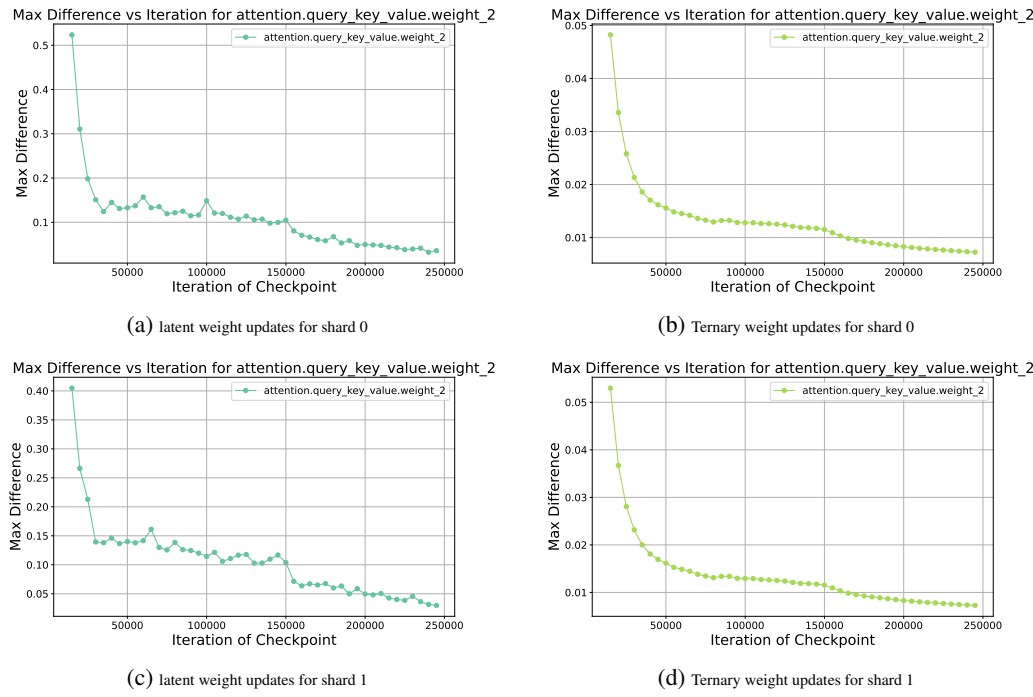

(a) latent weight updates for shard 0

(b) Ternary weight updates for shard 0

(c) latent weight updates for shard 1

(d) Ternary weight updates for shard 1

Figure 21: Max Difference of Checkpoints for Layer 2

Following the methodology of Hu et al. (2024), we examine gradient data by training a 190M parameter TriLM model. We record the gradients at every step to obtain fine-grained values for deriving second-order gradient information. In Figures 22 and 23, we plot the maximum, $L_2$ norm, first and second-order directional derivatives of the gradient, along with the cosine of the gradient angle and the loss curvature along the training trajectory.

While the $L_2$ norm and maximum of the gradient do not provide notable insights, the loss curvature and second-order directional derivative show significant correlations with both the peak learning rate reduction at 5000 steps and the removal of weight decay at 7500 steps. Interestingly, we also observe an increase in the cosine of the gradients and a decrease in the first-order directional derivative at both 5000 and 7500 steps, with trends becoming more pronounced after the removal of weight regularization. Moreover, the trends for loss curvature along the parameter trajectory and the second-order directional derivative are opposite for the latent and ternary gradient statistics. A more in dept understanding these trends in the learning dynamics under our current training strategy for TriLM offers a promising direction for future exploration.

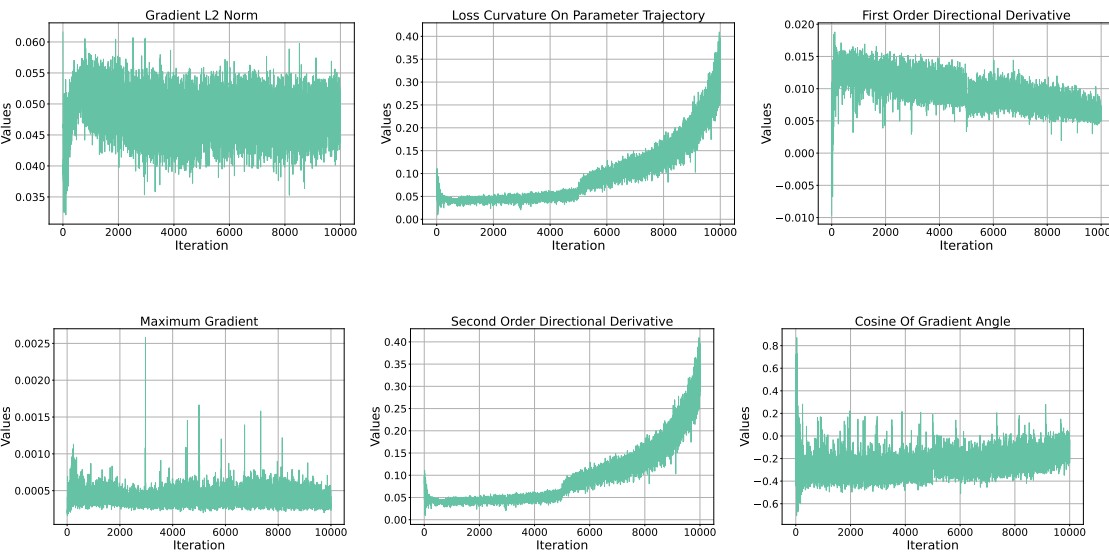

Figure 22: Latent gradient statistics over the training of 190M model. We drop the peaked learning rate at step 5000 and reduce the weight decay to 0 at step 7500

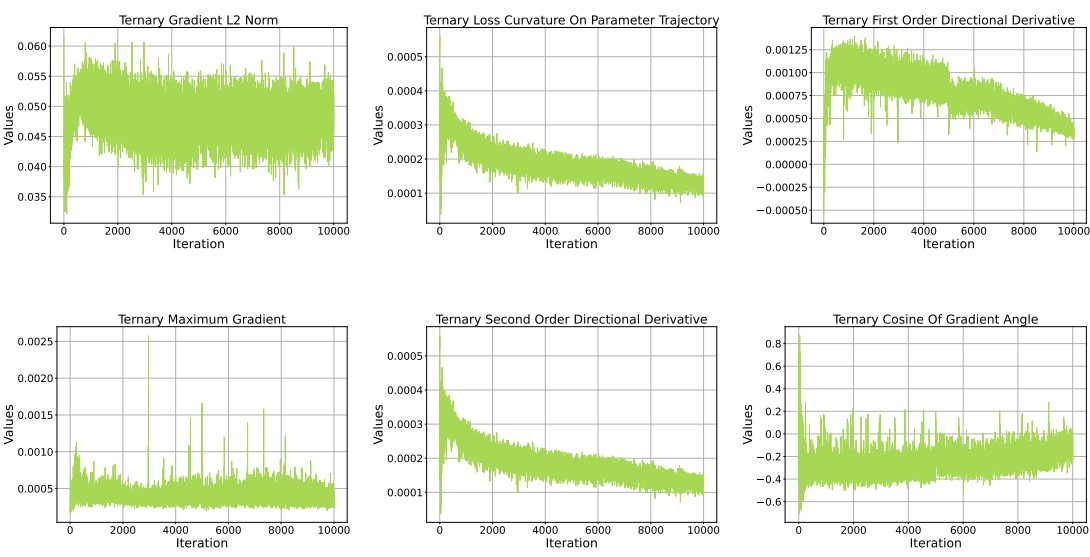

Figure 23: Ternary gradient statistics over the training of 190M model. We drop the peaked learning rate at step 5000 and reduce the weight decay to 0 at step 7500

## G   MEMORY BOTTLENECKS AND LOW-BITWIDTH LANGUAGE MODELLING

Recent observations (Gholami et al., 2024) suggest that, given the slower pace of improvements in memory and communication compared to compute (FLOPs), the bottleneck continues to shift away from computation towards the memory-related characteristics of hardware for deploying large language models. This shift underscores the importance of exploring solutions that directly address memory constraints. Below, we formally analyze this trend and the impact of low-bitwidth language models on addressing memory bottlenecks during inference.

### G.1   OVERVIEW OF RECENT DATACENTER GPUS AND ACCELERATORS.

We begin our analysis by surveying a broad range of recent datacenter General Purpose GPUs (GPG-PUs) employed for neural network development and research since 2018. This includes hardware from multiple providers, covering various configurations across the latest microarchitectures.

From Nvidia, we consider the following:

- **Volta**: V100 (SXM/PCIe) (Nvidia Team, 2018),
- **Ampere**: A100 (40GB/80GB SXM/PCIe) (Nvidia Team, 2020),
- **Hopper**: H100 (SXM/PCIe) and H200 (Nvidia Team, 2022; 2023),
- **Blackwell**: This includes preliminary data for Blackwell microarchitectures, which at the time of access were subject to change (Nvidia Team, 2024).

From AMD, we analyze the following models:

- **MI200 Series**: MI210, MI250, MI250X (AMD Team, 2022a;b),
- **MI300 Series**: MI300A, MI300X, MI325X (AMD Team, 2023a;b; 2024).

Additionally, we include hardware from Intel and Google:

- From Intel, the **Gaudi Series**: Gaudi 2 and Gaudi 3 (Intel Gaudi Team, 2024),
- From Google, the **Tensor Processing Units (TPUs)**: TPUv3 (Google TPU Team, 2018), TPUv4 (Google TPU Team, 2021), and TPUv5 (TPUv5e, TPUv5p) (Google TPU Team, 2023a;b).

All data was sourced from the respective datasheets, technical documentation, or press releases of the cited hardware. Over the past several years, each of these four accelerator families has improved in three areas - FLOPS, memory capacity, and bandwidth.

### G.2   MEMORY TRENDS AND SPEEDUP OPPORTUNITIES IN LOW-BITWIDTH LANGUAGE MODELING

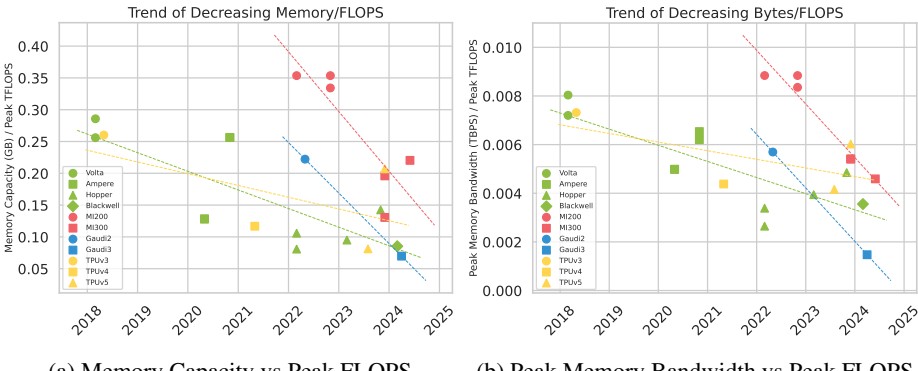

(a) Memory Capacity vs Peak FLOPS          (b) Peak Memory Bandwidth vs Peak FLOPS

Figure 24: Trends of Memory/FLOP and Bandwidth/FLOP across different (datacenter) GPGPUs.

**Memory Capacity and Bandwidth of GPGPUs Relative to Peak TFLOPs.** In Figure 24a, we show the trends of Memory Capacity over Peak TFLOPS (Half Precision - FP16/BF16) for various accelerators over the years. We also perform a linear fit for each family of accelerators separately. The linear fit for all the families has a downward slope, showing that memory capacity is improving at a slower pace than computation capability. This trend holds true even for the most recent hardware, such as Blackwell, MI325X, and Gaudi3. Though we consider Half-Precision TFLOPs, the slope is expected to become steeper when considering peak TFLOPS over Ampere sparse or FP8. Similarly, in Figure 24b, we present the trends of Memory Bandwidth (specifically for DRAM or its equivalent memory) over FLOPs for the accelerators over the years, along with the linear fit for each family. We observe a downward slope here as well, indicating the trend that memory bandwidth is growing much slower than computation.

**Comparison of model size and maximum speed up for quantized models:** In this analysis, we include transformer configurations from the LLaMa family (Touvron et al., 2023a;b). As larger vocabularies in LLMs are becoming increasingly common for efficient multilingual modeling, we use a vocabulary size of 128k from LLaMa 3 (AI@Meta, 2024) for our analysis. We assume the Embedding and LM Head weights are retained in Half-Precision across all bitwidths. In Figure 25, we provide the scaling plots of FloatLMs, QuantLMs and TriLMs in terms of model size (in GB) and maximum speedup compared to the FloatLM baseline.

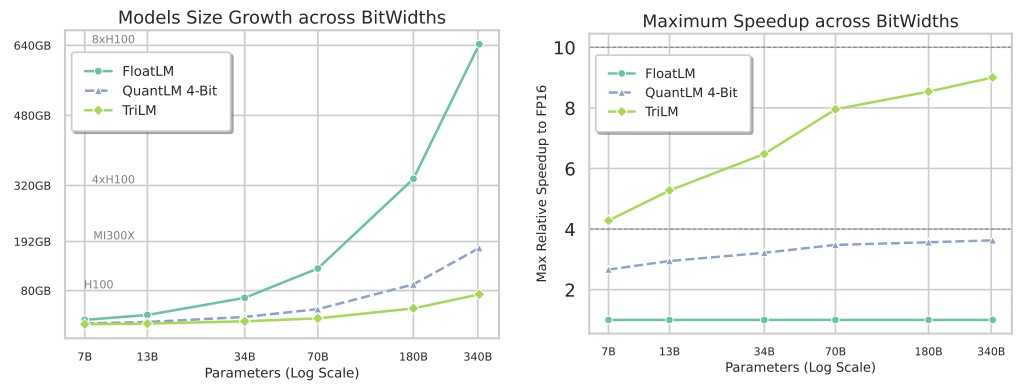

(a) Model Size growth at various BitWidths  (b) Maximum Possible Speedup at different BitWidths

Figure 25: Expected gains from low bitwidth modeling. TriLMs can fit over 300B parameters on a single H100 and achieve up to a theoretical maximum of 10x faster autoregressive decoding compared to FloatLM.

# H ABLATIONS

Table 14 and 13 shows the performance of ablation 100B token training runs over the six commonsense benchmarks from BitNet b1.58 at 1.1B parameters. The first two rows show the performance of TriLM 1.1B and Float 1.1B at this token count, followed by our replication of BitNet b1.58 (Ours) as well as the scores from BitNet b1.58 over 700M and 1.3B parameters. We observe that at this scale, TriLM does not come close to matching the performance of FloatLM, but it outperforms much larger BitNets. The next two rows show the performance of TriLM 1.1B and FloatLM 1.1B when trained on 100B tokens of FineWeb, instead of SlimPajama. While the performance of both the models improves on FineWeb, the average difference in their performance across datasets remains the same. Lastly, we show the performances across various optimization schedules. A significant drop in averaged performance is noticed when the baseline schedule of linear decay with constant weight decay is used. The gains from dropping l2 regularization in the schedule are more than that of dropping the peak learning rate, however, not enough to match that of TriLM 1.1B's schedule.

| Models | HellaSwag | | PIQA | | WinoGrande |
|---|---|---|---|---|---|
| | Acc N. | Acc | Acc N. | Acc | Acc |
| FloatLM 1.1B | $50.0 \pm 0.5$ | $39.3 \pm 0.5$ | $70.9 \pm 1.0$ | $70.1 \pm 1.0$ | $55.4 \pm 1.4$ |
| TriLM 1.1B | $46.8 \pm 0.5$ | $37.1 \pm 0.4$ | $69.4 \pm 1.0$ | $69.4 \pm 1.0$ | $53.8 \pm 1.4$ |
| BitNet b1.58 1.1B (Ours) | $47.0 \pm 0.5$ | $37.1 \pm 0.4$ | $69.4 \pm 1.0$ | $69.6 \pm 1.0$ | $53.4 \pm 1.4$ |
| BitNet b1.58 700M | 35.1 | | 68.1 | | 55.2 |
| BitNet b1.58 1.3B | 37.7 | | 68.8 | | 55.8 |
| TriLM 1.1B FineWeb | $50.0 \pm 0.5$ | $39.2 \pm 0.4$ | $70.2 \pm 1.0$ | $70.1 \pm 1.0$ | $56.6 \pm 1.3$ |
| FloatLM 1.1B FineWeb | $52.7 \pm 0.5$ | $41.1 \pm 0.4$ | $73.0 \pm 1.0$ | $71.3 \pm 1.0$ | $56.7 \pm 1.3$ |
| TriLM 1.1B Only Peak LR Dropped | $46.7 \pm 0.5$ | $36.8 \pm 0.4$ | $68.9 \pm 1.0$ | $69.5 \pm 1.0$ | $55.4 \pm 1.4$ |
| TriLM 1.1B Only L2 Reg. Dropped | $47.1 \pm 0.5$ | $37.5 \pm 0.4$ | $68.6 \pm 1.0$ | $69.4 \pm 1.0$ | $55.2 \pm 1.4$ |
| TriLM 1.1B Baseline Schedule | $46.0 \pm 0.5$ | $36.9 \pm 0.4$ | $69.3 \pm 1.0$ | $69.1 \pm 1.0$ | $56.2 \pm 1.3$ |

Table 13: Ablation Common Sense Task Performance: HellaSwag, PIQA, WinoGrande, Arc Easy, Arc Challenge, BoolQ (Contd.). BitNet b1.58's scores from Ma et al. (2024). All runs are for 100B tokens on Slim Pajama, except those explicitly stated as FineWeb

| Models | Arc Challenge | | Arc Easy | | BoolQ | Avg ( HellaSwag, PIQA, WinoGrande, |
|---|---|---|---|---|---|---|
| | Acc N. | Acc | Acc N. | Acc | Acc | Arc Easy, Arc Challenge, and BoolQ) |
| FloatLM 1.1B | $26.3 \pm 1.3$ | $22.5 \pm 1.2$ | $50.3 \pm 1.0$ | $56.8 \pm 1.0$ | $60.6 \pm 0.8$ | 52.2 |
| TriLM 1.1B | $26.7 \pm 1.3$ | $22.9 \pm 1.2$ | $49.7 \pm 1.0$ | $55.0 \pm 1.0$ | $54.9 \pm 0.8$ | 50.2 |
| BitNet b1.58 1.1B (Ours) | $26.1 \pm 1.2$ | $23.6 \pm 1.2$ | $47.7 \pm 1.0$ | $55.3 \pm 1.0$ | $49.7 \pm 0.8$ | 48.9 |
| BitNet b1.58 700M | 21.4 | | 51.8 | | 58.2 | 48.3 |
| BitNet b1.58 1.3B | 24.2 | | 54.9 | | 56.7 | 49.6 |
| TriLM 1.1B FineWeb | $31.7 \pm 1.3$ | $31.9 \pm 1.3$ | $63.1 \pm 0.9$ | $66.8 \pm 0.9$ | $58.3 \pm 0.8$ | 54.9 |
| FloatLM 1.1B FineWeb | $34.4 \pm 1.3$ | $33.0 \pm 1.3$ | $65.7 \pm 0.9$ | $70.2 \pm 0.9$ | $59.3 \pm 0.8$ | 56.9 |
| TriLM 1.1B Only Peak LR Dropped | $27.4 \pm 1.3$ | $23.6 \pm 1.2$ | $48.3 \pm 1.0$ | $55.1 \pm 1.0$ | $51.6 \pm 0.8$ | 49.7 |
| TriLM 1.1B Only L2 Reg. Dropped | $27.6 \pm 1.3$ | $24.8 \pm 1.2$ | $49.2 \pm 1.0$ | $55.1 \pm 1.0$ | $53.1 \pm 0.8$ | 50.1 |
| TriLM 1.1B Baseline Schedule | $26.2 \pm 1.2$ | $23.2 \pm 1.2$ | $48.0 \pm 1.0$ | $54.0 \pm 1.0$ | $49.4 \pm 0.8$ | 49.1 |

Table 14: Ablation Common Sense Task Performance: HellaSwag, PIQA, WinoGrande, Arc Easy, Arc Challenge, BoolQ. BitNet b1.58's scores from Ma et al. (2024). All runs are for 100B tokens on Slim Pajama, except those explicitly stated as FineWeb

| Models | TriviaQA | CrowsPairs | | Big Bench BBQ Lite | TruthfulQA |
| | Exact Match | Likelihood diff. | Pct stereotype | Acc | Acc |
|---|---|---|---|---|---|
| FloatLM 99M | 0.6± 0.1 | 372.4± 14.6 | 55.4± 1.2 | 30.8± 0.4 | 24.4± 1.5 |
| QuantLM 99M 8-Bit | 0.6± 0.1 | 370.9± 14.6 | 55.1± 1.2 | 26.5± 0.3 | 24.1± 1.5 |
| QuantLM 99M 6-Bit | 0.6± 0.1 | 389.8± 14.8 | 56.9± 1.2 | 26.7± 0.3 | 24.2± 1.5 |
| QuantLM 99M 4-Bit | 0.3± 0 | 425.5± 15.2 | 54.0± 1.2 | 26.2± 0.3 | 22.9± 1.5 |
| QuantLM 99M 3-Bit | 0.1± 0 | 611.1± 18.8 | 51.0± 1.2 | 31.4± 0.4 | 24.6± 1.5 |
| TriLM 99M | 0.1± 0 | 362.4± 10.8 | 54.2± 1.2 | 30.8± 0.4 | 24.2± 1.5 |
| Binary 99M | 0.2± 0.0 | 353.5± 11.1 | 53.3± 1.2 | 31.5± 0.3 | 25.7± 1.5 |
| FloatLM 190M | 0.6± 0.1 | 348.2± 11.3 | 55.9± 1.2 | 27.3± 0.4 | 22.4± 1.5 |
| QuantLM 190M 8-Bit | 0.7± 0.1 | 352.7± 11.4 | 56.2± 1.2 | 27.1± 0.4 | 22.5± 1.5 |
| QuantLM 190M 6-Bit | 0.7± 0.1 | 368.9± 11.7 | 56.2± 1.2 | 27.2± 0.4 | 22.5± 1.5 |
| QuantLM 190M 4-Bit | 0.0± 0 | 961.9± 25.4 | 43.8± 1.2 | 35.0± 0.4 | 24.2± 1.5 |
| QuantLM 190M 3-Bit | 0.1± 0 | 482.7± 15.2 | 53.7± 1.2 | 26.4± 0.3 | 25.0± 1.5 |
| TriLM 190M | 0.2± 0 | 343.5± 10.9 | 55.5± 1.2 | 29.7± 0.4 | 23.9± 1.5 |
| FloatLM 390M | 2.8± 0 | 355.5± 10.4 | 59.6± 1.2 | 25.4± 0.3 | 22.4± 1.5 |
| QuantLM 390M 8-Bit | 2.9± 0 | 355.8± 10.4 | 59.8± 1.2 | 25.4± 0.3 | 22.2± 1.5 |
| QuantLM 390M 6-Bit | 2.4± 0 | 360.5± 10.4 | 60.6± 1.2 | 25.3± 0.3 | 22.8± 1.5 |
| QuantLM 390M 4-Bit | 1.3± 0.1 | 368.2± 10.2 | 59.4± 1.2 | 25.5± 0.3 | 22.8± 1.5 |
| QuantLM 390M 3-Bit | 0.8± 0.1 | 444.4± 12.2 | 54.3± 1.2 | 26.3± 0.3 | 23.0± 1.5 |
| TriLM 390M | 1.3± 0.1 | 344.5± 10.3 | 58.3± 1.2 | 26.9± 0.3 | 24.4± 1.5 |
| FloatLM 560M | 4.6± 0.2 | 351.8± 9.9 | 58.9± 1.2 | 25.7± 0.3 | 21.7± 1.4 |
| QuantLM 560M 8-Bit | 4.7± 0.2 | 352.9± 10.0 | 59.2± 1.2 | 25.7± 0.3 | 21.8± 1.4 |
| QuantLM 560M 6-Bit | 3.5± 0.1 | 353.7± 9.9 | 59.3± 1.2 | 25.8± 0.3 | 22.0± 1.5 |
| QuantLM 560M 4-Bit | 2.1± 0.1 | 372.7± 10.7 | 59.2± 1.2 | 27.0± 0.4 | 22.2± 1.5 |
| QuantLM 560M 3-Bit | 1.5± 0.1 | 411.2± 11.3 | 57.9± 1.2 | 29.0± 0.4 | 22.9± 1.5 |
| TriLM 560M | 2.4± 0.1 | 345.1± 10.1 | 58.7± 1.2 | 25.5± 0.3 | 23.6± 1.5 |
| Binary 560M | 0.2± 0.0 | 356.3± 10.4 | 58.5± 1.2 | 26.3± 0.3 | 23.1± 1.4 |
| FloatLM 830M | 8.5± 0.2 | 354.6± 9.6 | 62.6± 1.2 | 25.7± 0.3 | 23.1± 1.5 |
| QuantLM 830M 8-Bit | 8.5± 0.2 | 354.5± 9.6 | 62.1± 1.2 | 25.6± 0.3 | 23.0± 1.5 |
| QuantLM 830M 6-Bit | 8.5± 0.2 | 354.6± 9.6 | 62.7± 1.2 | 25.5± 0.3 | 22.5± 1.5 |
| QuantLM 830M 4-Bit | 10.6± 0.2 | 364.2± 9.8 | 59.9± 1.2 | 25.9± 0.3 | 21.8± 1.4 |
| QuantLM 830M 3-Bit | 3.1± 0.1 | 389.5± 10.9 | 59.9± 1.2 | 30.5± 0.4 | 24.4± 1.5 |
| TriLM 830M | 4.3± 0.2 | 344.9± 10.0 | 60.7± 1.2 | 25.1± 0.3 | 22.8± 1.5 |
| FloatLM 1.1B | 12.9± 0.3 | 349.2± 9.7 | 61.2± 1.2 | 25.4± 0.3 | 21.4± 1.4 |
| QuantLM 1.1B 8-Bit | 12.7± 0.2 | 349.5± 9.7 | 61.1± 1.2 | 25.4± 0.3 | 21.7± 1.4 |
| QuantLM 1.1B 6-Bit | 12.4± 0.2 | 349.7± 9.6 | 59.9± 1.2 | 25.5± 0.3 | 21.9± 1.4 |
| QuantLM 1.1B 4-Bit | 9.3± 0.2 | 359.1± 10.1 | 60.9± 1.2 | 25.4± 0.3 | 21.3± 1.4 |
| QuantLM 1.1B 3-Bit | 6.8± 0.2 | 422.4± 11.5 | 58.7± 1.2 | 29.9± 0.4 | 24.2± 1.5 |
| TriLM 1.1B | 1.9± 0.1 | 343.4± 9.9 | 61.4± 1.2 | 25.8± 0.3 | 21.5± 1.4 |
| Binary 1.1B | 2.2± 0.1 | 351.6± 9.8 | 58.3± 1.2 | 26.4± 0.3 | 23.2± 1.4 |
| FloatLM 1.5B | 12.2± 0.2 | 351.9± 9.6 | 61.6± 1.2 | 26.8± 0.3 | 21.8± 1.4 |
| QuantLM 1.5B 8-Bit | 12.5± 0.2 | 352.4± 9.6 | 61.6± 1.2 | 26.8± 0.3 | 21.8± 1.4 |
| QuantLM 1.5B 6-Bit | 11.3± 0.2 | 350.9± 9.7 | 61.9± 1.2 | 27.1± 0.4 | 21.5± 1.4 |
| QuantLM 1.5B 4-Bit | 9.0± 0.2 | 357.9± 9.8 | 60.7± 1.2 | 25.9± 0.3 | 20.8± 1.4 |
| QuantLM 1.5B 3-Bit | 4.2± 0.1 | 400.0± 10.6 | 60.9± 1.2 | 26.8± 0.3 | 20.8± 1.4 |
| TriLM 1.5B | 5.9± 0.1 | 348.9± 9.9 | 59.9± 1.2 | 25.2± 0.3 | 21.7± 1.4 |
| FloatLM 2.4B | 20.7± 0.3 | 360.4± 9.4 | 64.2± 1.2 | 26.7± 0.3 | 21.7± 1.4 |
| QuantLM 2.4B 8-Bit | 20.7± 0.3 | 360.5± 9.4 | 64.2± 1.2 | 26.5± 0.3 | 21.9± 1.4 |
| QuantLM 2.4B 6-Bit | 20.4± 0.3 | 360.8± 9.5 | 63.4± 1.2 | 26.4± 0.3 | 21.8± 1.4 |
| QuantLM 2.4B 4-Bit | 21.1± 0.3 | 358.7± 9.6 | 63.4± 1.2 | 26.0± 0.3 | 21.7± 1.4 |
| QuantLM 2.4B 3-Bit | 10.9± 0.2 | 360.2± 9.5 | 59.9± 1.2 | 25.8± 0.3 | 21.5± 1.4 |
| TriLM 2.4B | 12.3± 0.1 | 353.0± 10.0 | 64.1± 1.2 | 25.4± 0.3 | 23.0± 1.5 |
| FloatLM 3.9B | 21.5± 0.3 | 359.2± 9.6 | 64.7± 1.2 | 25.4± 0.3 | 23.6± 1.5 |
| QuantLM 3.9B 8-Bit | 21.7± 0.3 | 359.8± 9.6 | 64.6± 1.2 | 25.4± 0.3 | 23.6± 1.5 |
| QuantLM 3.9B 6-Bit | 21.0± 0.3 | 359.5± 9.6 | 63.9± 1.2 | 25.4± 0.3 | 23.5± 1.5 |
| QuantLM 3.9B 4-Bit | 17.9± 0.3 | 365.5± 9.7 | 64.8± 1.2 | 25.3± 0.3 | 24.2± 1.5 |
| QuantLM 3.9B 3-Bit | 8.2± 0.2 | 365.9± 9.8 | 64.3± 1.2 | 25.5± 0.3 | 21.9± 1.4 |
| TriLM 3.9B | 21.3± 0.3 | 362.4± 9.6 | 65.4± 1.2 | 25.9± 0.3 | 24.1± 1.5 |

Table 15: Spectra Suite Performance (Part 4): TriviaQA, CrowsPairs, Big Bench BBQ Lite, TruthQA. We additionally also include Pythia's performance scores.

| Models | MMLU Accuracy | | | | |
| | Stem | Humanities | Social Sciences | Other | Avg. |
| --- | --- | --- | --- | --- | --- |
| FloatLM 99M | 22.8± 0.7 | 24.0± 0.6 | 27.0± 0.8 | 28.0± 0.8 | 25.3± 0.4 |
| QuantLM 99M 8-Bit | 22.9± 0.7 | 24.2± 0.6 | 26.9± 0.8 | 27.9± 0.8 | 25.3± 0.4 |
| QuantLM 99M 6-Bit | 22.7± 0.7 | 24.1± 0.6 | 26.6± 0.8 | 28.2± 0.8 | 25.2± 0.4 |
| QuantLM 99M 4-Bit | 22.9± 0.7 | 24.1± 0.6 | 26.7± 0.8 | 27.4± 0.8 | 25.1± 0.4 |
| QuantLM 99M 3-Bit | 23.5± 0.8 | 23.9± 0.6 | 26.2± 0.8 | 25.9± 0.8 | 24.8± 0.4 |
| TriLM 99M | 23.9± 0.8 | 23.6± 0.6 | 26.7± 0.8 | 26.6± 0.8 | 25.0± 0.4 |
| Binary 99M | 21.6± 0.7 | 24.3± 0.6 | 21.8± 0.7 | 24.0± 0.7 | 23.1± 0.3 |
| FloatLM 190M | 24.0± 0.8 | 24.4± 0.6 | 28.8± 0.8 | 30.1± 0.8 | 26.5± 0.4 |
| QuantLM 190M 8-Bit | 24.1± 0.8 | 24.5± 0.6 | 28.9± 0.8 | 30.0± 0.8 | 26.6± 0.4 |
| QuantLM 190M 6-Bit | 24.1± 0.8 | 24.5± 0.6 | 28.3± 0.8 | 29.8± 0.8 | 26.4± 0.4 |
| QuantLM 190M 4-Bit | 22.9± 0.7 | 22.9± 0.6 | 24.5± 0.8 | 23.4± 0.8 | 23.4± 0.4 |
| QuantLM 190M 3-Bit | 23.9± 0.8 | 23.2± 0.6 | 25.4± 0.8 | 27.5± 0.8 | 24.8± 0.4 |
| TriLM 190M | 22.5± 0.7 | 23.8± 0.6 | 26.7± 0.8 | 28.4± 0.8 | 25.2± 0.4 |
| FloatLM 390M | 25.8± 0.8 | 25.9± 0.6 | 30.3± 0.8 | 32.8± 0.8 | 28.3± 0.4 |
| QuantLM 390M 8-Bit | 25.7± 0.8 | 25.9± 0.6 | 30.2± 0.8 | 32.4± 0.8 | 28.2± 0.4 |
| QuantLM 390M 6-Bit | 26.0± 0.8 | 25.8± 0.6 | 30.2± 0.8 | 32.3± 0.8 | 28.3± 0.4 |
| QuantLM 390M 4-Bit | 25.5± 0.8 | 25.4± 0.6 | 30.5± 0.8 | 31.6± 0.8 | 27.9± 0.4 |
| QuantLM 390M 3-Bit | 24.4± 0.8 | 25.0± 0.6 | 29.4± 0.8 | 29.3± 0.8 | 26.8± 0.4 |
| TriLM 390M | 24.1± 0.8 | 24.8± 0.6 | 28.3± 0.8 | 29.0± 0.8 | 26.4± 0.4 |
| FloatLM 560M | 24.8± 0.8 | 26.7± 0.6 | 30.5± 0.8 | 32.3± 0.8 | 28.4± 0.4 |
| QuantLM 560M 8-Bit | 24.8± 0.8 | 26.6± 0.6 | 30.5± 0.8 | 32.1± 0.8 | 28.3± 0.4 |
| QuantLM 560M 6-Bit | 24.6± 0.8 | 26.7± 0.6 | 30.5± 0.8 | 31.3± 0.8 | 28.1± 0.4 |
| QuantLM 560M 4-Bit | 24.7± 0.8 | 25.9± 0.6 | 29.9± 0.8 | 31.1± 0.8 | 27.7± 0.4 |
| QuantLM 560M 3-Bit | 24.5± 0.8 | 24.2± 0.6 | 28.1± 0.8 | 28.2± 0.8 | 26.0± 0.4 |
| TriLM 560M | 25.0± 0.8 | 25.1± 0.6 | 29.0± 0.8 | 30.2± 0.8 | 27.0± 0.4 |
| Binary 560M | 21.4± 0.7 | 24.2± 0.6 | 21.6± 0.7 | 23.9± 0.7 | 22.9± 0.3 |
| FloatLM 830M | 25.8± 0.8 | 27.5± 0.6 | 32.3± 0.8 | 34.6± 0.8 | 29.7± 0.4 |
| QuantLM 830M 8-Bit | 25.8± 0.8 | 27.4± 0.6 | 32.1± 0.8 | 34.7± 0.8 | 29.7± 0.4 |
| QuantLM 830M 6-Bit | 25.6± 0.8 | 27.3± 0.6 | 32.1± 0.8 | 34.2± 0.8 | 29.5± 0.4 |
| QuantLM 830M 4-Bit | 25.9± 0.8 | 26.8± 0.6 | 31.2± 0.8 | 33.6± 0.8 | 29.1± 0.4 |
| QuantLM 830M 3-Bit | 24.8± 0.8 | 25.1± 0.6 | 28.9± 0.8 | 30.8± 0.8 | 27.1± 0.4 |
| TriLM 830M | 24.9± 0.8 | 25.8± 0.6 | 30.1± 0.8 | 31.1± 0.8 | 27.7± 0.4 |
| FloatLM 1.1B | 26.4± 0.8 | 27.6± 0.6 | 32.5± 0.8 | 34.8± 0.8 | 30.0± 0.4 |
| QuantLM 1.1B 8-Bit | 26.2± 0.8 | 27.4± 0.6 | 32.5± 0.8 | 34.9± 0.8 | 29.9± 0.4 |
| QuantLM 1.1B 6-Bit | 26.0± 0.8 | 27.5± 0.6 | 32.7± 0.8 | 34.9± 0.8 | 29.9± 0.4 |
| QuantLM 1.1B 4-Bit | 26.0± 0.8 | 26.6± 0.6 | 32.4± 0.8 | 33.8± 0.8 | 29.3± 0.4 |
| QuantLM 1.1B 3-Bit | 25.9± 0.8 | 26.1± 0.6 | 30.0± 0.8 | 33.0± 0.8 | 28.4± 0.4 |
| TriLM 1.1B | 25.2± 0.8 | 26.1± 0.6 | 30.6± 0.8 | 32.2± 0.8 | 28.3± 0.4 |
| Binary 1.1B | 21.0± 0.7 | 24.2± 0.6 | 21.7± 0.8 | 24.4± 0.7 | 23.0± 0.3 |
| FloatLM 1.5B | 26.1± 0.8 | 28.0± 0.7 | 33.0± 0.8 | 35.6± 0.8 | 30.4± 0.4 |
| QuantLM 1.5B 8-Bit | 26.1± 0.8 | 28.1± 0.7 | 32.9± 0.8 | 35.5± 0.8 | 30.3± 0.4 |
| QuantLM 1.5B 6-Bit | 26.3± 0.8 | 28.0± 0.7 | 33.0± 0.8 | 35.4± 0.8 | 30.4± 0.4 |
| QuantLM 1.5B 4-Bit | 26.2± 0.8 | 28.1± 0.7 | 32.4± 0.8 | 34.8± 0.8 | 30.1± 0.4 |
| QuantLM 1.5B 3-Bit | 25.5± 0.8 | 26.7± 0.6 | 31.2± 0.8 | 33.4± 0.8 | 28.9± 0.4 |
| TriLM 1.5B | 25.7± 0.8 | 27.4± 0.6 | 31.5± 0.8 | 34.6± 0.8 | 29.5± 0.4 |
| FloatLM 2.4B | 26.9± 0.8 | 29.4± 0.7 | 34.2± 0.8 | 38.1± 0.9 | 31.8± 0.4 |
| QuantLM 2.4B 8-Bit | 27.0± 0.8 | 29.4± 0.7 | 34.1± 0.8 | 38.0± 0.9 | 31.8± 0.4 |
| QuantLM 2.4B 6-Bit | 26.8± 0.8 | 29.5± 0.7 | 34.2± 0.8 | 38.2± 0.9 | 31.8± 0.4 |
| QuantLM 2.4B 4-Bit | 26.5± 0.8 | 28.8± 0.7 | 34.3± 0.8 | 38.1± 0.9 | 31.5± 0.4 |
| QuantLM 2.4B 3-Bit | 25.5± 0.8 | 27.1± 0.6 | 32.3± 0.8 | 36.4± 0.9 | 29.9± 0.4 |
| TriLM 2.4B | 27.4± 0.8 | 27.8± 0.6 | 34.6± 0.9 | 35.1± 0.8 | 30.8± 0.4 |
| FloatLM 3.9B | 27.7± 0.8 | 30.6± 0.7 | 36.9± 0.9 | 39.8± 0.9 | 33.3± 0.4 |
| QuantLM 3.9B 8-Bit | 27.6± 0.8 | 30.7± 0.7 | 37.0± 0.9 | 39.7± 0.9 | 33.4± 0.4 |
| QuantLM 3.9B 6-Bit | 27.3± 0.8 | 30.3± 0.7 | 36.9± 0.9 | 39.3± 0.9 | 33.1± 0.4 |
| QuantLM 3.9B 4-Bit | 27.1± 0.8 | 30.3± 0.7 | 36.3± 0.9 | 38.8± 0.9 | 32.8± 0.4 |
| QuantLM 3.9B 3-Bit | 27.3± 0.8 | 28.4± 0.7 | 34.3± 0.9 | 37.2± 0.9 | 31.4± 0.4 |
| TriLM 3.9B | 28.3± 0.8 | 29.5± 0.7 | 35.4± 0.9 | 39.6± 0.9 | 32.8± 0.4 |

Table 16: Spectra Suite Performance (Part 5): MMLU- STEM, Humanities, Social Sciences, Others.

## I  ILLUSTRATIVE EXAMPLES OF TRILM 3.9B'S COMPLETION CAPABILITIES

We showcase instances of outputs produced by TriLM (3.9B) across diverse tasks, highlighting its proficiency in tasks such as comprehension, prompt completion, and creative composition.

---

**Generated Output on Reading Comprehension by TriLM (3.9B)**

**Title: The Blitz Background:** From the German point of view, March 1941 saw an improvement. The Luftwaffe flew 4,000 sorties that month, including 12 major and three heavy attacks. The electronic war intensified but the Luftwaffe flew major inland missions only on moonlit nights. Ports were easier to find and made better targets. To confuse the British, radio silence was observed until the bombs fell. X- and Y-Gerät beams were placed over false targets and switched only at the last minute. Rapid frequency changes were introduced for X-Gerät, whose wider band of frequencies and greater tactical flexibility ensured it remained effective at a time when British selective jamming was degrading the effectiveness of Y-Gerät.

**Q: How many sorties were flown in March 1941?**
**A:** 4,000

**Q: When did the Luftwaffe fly inland missions?**
**A:** Only during moonlit nights

- - - - - - - - - - - - - - - - - - - - - - - - - - - - - - - - - - - - - - - - - - - - - -

**Title: Helsinki**
**Background:** Helsinki is the capital and largest city of Finland. It is in the region of Uusimaa, in southern Finland, on the shore of the Gulf of Finland. Helsinki has a population of , an urban population of , and a metropolitan population of over 1.4 million, making it the most populous municipality and urban area in Finland. Helsinki is some north of Tallinn, Estonia, east of Stockholm, Sweden, and west of Saint Petersburg, Russia. Helsinki has close historical connections with these three cities. The Helsinki metropolitan area includes the urban core of Helsinki, Espoo, Vantaa, Kauniainen, and surrounding commuter towns. It is the world's northernmost metro area of over one million people, and the city is the northernmost capital of an EU member state. The Helsinki metropolitan area is the third largest metropolitan area in the Nordic countries after Stockholm and Copenhagen, and the City of Helsinki is the third largest after Stockholm and Oslo. Helsinki is Finland's major political, educational, financial, cultural, and research center as well as one of northern Europe's major cities. Approximately 75% of foreign companies that operate in Finland have settled in the Helsinki region. The nearby municipality of Vantaa is the location of Helsinki Airport, with frequent service to various destinations in Europe and Asia.

**Q: What is the most populous municipality in Finland?**
**A:** Helsinki

**Q: How many people live there?**
**A:** 1.4 million in the metropolitan area

**Q: What percent of the foreign companies that operate in Finland are in Helsinki?**
**A:** 75%

**Q: What towns are a part of the metropolitan area?**
**A:** Espoo, Vantaa, Kauniainen, and surrounding commuter towns

---

**Auto-complete on a creative writing task by TriLM (3.9B)**

**Essay Writing**

In the year 2150, humanity had established colonies on Mars. Captain Ramirez was on a routine mission to explore an uncharted region of the planet when he and his crew were attacked by Martian natives who wanted them dead for their technology. The captain's ship crashed into the surface of Mars where it exploded in flames.
The survivors managed to escape from the wreckage but they soon found themselves stranded with no way out or back home. They would have to survive alone until rescue came. But what if there is nothing waiting for them? What if they are trapped forever?

---

**Poem Writing**

In a garden where the roses bloom,
Underneath the silver moon,
A gentle breeze begins to sway,
And whispers secrets of the day.
The stars above start to appear,
While shadows dance and disappear,
The night unfolds in its own time.
The sun rises over the horizon,
As it does every morning,
But this is no ordinary dawn.
It's a new beginning for all who are here today.
For those who have come before us,
And those yet to be born.
We celebrate life with music and laughter,
With friends old and new.
This is our moment to shine!

---

**Auto-complete on various prompts by TriLM (3.9B)**

Making a cake: Several cake pops are shown on a display. A woman and girl are shown making the cake pops in a kitchen. They use a mixer to mix ingredients, then knead dough by hand. The women roll out the dough with a rolling pin, cut it into shapes using cookie cutters, bake them, and decorate them with frosting or sprinkles.

The energy given up by electrons as they move through the electron transport chain is used to produce ATP. In this process, oxygen and water are produced in a reaction called cellular respiration.

When talking about sensitive topics, it is crucial to be aware of the fact that there are many different types of people. Some will react negatively and others positively. It's important not to take offense when someone says something you don't like.

If someone disagrees with you, it is best to ignore them. The most important thing in life is not what we have but who we are and how we treat others.

## J  BROADER IMPACT

**Interpretability Beyond Neuron Level:**    While several efforts have been made to understand how language models work and the means to steer them without training, these methods have mostly focussed on intervening at the neuron level. TriLMs open a new degree of interpretability - at the connection level. Here, the connections between any two neurons in a layer are in one of the three states - 0 (no connection), -1 (negative connection) and +1 (positive connection), each with equal strength. This is in sharp contrast to FloatLMs, where these connections can be of varying strengths, making it harder to study interpretability beyond the neuron level. By releasing the checkpoints across our training runs, we facilitate research along these directions.

**Environmental Benefits and Resource Efficiency:**    The open release of our models mitigates future emissions by allowing others to bypass the need for pretraining models from scratch. Moreover, TriLMs need much fewer resources during deployment and can perform the autoregressive generation at a faster pace - making them critical to scenarios demanding strict latency. Additionally, TriLMs represent a substantial advancement in enhancing performance on resource-constrained edge devices, including smartphones, laptops, and automobiles.

**Impact on Specialised Hardware:**    While TriLMs offer significant memory reduction and latency improvements on General Purpose GPUs like H100 and RTX4090, certain specialized hardware benefits more from ternary modeling. Hardware (like Cerabras[3]) that supports high byte-to-flop ratio computations, can leverage the sparsity stemming from ternarization for speedup in both training as well as inference. On the other hand, hardware with limited Memory/SRAM (like Groq[4]), benefits from the reduction in the number of chips needed to deploy an LLM.

**Reduced Training Costs:**    The Chinchilla scaling laws established that for training compute optimality, it may be recommended to train larger LLMs for fewer tokens than smaller LLMs for more tokens to achieve the desired model performance. However, memory requirements and latency associated with deployment of larger models, have motivated costlier training runs that go far beyond Chinchilla optimality. For example, a LLaMa 3 model with only 8B parameters was trained for 15T tokens. Since TriLM and ternary models, in general, can reduce the memory requirements and latency, this can motivate a shift inparameter-token tradeoff for efficient training runs towards Chinchilla's compute-optimal regime.

**Advancing Research through Open Access:**    The open suite of TriLM, FloatLM, and QuantLM families aims to empower researchers to explore the nuanced impacts of precision levels on model performance and efficiency, thereby catalyzing ongoing advancements in the development and deployment of language models, as well as enhancing their interpretability and safety Li (2024). By providing a range of publicly accessible models trained on openly available data, the suite offers unprecedented transparency in the training process. Intermediate checkpoints are available for all models, accompanied by detailed documentation of training procedures and hyperparameters.

---

[3]https://www.cerebras.net/product-chip/
[4]https://groq.com/

