# OpenReview forum: "Surprising Effectiveness of pretraining Ternary  Language Model at Scale"
_ICLR.cc/2025/Conference — ICLR 2025 Spotlight_

### Official Review · Reviewer_DqjN · 2024-10-30

**Soundness:** 3
**Presentation:** 2
**Contribution:** 2
**Rating:** 5
**Confidence:** 5

**Summary:**

This paper investigates the scaling law of low bit-width models, specifically ternary language models (TriLMs). The authors present Spectra LLM, an open suite for the quantization-aware training of LLMs. They conduct detailed analysis about TriLM and FloatLM in terms of reasoning, knowledge and toxicity. 3.9B parameter TriLM matches the performance of the FloatLM 3.9B across various benchmarks.

**Strengths:**

1) The authors provide detailed results of LLMs in various sizes and bits.
2) The authors conduct extensive evaluations in terms of reasoning, knowledge and toxicity.

**Weaknesses:**

The novelty of the proposed work is unclear. The architecture and training approach of TriLM appear quite similar to BitNet b1.58 [1], including methods such as two-stage weight decay and learning rate scheduling. Previous work [1] has already shown that a 3B ternary LLM can match half-precision LLMs with similar parameter counts and training costs. What additional contributions does this paper offer beyond BitNet b1.58?

[1] Ma, S., Wang, H., Ma, L., Wang, L., Wang, W., Huang, S., ... & Wei, F. (2024). The era of 1-bit llms: All large language models are in 1.58 bits.

**Questions:**

See Weaknesses

---

> ### Author Response · Authors · 2024-11-19
> **Response to Reviewer DqjN**
>
> We respectfully disagree with the reviewer and would like to clarify several important points that emphasize the novelty and key contributions of our research.
>
> **Systematic Study on Pretraining Ternary Language Models:**
> Firstly, the novelty of our paper does not lie in the architecture or training approach, as these are well-established in the literature [1], [2]. To the best of our knowledge, we are the first to systematically study and demonstrate the feasibility of pretraining ternary language models compared to their floating-point counterparts. Additionally, we establish their scaling laws and compare them to those of FloatLMs in a controlled setting. Moreover, we address additional questions related to scaling with respect to model size and compare them with competitive post-training quantization benchmarks to provide practical trade-off between performance and inference efficiency.
>
> **Fundamentally different conclusions:**
> Additionally, the conclusions drawn in BitNet’s [3] work differ from ours. While BitNet claims that a 3B BitNet model (perplexity: 9.91) outperforms a 3B Llama model (perplexity: 10.04), these conclusions are based on poor baselines, limited evaluation and without any systematic scaling studies. In contrast, our findings show that our TriLM models do not outperform their full-precision counterparts in terms of perplexity. Furthermore, the scaling equation derived for both TriLMs and FloatLMs suggest that, at larger scales, TriLMs will likely achieve performance comparable to FloatLMs, though they will never fully match them. We believe our conclusions are more robust and reflective of real-world performance, as they are derived from comprehensive evaluation, rigorous baselines, and systematic scaling studies.
>
>
> **Platform for Research through Open Reproducible Models:**
> Furthermore, we believe that the Spectra LLM suite makes a valuable contribution (see appendix I) to the LLM research community by facilitating comparative studies and exploring the scalability and efficiency of ternary modeling. Additionally, we release intermediate checkpoints, which are crucial for advancing research on learning dynamics, model safety, and interpretability.
>
> **TriLM vs Bitnet:**
> Lastly, we refer the reviewer to Appendix A.7, which provides additional details comparing our TriLM model with BitNet B1.58 architecturally. This comparison demonstrates that, for the same number of parameters and training tokens, TriLM consistently outperforms BitNet, further highlighting the strengths of our approach.
> In conclusion, our paper makes novel contributions to the field by providing a comprehensive, rigorous study of ternary language models and their scaling laws with proper benchmarking, offering insights that were not addressed in the BitNet work. We believe that our findings significantly expand the understanding of low-bitwidth models, particularly in terms of their potential for large-scale pretraining.
>
> [1] Hubara, I., Courbariaux, M., Soudry, D., El-Yaniv, R., & Bengio, Y. (2016). Binarized Neural Networks: Training Deep Neural Networks with Weights and Activations Constrained to +1 or -1. In Advances in Neural Information Processing Systems 29 (NIPS 2016).
> [2] Liu, Z., Oguz, B., Pappu, A., Shi, Y., & Krishnamoorthi, R. (2023). Binary and Ternary Natural Language Generation. In Proceedings of the 61st Annual Meeting of the Association for Computational Linguistics (Volume 1: Long Papers) (pp. 65-77). Toronto, Canada: Association for Computational Linguistics.
> [3] Ma, S., Wang, H., Ma, L., Wang, L., Wang, W., Huang, S., ... & Wei, F. (2024). The era of 1-bit llms: All large language models are in 1.58 bits.

---

> > ### Comment · Reviewer_DqjN · 2024-11-20
> >
> > Thanks for the authors’ detailed response. However, I still have the following concerns:
> >
> > 1)	The claim that “Spectra is the first to demonstrate the feasibility of pretraining ternary language models compared to their floating-point counterparts” is clearly over-estimated.  BitNet b1.58 already shows this. Furthermore, the training method of Spectra is quite like BitNet b1.58, especially two-stage weight decay scheduling. As for architecture, Spectra only replaces normalization per projection with normalization per layer, which is a very minor difference, and the authors do not explain clear motivations.
> >
> > 2)	The scalability of training data is also very important for pre-trained models. However, the paper does not conduct studies on this problem.
> >
> > 3)	Lack of details about comparison with BitNet b1.58. The authors do not provide experimental details in Appendix A.7, e.g., training data, hyper-parameters. Furthermore, the authors do not report on the validation perplexity of the two models on public dataset, e.g., wikitext or C4.

---

> > > ### Author Response · Authors · 2024-11-20
> > > **Response to Reviewer DqjN**
> > >
> > > We thank the reviewer for their prompt reply and for actively engaging with us during the rebuttal period.
> > >
> > > > The claim that “Spectra is the first to demonstrate the feasibility of pretraining ternary language models compared to their floating-point counterparts” is clearly over-estimated. BitNet b1.58 already shows this. Furthermore, the training method of Spectra is quite like BitNet b1.58, especially two-stage weight decay scheduling. As for architecture, Spectra only replaces normalization per projection with normalization per layer, which is a very minor difference, and the authors do not explain clear motivations.
> > >
> > > We acknowledge that the architecture and training method of our work is similar to BitNet 1.58B, which is appropriately cited in our work; however, this was neither our core contribution nor stated as such. We also believe the statement “we are the first to **systematically study** and demonstrate the feasibility of pretraining ternary language models compared to their floating-point counterparts” is valid. In Section 2.2, we discuss the theoretical feasibility for the first time, and to systematically understand and validate its scalability, we empirically analyze the scaling laws of both models in Section 4.3 and Appendix C by fitting nine models. The results of the scaling laws show that it is possible to train a ternary model comparable to its floating-point counterparts at scale. Furthermore, TriLM uses normalization per layer because it empirically provides better or similar results (See Table 10 and 11) compared to normalization per projection. This also simplifies the architecture by using fewer number of normalization steps.
> > >
> > > > The scalability of training data is also very important for pre-trained models. However, the paper does not conduct studies on this problem.
> > >
> > > We appreciate the reviewer’s insightful comment regarding the scalability of training data for pre-trained models. While most of our experiments are performed with 300B tokens, which already places them in the overtrained regime according to the Chinchilla Scaling Law, we provide additional experiments across varying training data below.
> > >
> > > | Training Tokens | Perplexity of TriLM 99M | Perplexity of TriLM 560M  |
> > > |------------------|------------------------------| ------------------------------|
> > > | 50B                  | 27.11                                | 15.48
> > > | 150B                | 26.30                                | 14.50
> > > | 300B                | 25.70                                | 14.01
> > >
> > > Table 1: Perplexity of TriLM on the validation set for different training token sizes.
> > >
> > > The above results show that the TriLMs improve with an increase in token size, demonstrating the scalability of the model with training data.
> > >
> > > > Lack of details about comparison with BitNet b1.58. The authors do not provide experimental details in Appendix A.7, e.g., training data, hyper-parameters. Furthermore, the authors do not report on the validation perplexity of the two models on public dataset, e.g., wikitext or C4.
> > >
> > > All the models used for comparison with BitNet 1.58B in Appendix A-7 were trained with 100B tokens from FineWeb (see Appendix G and line 2006); we will clarify this further in line 1254. We also provide the reported benchmarks from their paper for comparison with our replication. We used [this](https://openreview.net/forum?id=TJo6aQb7mK&noteId=vx1zBD0sC5) strategy for hyperparameter selection for both models.
> > >
> > > Additionally, we would like to reiterate ICLR’s policy on concurrent or contemporaneous work:
> > >
> > > *“We consider papers contemporaneous if they are published within the last four months. That means, since our full paper deadline is October 1, if a paper was published (i.e., at a peer-reviewed venue) on or after July 1, 2024, authors are not required to compare their own work to that paper. Authors are encouraged to cite and discuss all relevant papers, but they may be excused for not knowing about papers not published in peer-reviewed conference proceedings or journals, which includes papers exclusively available on arXiv.”*.
> > >
> > > We kindly request the reviewer to note that BitNet b1.58 has not undergone peer review nor has it been published in any conference proceedings or journals. It was only posted on arXiv earlier this year. However, we recognize its similarity to our work and, therefore, cite and compare it to the best of our ability, acknowledging their valuable contribution to the community.

---

> > > > ### Author Response · Authors · 2024-11-28
> > > > **Response to Reviewer DqjN**
> > > >
> > > > As we approach the conclusion of the rebuttal phase, we kindly request an acknowledgment of our response and paper revision. We would be happy to address any remaining major concerns the reviewer may have during the remaining time.
> > > >
> > > > We have made the following changes to address the reviewer’s concerns.
> > > > - New sections have been added to Appendix A.10: “Perplexity for Increasing Training Tokens in TriLM Models.”
> > > >
> > > > Given the reviewer’s initial negative assessment, it would be valuable to know if they still have any major concerns after our rebuttal response. This would provide us with an opportunity to address them and potentially improve the reviewer’s score.

---

### Official Review · Reviewer_AzmB · 2024-11-01

**Soundness:** 2
**Presentation:** 2
**Contribution:** 3
**Rating:** 5
**Confidence:** 4

**Summary:**

The author developed a series of LLMs with parameters ranging from 99M to 3.9B. These models include a 16-bit floating-point model and a ternary model, both trained from scratch using 300B tokens. Additionally, the author created 3, 4, 6, and 8-bit PTQ variants by applying GPTQ to quantize the floating-point model. Detailed evaluations and comparisons were conducted for each model. The results indicate that the ternary variant outperforms others at the same bit size, and the 3.9B parameter ternary model performs comparably to the 3.9B floating-point model on several evaluation datasets.

**Strengths:**

The author released a family of a fp and quantized model variant, as well as all the train details/loss and training techniques. Further more, the paper offers detailed evaluations results. Together, it provide good reference for the commnity for lower bit models.

**Weaknesses:**

The paper extensively discusses the theoretical or maximum possible speedup of the Ternary model, but it lacks results on actual inference speed. It would be beneficial to test the model in a real kernel environment, as many factors could influence speedup, such as activation quantization, KV cache, and kernel implementations.

*FP vs Ternary*: Both models were trained with 300B tokens, and the training loss does not appear to have fully converged yet. It remains to be seen how the models compare once both have converged, and the FP model may require more tokens due to its larger model capacity.

*Ternary vs Quant 3/4-bit Variant*: The comparison seems unfair, as the Ternary model is trained from scratch with QAT, while the Quant models are derived from GPTQ. The conclusion in the paper may not hold when using QAT for 3/4 bit variant.

The novelty is also limited. All components, such as the Ternary network, QAT with STE training, GPTQ, and model architecture, are well-known. The released pretrained models do not have much practical use, as there is a significant performance gap compared to state-of-the-art models of the same size.

**Questions:**

1. Can the author provide the actual inference speedup numbers for the quantized model compared to the FP model?

2. Train the FP models with more tokens and compare them with the Ternary models.

3. For the PTQ quantized model comparison versus QAT, can the author compare with QAT variant or use more recent PTQ methods such as SpinQuant/QuaRot?

4. What is the training cost of full QAT training compared to the FP baseline? The paper mentions model parallelism at scale. Why is it even necessary, given that the largest model is just 3.9B parameters?

---

> ### Author Response · Authors · 2024-11-19
> **Response to Reviewer AzmB (1/2)**
>
> We appreciate the reviewer's constructive criticism as it will contribute to enhancing the quality of our work.
>
> ### Weaknesses:
>
> > The paper extensively discusses the theoretical or maximum possible speedup of the Ternary model, but it lacks results on actual inference speed. It would be beneficial to test the model in a real kernel environment, as many factors could influence speedup, such as activation quantization, KV cache, and kernel implementations.
>
> Thank you for your valuable feedback. We acknowledge the importance of testing the Ternary model in a real kernel environment. While our paper primarily discusses the theoretical maximum speedup as a motivation, we are actively working on kernel-level optimizations. To demonstrate the potential speedup, we have benchmarked the models using open-source CPU kernels from llama.cpp, as mentioned in the [general comments](https://openreview.net/forum?id=TJo6aQb7mK&noteId=9Lym0unZ9V).
>
> We will include these results in the updated version of the paper. Furthermore, we will address GPU kernel optimizations in the “Future Work” section, as further improvements in GPU kernel implementation remain an ongoing area of research and development.
>
>
>
> >FP vs Ternary: Both models were trained with 300B tokens, and the training loss does not appear to have fully converged yet. It remains to be seen how the models compare once both have converged, and the FP model may require more tokens due to its larger model capacity.
>
> All the models considered in the work were trained on a fixed data regime of 300B tokens. According to Chinchilla Scaling law [1], the compute optimal number of tokens for our largest model with 3.9B parameters is 78B. This shows that all our models including the largest ones are in the overtrained regime. While further convergence is possible with more numbers of tokens, this does not diminish the current findings of the paper. Moreover, as demonstrated by the LLaMA series [2], achieving full convergence is unattainable even with training scales exceeding 15 trillion tokens. This underscores an inherent limitation in reaching complete convergence regardless of token scale, making it impractical to validate our research findings under such conditions. .
>
> > Ternary vs Quant 3/4-bit Variant: The comparison seems unfair, as the Ternary model is trained from scratch with QAT, while the Quant models are derived from GPTQ. The conclusion in the paper may not hold when using QAT for 3/4 bit variant.
>
> As outlined in Section 2.3, we only scale the ternary model with QAT because models quantized to 2, 3, or 4 bits involve more expensive arithmetic operations, such as full multiplications. These are computationally more demanding than the operations required for binary and ternary models. Although comparing the scaling and downstream performance of our ternary model with the floating point one is sufficient, we provide post-training quantized models to establish a practical trade-off between performance and inference efficiency during deployment. Previous findings [3] have shown that models quantized below 4 bits experience significant degradation in performance. This is why we only present post-training quantization (PQT) findings up to 3 bits, which already lead to notable performance degradation in comparison to their floating point counterpart.
>
> [1] Hoffmann, J. et. (2022). Training Compute-Optimal Large Language Models. arXiv preprint arXiv:2203.15556.
> [2] Meta AI. (2024). The Llama 3 Herd of Models. arXiv preprint arXiv:2407.217832.
> [3] Dettmers and Zettlemoyer, 2023. The case for 4-bit precision: k-bit Inference Scaling Laws.

---

> ### Author Response · Authors · 2024-11-19
> **Response to Reviewer AzmB (2/2)**
>
> > The novelty is also limited. All components, such as the Ternary network, QAT with STE training, GPTQ, and model architecture, are well-known. The released pretrained models do not have much practical use, as there is a significant performance gap compared to state-of-the-art models of the same size.
>
> We politely disagree with the reviewer. The novelty of this work is not in the architecture or the training approach, which are well-known in the scientific community. We are the first to systematically demonstrate the feasibility of pretraining ternary language models that are competitive with floating-point ones. We also pretrain models at various sizes to establish the scaling law for ternary language models and show that they approach their floating-point counterparts at scale. Additionally, we provide a suite of models and benchmark them extensively to validate our findings. Moreover, making all our intermediate checkpoints public at various training stages will not only assist researchers in conducting experiments on low-bit language models but also encourage them to better understand their training dynamics (see Appendix I/Broader Impact). Despite not achieving state-of-the-art results, language model suites like Pythia [1], and OLMo [2]  have been extremely impactful to the LLM research community. Furthermore, to the best of our knowledge, our TriLMs are state-of-the-art in comparison to existing ternary language models like BitNet 1.58Bit, as shown in Appendix A.7.
>
> ## Questions:
> > Can the author provide the actual inference speedup numbers for the quantized model compared to the FP model?
>
>  In response to the reviewer’s comment, we have included the inference speedup numbers using kernels from llama.cpp in the general comment to all the reviewers. We demonstrate the performance gains of TriLM models over both post-training quantized and floating-point models. We’ll add this in the Appendix of the final version.
>
> > Train the FP models with more tokens and compare them with the Ternary models.
>
> In response to the reviewer’s suggestion, we would like to reiterate that training all the model sizes considered in this work with 300B tokens already places them in the overtrained regime according to the Chinchilla Scaling Law [4]. Due to resource constraints and the limited time for rebuttal, further increasing the token count is neither feasible nor likely to produce significantly different trends in results. For example, training our 3.9B model with 300B tokens takes 30,000 GPU hours on a 256-V100 gpu cluster. Therefore, we believe the current findings still remain valid and representative. However, if the reviewer feels strongly about this, we can try to provide additional experiments on smaller models with more tokens by the camera ready deadline.
>
> > For the PTQ quantized model comparison versus QAT, can the author compare with QAT variant or use more recent PTQ methods such as SpinQuant/QuaRot?
>
> We appreciate the suggestion to compare our QAT quantized models with more recent PTQ variants such as SpinQuant or QuaRot. However, the primary focus of our work is to demonstrate the performance and scalability of extreme low-bitwidth models like Ternary LLMs. Our PTQ baselines are to establish a practical trade-off between performance and inference efficiency during deployment of FloatLMs. We chose GPTQ due to its widespread adoption in industry and its practical benefits. That said, we agree that exploring more advanced PTQ techniques could offer valuable insights, and we encourage future work to investigate such comparisons. If the reviewers feel strongly about this, we can try to provide experiments by the camera-ready deadline.
>
>
> > What is the training cost of full QAT training compared to the FP baseline? The paper mentions model parallelism at scale. Why is it even necessary, given that the largest model is just 3.9B parameters?
>
> The cost of Quantization-Aware Training (QAT) is comparable to that of the floating-point (FP) baseline. This is because we employ fake quantization, which allows us to simulate quantization during training without requiring significant computational overhead compared to FloatLM.
> Regarding model parallelism, while the largest model in our study has 3.9B parameters, training at scale still necessitates parallelization. This is due to the memory and computational constraints of training large models that requires calculating/storing the backpropagated gradients and forward activations for sequences of length 2k at every layer, especially on a cluster of V100 GPUs with 16GB of high-bandwidth memory (HBM2).
>
> [1] Biderman, S., et al. Pythia: A Suite for Analyzing Large Language Models Across Training and Scaling. ArXiv, abs/2304.01373.
> [2] Groeneveld, D., et al. OLMo: Accelerating the Science of Language Models. ArXiv, abs/2402.00838.
> [3] Hoffmann, J. et. (2022). Training Compute-Optimal Large Language Models. arXiv preprint arXiv:2203.15556.

---

> > ### Comment · Reviewer_AzmB · 2024-11-20
> >
> > Thanks for the response. I increase scores due to the release of detailed benchmarks for inference speed and memory usage.
> >
> > ***Ternary vs Quant 3/4-bit Variant:*** The paper's claim that the Ternary model outperforms floating-point or 3/4-bit quantized models under size constraints is unconvincing. For 3/4-bit models, although Quantization-Aware Training (QAT) with pretraining is costly, in practice, we can conduct QAT during fine-tuning, which significantly improves performance compared to Post-Training Quantization (PTQ)[1]. Even with PTQ alone, the GPTQ method is not the state-of-the-art for a fair comparison.
> >
> > ***Training data convergence:*** I would argue that training with 300 billion tokens did not overtrain the model, especially the 3.9 billion parameter version, considering that Llama3 was trained on 15 trillion tokens and showed continuous improvement. Therefore, the concern remains regarding the performance of floating-point versus quantized models after convergence.
> >
> > [1] LLM-QAT: Data-Free Quantization Aware Training for Large Language Models

---

> ### Author Response · Authors · 2024-11-21
> **Response to Reviewer AzmB: Ternary vs Quant 3/4-bit Variant**
>
> We thank the reviewer for their prompt reply.
>
> > Ternary vs Quant 3/4-bit Variant: The paper's claim that the Ternary model outperforms floating-point or 3/4-bit quantized models under size constraints is unconvincing. For 3/4-bit models, although Quantization-Aware Training (QAT) with pretraining is costly, in practice, we can conduct QAT during fine-tuning, which significantly improves performance compared to Post-Training Quantization (PTQ)[1]. Even with PTQ alone, the GPTQ method is not the state-of-the-art for a fair comparison.
>
> We believe the exact statement is not explicitly provided in the paper, but the implied findings and statements in the contributions section suggest that TriLMs consistently outperform their QuantLM and FloatLM counterparts of the same bit size (see line 116 in the contributions). QuantLM is defined as a post-training quantized model (see line 046). Additionally, we do not claim that TriLM scales better than Quantization-Aware Training (QAT) models of higher precision, such as 3- or 4-bit, with respect to size.
> We would like to emphasize that the motivation for comparing the Post-Training Quantization (PTQ) model with the pre-trained Ternary model (TriLM) is to assess whether quantization-aware training offers any advantage over its PTQ counterparts. We did not consider QAT or QAT during fine-tuning, as it would increase the overall FLOPs and training cost and, more importantly, undermine the original motivation. Furthermore, methods like LLM-QAT do not report any successful results for bit widths lower than 4 bits [1].
>
> Additionally, GPTQ is widely regarded as a reasonable baseline for PTQ weight-only quantization due to its widespread adoption and ability to achieve nearly lossless compression up to 4 bits (see benchmark in Table 9). Studies on the scaling laws of PTQ models provide evidence supporting optimal post-training quantization at 4 bits [2]. Even under the assumption of completely lossless compression up to 3 or 4 bits, our findings remain valid, as ternary models require only 1.58 bits and achieve comparable results to FloatLM at around 3.9 billion parameters. The only difference would be that, instead of 1 billion parameters, the TriLM curve would overtake 3 bit curves between 1.5 to 2.4 billion parameters (see the [attached figure](https://i.postimg.cc/BnRyW791/output-20.png)). Therefore, we believe our findings and baselines are well-supported. Moreover, we will include this point in our paper for further clarity.
>
> We acknowledge that incorporating more recent and advanced PTQ methods could help us determine the precise curve, but the trend will likely remain the same. However, due to time constraints and the scope of our work, and considering that it would not impact our findings, we plan to include them in future work.
>
> [1] LLM-QAT: Data-Free Quantization Aware Training for Large Language Models.
> [2] Dettmers, Tim, and Luke Zettlemoyer. "The case for 4-bit precision: k-bit Inference Scaling Laws." arXiv preprint arXiv:2212.09720 (2022).

---

> ### Author Response · Authors · 2024-11-21
> **Response to Reviewer AzmB: Training data convergence**
>
> > Training data convergence: I would argue that training with 300 billion tokens did not overtrain the model, especially the 3.9 billion parameter version, considering that Llama3 was trained on 15 trillion tokens and showed continuous improvement. Therefore, the concern remains regarding the performance of floating-point versus quantized models after convergence.
>
> To the best of our knowledge, no one has ever observed the complete convergence of a large language model with billions of parameters [1], [2], [3]. The reviewer themselves notes that LLaMA 3 demonstrates consistent improvement even after processing 15 trillion tokens. Training a 3.9B-parameter model with 15 trillion tokens is impractical given the computational resources available in academia. Instead, we provide experiments at a smaller scale to demonstrate that TriLM continues to improve with more data. This improvement is evident even after training on over 3030 times the amount of data (tokens) relative to the number of parameters, specifically in the smallest model (with limited capacity) of our suite. Our 99M model continues training with up to 300B tokens, as shown in Table 1 below. This indicates that TriLM continues to improve even at higher training token-to-parameter ratios. Therefore, our larger model (3.9B) should be able to train on up to 11.8 trillion tokens without reaching convergence, which is comparable to the magnitude of LLaMA 3’s training tokens. However, it is infeasible for us to train models at that scale to convergence.
>
> | Training Tokens | Perplexity of TriLM 99M | Perplexity of TriLM 560M |
> |-----------------|-------------------------|--------------------------|
> | 50B              | 27.11                            | 15.48                                 |
> | 150B            | 26.30                            | 14.50                                 |
> | 300B            | 25.70                            | 14.01                                 |
>
> **Table 1**: Perplexity of TriLM (99M and 560M) on the validation set across varying training token sizes. The results show that TriLMs improve as the training data size increases, highlighting the model’s scalability with larger datasets.
>
> [1] Touvron, H., Lavril, T., Izacard, G., Martinet, X., Lachaux, M. A., Lacroix, T., ... & Lample, G. (2023). LLaMA: Open and Efficient Foundation Language Models. arXiv preprint arXiv:2302.13971.
> [2] Touvron, H., Martin, L., Stone, K., Albert, P., Almahairi, A., Babaei, Y., ... & Scialom, T. (2023). Llama 2: Open Foundation and Fine-Tuned Chat Models. arXiv preprint arXiv:2307.09288.
> [3] Meta AI. (2024). The Llama 3 Herd of Models. arXiv preprint arXiv:2407.217836.

---

> ### Author Response · Authors · 2024-11-25
> **Response to Reviewer AzmB: Other PTQ Methods**
>
> > For the PTQ quantized model comparison versus QAT, can the author compare with QAT variant or use more recent PTQ methods such as SpinQuant/QuaRot?
>
> We are releasing additional benchmarks for OmniQuant, one of the most recently published quantization methods (ICLR 2024), in the [general comments section](https://openreview.net/forum?id=TJo6aQb7mK&noteId=tY3KHqgBZD), as requested by Reviewer Fojc. Regarding the reviewer’s suggestion on PTQ methods, we would like them to note that SpinQuant is currently under peer review at **ICLR 2025 (the same conference)** and QuaRot is still awaiting publication at **NeurIPS 2024**. As per **ICLR policy** on concurrent or contemporaneous work, if a paper was published (i.e., at a peer-reviewed venue) on or after July 1, 2024, authors are not required to compare their own work to that paper. We strongly believe that the findings of our paper remain valuable and valid irrespective of the PTQ technique.

---

> > ### Author Response · Authors · 2024-11-27
> > **Response to Reviewer AzmB**
> >
> > As we approach the conclusion of the rebuttal phase, we kindly request an acknowledgment of our response and paper revision. We would be happy to address any remaining major concerns the reviewer may have during the remaining time.
> >
> > We have made the following changes to address the reviewer’s concerns:
> > - Appendix H has been added, presenting inference benchmarks titled “Latency and Throughput Analysis of Ternary Quantization Formats”.
> > - An additional, most recently published QuantLM variant using OmniQuant has been added as Appendix D.5: Omniquant: 3-bit  Post-Training Qunatised Models.
> > - New sections have been added to Appendix A.10: “Perplexity for Increasing Training Tokens in TriLM Models.”
> >
> > Given the reviewer’s initial negative assessment, it would be valuable to know if they still have any major concerns after our rebuttal response. This would provide us with an opportunity to address them and potentially improve the reviewer’s score.

---

> ### Author Response · Authors · 2024-12-01
> **Response to Reviewer AzmB**
>
> We thank the reviewer for their time and feedback.
> Given the initial negative assessment of our work, and with the discussion period drawing to a close, we would be truly grateful if you could kindly review the revisions made in response to the initial comments. We would appreciate it if you could let us know whether you are satisfied with the revised manuscript and whether any further changes are necessary. Your feedback is highly valuable to us.

---

### Official Review · Reviewer_Z2nY · 2024-11-02

**Soundness:** 4
**Presentation:** 3
**Contribution:** 4
**Rating:** 10
**Confidence:** 3

**Summary:**

This paper presents an extensive study on the scaling law of low-bit models, specifically 4-bit, 3-bit, and ternary models. Ternary models (TriLMs) demonstrate impressive effectiveness when scaling model size, with the authors concluding that TriLMs achieve higher performance gains more rapidly during scaling compared to FloatLMs and QuantLMs. A 3.9B TriLM model exhibits performance similar to a 3.9B FloatLM but has a size smaller than that of an 830M FloatLM. The paper first introduces the theoretical advantages of scaling low-bit LMs using information theory. Following this, the authors propose the Spectra LLM suite, which facilitates training experiments on models of varying sizes and bit widths. Finally, scaling experiments conducted on the Spectra LLM platform reveal that scaling a TriLM is more efficient than scaling QuantLMs or FloatLMs. Detailed experimental results are provided, covering the training dynamics of low-bit LMs, pretraining outcomes, and evaluations on downstream tasks. The experiments show that, when scaled to 3.9B, the TriLM achieves comparable performance to the original FloatLM on most aspects, except in areas like toxicity and stereotyping. Further studies on BiLMs are also included in the appendix, where BiLMs show scaling effectiveness, although the performance gap reduction between BiLMs and FloatLMs is slower than with TriLMs.

**Strengths:**

This paper provides an exceptional perspective on the scaling law of low-bit models, with a particular focus on Ternary Models (TriLMs). The strengths of this study are summarized below:

1. Extensive and thorough workload. This paper comprehensively addresses major concerns regarding the scaling law of low-bit models, including training dynamics, loss curves during pretraining, and evaluation results on downstream tasks. The authors conducted numerous model training sessions and extensive evaluations to support their conclusions. Additionally, the paper provides well-articulated theoretical explanations for why low-bit models scale more effectively and examines how these models align with modern GPU architectures. The paper is inclusive and well-scaled in its coverage of relevant aspects.

2. High novelty and significant contribution. To the best of my knowledge, research on the scaling law of low-bit models is limited, making this work a valuable contribution with well-supported, insightful conclusions. By demonstrating the efficiency of scaling low-bit models, this paper encourages more effective model designs for scaling.

3. Valuable software contribution. As described in the paper, the Spectra LLM Suite serves as a platform for studying the scaling law of low-bit models. This platform is beneficial for the research community, facilitating future work on quantization and low-bit model training.

**Weaknesses:**

Although this paper provides valuable contributions, there are some areas for improvement. The overall score could be raised if the following issues are fixed and good discussion is made during the rebuttal phase.

1. Presentation needs refinement. While the figures in this paper offer essential information, some are poorly displayed. For instance, Figures 6 and 13 appear compressed, with text that is has low readability. Figure 5 is also confusing due to the similar colors used for arrows. A detailed explanation should guide readers through each part of Figure 5. Additionally, presenting Table 1 alongside Figure 5 could clarify the calculation details.

2. Content organization for conciseness. To fit within the constraints of a 10-page conference paper, the paper could better prioritize content. Since the main contribution is the scaling law of TriLMs versus FloatLMs, related content should be given more prominence. Sections 2.1 and 2.3 are to some extent redundant and could be moved to the Appendix. Section 4.3 could be split into separate sections on scaling law and training dynamics, with each elaborated more thoroughly. Moving parts of Appendix C and findings in Appendix A.5 to the main text would improve the discussion of the corresponding aspect. Moving some BiLM discussions to the main text would also help clarify the outline.

3. Deeper analysis of training dynamics. The paper notes a loss reduction halfway through training but could benefit from further examination of this phenomenon, as it may relate to efficient convergence and scaling. A detailed discussion of changes in the model at this stage would be valuable. Incorporating experiments similar to Appendix D "Analysis of the decay stage" of the MiniCPM paper [1] (COLM 2024 official version) might be helpful to explain this effect. Additionally, a deeper exploration of why low-bit models scale more efficiently than FloatLMs in terms of training dynamics would be insightful.

[1] MiniCPM: Unveiling the Potential of Small Language Models with Scalable Training Strategies (COLM 2024)

**Questions:**

1. Section 2.2 suggests that low-bit models should theoretically provide a better approach for capturing weight variance. In the experiments, TriLMs—where each parameter is restricted to {-1, 0, 1}—demonstrate better scalability than 4-bit models, which in turn scale better than floating-point models. However, in Section 5, the results show that 3-bit models exhibit lower scalability compared to 4-bit models and TriLMs. What might be the underlying reason for this? Additionally, why could TriLMs be more scalable than BiLMs?

2. Based on the discussion in Section 2.2, there may be a relationship between the number of states in each numeric representation and scaling efficiency. What might this relationship look like? Could there be an optimal number representation? This is intended as a discussion question, and a concrete answer is not required during the rebuttal phase. Personally, I believe this could be an interesting direction for future research.

---

> ### Author Response · Authors · 2024-11-20
> **Response to Reviewer Z2nY (1/2)**
>
> We appreciate the reviewer's positive assessment of our work and their thoughtful comments.
>
> > Presentation needs refinement. While the figures in this paper offer essential information, some are poorly displayed. For instance, Figures 6 and 13 appear compressed, with text that has low readability. Figure 5 is also confusing due to the similar colors used for arrows. A detailed explanation should guide readers through each part of Figure 5. Additionally, presenting Table 1 alongside Figure 5 could clarify the calculation details.
>
> We thank the reviewer for their feedback on the presentation and their comments on the figures and tables. Regarding Figures 6 and 13, we acknowledge the issue of compression and low text readability. We will ensure that the final version includes higher-resolution versions of these figures to improve clarity.
>
> For Figure 5, our initial idea was to use the same colors for TriLMs and FloatLMs throughout the paper. However, we understand the confusion caused by the similar colors of the arrows. We will revise the figure by using distinct and easily distinguishable colors for the arrows and provide a more detailed explanation in the caption to guide readers effectively through its components.
>
> We also appreciate the suggestion to present Table 1 alongside Figure 5 to clarify the calculation details. However, we would like to keep the computation flow figure in the main paper, as the information provided in the table is well-known. Additionally, we aim to keep the table of the forward pass beside the computation flow in the main paper.
>
> > Content organization for conciseness. To fit within the constraints of a 10-page conference paper, the paper could better prioritize content. Since the main contribution is the scaling law of TriLMs versus FloatLMs, related content should be given more prominence. Sections 2.1 and 2.3 are to some extent redundant and could be moved to the Appendix. Section 4.3 could be split into separate sections on scaling law and training dynamics, with each elaborated more thoroughly. Moving parts of Appendix C and findings in Appendix A.5 to the main text would improve the discussion of the corresponding aspect. Moving some BiLM discussions to the main text would also help clarify the outline.
>
> We appreciate the reviewer’s thoughtful suggestions regarding the content organization and conciseness of the paper. We agree that the primary contribution—the scaling law comparison between TriLMs and FloatLMs—should be more prominent. We will trim Sections 2.1 and 2.3 in the final version and move some findings from Appendices C, Figure-13 on BiLMs (Section B) and A.5 into the main text. Furthermore, we will revise Section 4.3 by splitting it into two distinct sections on scaling laws and training dynamics.
>
> > Deeper analysis of training dynamics. The paper notes a loss reduction halfway through training but could benefit from further examination of this phenomenon, as it may relate to efficient convergence and scaling. A detailed discussion of changes in the model at this stage would be valuable. Incorporating experiments similar to Appendix D "Analysis of the decay stage" of the MiniCPM paper [1] (COLM 2024 official version) might be helpful to explain this effect. Additionally, a deeper exploration of why low-bit models scale more efficiently than FloatLMs in terms of training dynamics would be insightful.
>
> We appreciate the reviewer’s suggestion to provide a deeper analysis of the training dynamics. We will aim to complete and include a section similar to Appendix D of the MiniCPM paper.

---

> ### Author Response · Authors · 2024-11-20
> **Response to Reviewer Z2nY (2/2)**
>
> Questions:
> >Section 2.2 suggests that low-bit models should theoretically provide a better approach for capturing weight variance. In the experiments, TriLMs—where each parameter is restricted to {-1, 0, 1}—demonstrate better scalability than 4-bit models, which in turn scale better than floating-point models. However, in Section 5, the results show that 3-bit models exhibit lower scalability compared to 4-bit models and TriLMs. What might be the underlying reason for this? Additionally, why could TriLMs be more scalable than BiLMs?
>
> We thank the reviewer for raising this interesting point. Our analysis in Section 2.2 suggests that low-bit models might be sufficient to capture the necessary weight variance as we scale the model size. This implies that lower-bit models could exhibit better scalability when trained with quantization-aware training (QAT).
> However, the results in Section 5 focus on post-training quantization (PTQ) benchmarks using 3 and 4-bit quantization, which doesn't fully align with our scaling theory. The motivation for Section 5 was to establish a practical trade-off between performance and inference efficiency during deployment. Ideally, to validate our scaling theory across different bit widths, we would perform QAT for various bit settings and include these results in Figure 7. Due to resource and time constraints, we leave this for future work.
> Regarding why TriLMs are more scalable than BiLMs, our paper shows that the weight distribution is centered around zero. This makes an odd number of quantized states (as in TriLMs with {-1, 0, 1}) more effective at capturing the weight variance than an even number (as in BiLMs with {-1, 1}), resulting in better scalability for TriLMs.
> >Based on the discussion in Section 2.2, there may be a relationship between the number of states in each numeric representation and scaling efficiency. What might this relationship look like? Could there be an optimal number representation? This is intended as a discussion question, and a concrete answer is not required during the rebuttal phase. Personally, I believe this could be an interesting direction for future research.
>
> Again this is a very interesting point and a great direction of future research. To establish such a relationship, we need to empirically establish a scaling law with different number of states/bits for QAT. We hypothesize that as we increase the number of bits/states the irreducible loss and the coefficient of the exponential term (A) should both decrease. However, a more in depth understanding and analysis requires a lot more experiments and is left as a future work.

---

> > ### Comment · Reviewer_Z2nY · 2024-11-20
> > **Response to the Authors**
> >
> > Thanks for providing such timely and detailed response. Your response fully answer my question. I would consider raising the overall rating if the additional experiments (W3) are added, content arangement (W2) are conducted and readability issues (W1) are resolved in the revised manuscript. Please let me know when the updated manuscript is uploaded.

---

> > > ### Author Response · Authors · 2024-11-26
> > > **Response to Reviewer Z2nY**
> > >
> > > We thank the reviewer for their timely and constructive feedback. We have made the following changes in our revised manuscript based on your suggestions.
> > >
> > > > W3: additional experiments
> > > -  Appendix F has been included, providing an analysis of the learning dynamics during the peak learning rate decrease and subsequent decay phase.
> > >
> > >
> > > >W2: content arangement
> > > - Section 2 has been streamlined for improved conciseness.
> > > - The section on training dynamics and scaling laws has been separated, with the training dynamics section expanded for further detail.
> > >
> > > >W1: readability issues
> > > - Figure 4 has been added, along with the forward pass equation for all models, to enhance clarity.
> > > -  Figures 2, 3, 5, 6, and 12 have been updated to correct scaling issues and ensure consistent color usage.
> > >
> > > Please let us know if you have any further suggestions.

---

> ### Comment · Reviewer_Z2nY · 2024-11-26
> **Response to the Authors**
>
> Thanks for your reply and the modifications in the manuscript. The newly added changes have fully addressed my issues and concerns. I would raise my overall rating while maintaining the confidence score.
>
> Again, thanks for the time and effort paid during the discussion period. I appreciate such contribution towards the community of scaling law and efficient LLMs.

---

### Official Review · Reviewer_fojc · 2024-11-02

**Soundness:** 4
**Presentation:** 4
**Contribution:** 4
**Rating:** 10
**Confidence:** 4

**Summary:**

In this paper, the authors present the Spectra LLM suite, i.e a suite of ternary (1.5-bit) language models (TriLLMs), ranging from 99M to 3.9B parameters.

Specifically, the authors first extensively pretrain TriLLMs in various size and the corresponding LLM in FP16 (FloatLLM) with exactly the same recipe, and study their differences. The findings are quite fruitful:
1. **Convergence behavior**: TriLLM in various scales can converge normally. With some techniques, like decreasing peak learning rate and removing weight decay at some points, they can converge faster.
2. **Scaling law**: Overall, TriLLM demonstrate a similar scaling law as FloatLLM. With a similar model size in GB, TriLLM achives lower perplexity than FloatLLM. With a similar parameter count, TriLLM is worse than FloatLLM.
3. **Benchmark accuracy**: The authors compare TriLLM to FloatLLM and the quantized version of FloatLLM with GPTQ (QuantLLM), the experimental results show: (1) With a similar model size, TriLLM outperforms FloatLLM and QuantLLM <= 4-bit; (2) With a similar parameter count, TriLLM is worse than FloatLLM and QuantLLM-4bit, but outperforms QuantLLM-3bit.

**Strengths:**

1. The paper is well-written, and the structure of the paper is clear.
2. The experiments are extensive and the settings are fair, with all claims being well supported.
3. The findings are interesting. And I believe that the release of all models (including TriLLM and FloatLLM) will further advance the study of TriLLM.

**Weaknesses:**

1. The only weakness is the applied quantization method in this paper, i.e. GPTQ. GPTQ performs well for bit-level >= 4-bit. However, with lower bit-level, its performance degrades significantly. It would be interesting to see the comparison with more advanced post-training quantization methods, like OmniQuant [1], BiLLM [2], AffineQaunt [3] and so on.


[1] OmniQuant: Omnidirectionally Calibrated Quantization for Large Language Models
[2] AffineQuant: Affine Transformation Quantization for Large Language Models
[3] BiLLM: Pushing the Limit of Post-Training Quantization for LLMs

**Questions:**

1. In Table 5, why isn't the number of skipped tokens proportional to the number of skipped batch?
2. In Figure 6b, what's the motivation for the comparison between TriLLM 2.4B and FloatLLM 1.1B and 1.5B, instead of FloatLLM 2.4B? I think it's better to compare with a similar number of paramters or with a similar model size.
3. Could you also offer some few-shot results on some benchmarks, like MMLU and Lambda? It would be interesting to see the in-context learning ability of TriLLM.

---

> ### Author Response · Authors · 2024-11-20
> **Response to Reviewer fojc**
>
> We are glad that the reviewer found the findings interesting and well-supported by the experiments.
>
> ## Weaknesses:
> > The only weakness is the applied quantization method in this paper, i.e. GPTQ. GPTQ performs well for bit-level >= 4-bit. However, with lower bit-level, its performance degrades significantly. It would be interesting to see the comparison with more advanced post-training quantization methods, like OmniQuant [1], BiLLM [2], AffineQaunt [3] and so on.
>
> We agree that investigating alternative post-training quantization techniques, such as OmniQuant, BiLLM, and AffineQuant, represents a promising avenue for future research. However, within the scope of this study, we prioritized GPTQ due to its widespread adoption and its relevance to the current state of the art in quantization methodologies for large language models. We will explicitly address this limitation in the manuscript and propose the exploration of more advanced methods as a direction for future work. If the reviewers feel strongly about this, we are willing to conduct additional experiments by the camera-ready deadline.
>
> ## Question:
>
> > In Table 5, why isn't the number of skipped tokens proportional to the number of skipped batch?
>
> We appreciate the reviewer’s comments. In response, we clarify that the number of skipped tokens is calculated as the product of the number of skipped batches and the batch size. Additionally, we follow the typical batch size used for FloatLM, which is approximately double that of TriLM, as reported in prior works [1], [2]. Specifically, we use a batch size of 1.08 million for TriLM and 2.06 million for FloatLM, consistent with these benchmarks.
>
> > In Figure 6b, what's the motivation for the comparison between TriLLM 2.4B and FloatLLM 1.1B and 1.5B, instead of FloatLLM 2.4B? I think it's better to compare with a similar number of paramters or with a similar model size.
>
> In response to the reviewer’s concern, our primary motivation for this comparison is to analyze the performance of TriLLM in relation to FloatLLM models that exhibit similar loss behaviors. The FloatLLM 1.1B and 1.5B models were selected because they showed comparable performance in terms of loss to TriLLM 2.4B. While we understand the preference for comparing models of similar parameter size, our intent was to focus on comparing models with closer loss characteristics rather than parameter count alone. However, we acknowledge the reviewer’s suggestion and will include a comparison with FloatLLM 2.4B in the revised manuscript for a more comprehensive analysis.
>
>
> > Could you also offer some few-shot results on some benchmarks, like MMLU and Lambda? It would be interesting to see the in-context learning ability of TriLLM.
>
> We appreciate the reviewer’s suggestion to include few-shot results on MMLU and Lambda benchmarks. We are currently working on expanding the evaluation of TriLLM’s in-context learning ability and will include results on these benchmarks in the updated version of the paper. This will provide further insights into the model’s performance under few-shot conditions.
>
> [1] Schoelkopf, H., et al. (2023). "Pythia: A Suite for Analyzing Large Language Models Across Training and Scaling"
> [2] Wang, H., Ma, S., Dong, L., Huang, S., Wang, H., Ma, L., Yang, F., Wang, R., Wu, Y., & Wei, F. (2023). BitNet: Scaling 1-bit Transformers for Large Language Models. arXiv:2310.114531.

---

> > ### Comment · Reviewer_fojc · 2024-11-22
> >
> > Thank you for the response. Since my question and weakness are mainly about clarification, I maintain my initial score.

---

> ### Author Response · Authors · 2024-11-25
> **Response to Reviewer fojc**
>
> We thank the reviewer for their response and original suggestion on trying more recent PTQ techniques. In response, we have included additional benchmarks for post-training quantized (QuantLM 3-bit) models, which were quantized using the OmniQuant, [in the general comments](https://openreview.net/forum?id=TJo6aQb7mK&noteId=tY3KHqgBZD) to the reviewer.

---

> > ### Comment · Reviewer_fojc · 2024-11-29
> >
> > Thank you for the results of new PTQ method. My only weakness is resolved. I'm strong inclined to the acceptance of this paper.

---

> > > ### Author Response · Authors · 2024-11-29
> > > **Response to Reviewer fojc**
> > >
> > > Thank you for your very positive feedback and valuable time. We sincerely appreciate your thoughtful evaluation and are grateful for your strong inclination to accept our work. If it is not too much trouble, we would also like to kindly inquire if any updates will be made to the score.

---

> > > > ### Comment · Reviewer_fojc · 2024-12-01
> > > >
> > > > Have raised my score. Overall, it is a paper with fruitful findings, and the experimental settings are strict ablative.

---

> ### Author Response · Authors · 2024-11-29
> **Response to Reviewer fojc**
>
> Dear Reviewer fojc,
>
> We sincerely thank you for your time and positive assessment of our work.
>
> As the discussion period draws to a close, we kindly request that you review our [updated manuscript](https://openreview.net/forum?id=TJo6aQb7mK&noteId=qNrtCMY68w) to determine if all initial concerns have been adequately addressed. During the review period, we have added new experiments and sections suggested by the other reviewers, as well as improved the presentation, writing, and overall structure of the paper. If you find the revisions satisfactory, we would greatly appreciate it if you could consider increasing your score accordingly. Your feedback is valuable to us.
>
> Best regards,
> The Authors

---

### Official Review · Reviewer_gRaf · 2024-11-02

**Soundness:** 4
**Presentation:** 4
**Contribution:** 4
**Rating:** 8
**Confidence:** 3

**Summary:**

This paper addressed the limitation of post-training quantization by investigating the pretraining of low-bitwidth models specifically Ternary Language Models as an alternative to traditional floating-point models and their post-training quantized versions (QuantLMs). The authors conducted extensive experiments to evaluate the model's performance in different aspects. The experiment results and analysis revealed valuable insights into the feasibility and scalability of low-bandwidth language models.

**Strengths:**

1. Authors comprehensively evaluate the model's commonsense reasoning performance, knowledge capacity, and toxicity. The experiment part is sufficient and convincing.
2. This work is well presented and it offers valuable insights into the pertaining of low-bitwidth models.

**Weaknesses:**

1. The pretraining cost of QuantLMs is not revealed (e.g., hardware, GPU hours).
2. The paper does not discuss how the hyperparameters are selected and tuned.

**Questions:**

1. Could the authors clarify what the "validation loss" in Figure.7 means? Is it "Log Perplexity" just as in the y-axis in Figure.19 in Appendix D.4? I am curious about the model's generation performance on commonly used datasets such as WikiText-2, C4, PTB.
2. Some fonts in the line chart are difficult to recognize, such as legends in Figure 6(a). In addition, many figures seem incorrectly scaled, making the label of the x/y-axis twisted. Authors should check all figures to improve the clarity.
3. This paper only provides the maximal speed-up compared with FP16. Authors are recommended to benchmark the end-to-end inference performance, such as throughput, first token time, and average latency.

---

> ### Author Response · Authors · 2024-11-19
> **Response to Reviewer gRaf**
>
> We greatly appreciate the reviewer’s feedback and suggestions.
>
> ## Weaknesses:
> > Pretraining Cost of QuantLMs
>
> We acknowledge the importance of providing a clearer breakdown of the pretraining costs for QuantLMs. While we did not implicitly state this information in the initial submission, we are happy to add more details regarding the hardware specifications and GPU hours required for the pretraining of QuantLMs in the revised manuscript. The pretraining cost of QuantLMs is almost the same as that of FloatLMs as the post training quantization cost is negligible. We only use 512 samples with a sequence length of 2048 on single A-100 NVIDIA GPU.
>
> > Hyperparameters details on selection and tuning.
>
>
> We appreciate the reviewer pointing out the need for more clarity on the selection and tuning of hyperparameters. In the revised manuscript, we will include a more comprehensive discussion on how the hyperparameters were selected, and provide additional details about the tuning process used for the experiments.
> **Model architecture hyperparameters.** We focused on minimizing the risks of loss spikes and slow convergence, which guided our choices of key hyperparameters, such as layer norms and training parameters (see below). To further improve hardware efficiency, we padded the vocabulary size and rounded the hidden dimension to the nearest power of two. Additionally, due to compute constraints, we limited our scaling curve and coefficient analysis to 9 data points within the 4-billion-parameter range, each trained on a dataset of 300 billion tokens for the suite. These considerations influenced our selection of the number of layers, hidden dimensions, embedding size, feedforward network size, and attention heads for each variant.
>
> **Training hyperparameters.** Throughout model training, we perform downstream evaluations to inform decisions regarding model initialization, optimizers, and learning rate schedules. This methodology is similar to the training-time evaluations used for hyperparameter selection in other model suites (e.g., Pythia and Olmo). Additionally, we employ a grid search approach to systematically explore a wider range of hyperparameter configurations, including learning rate, weight decay, batch size, and optimizer. Evaluations are conducted at different stages of training, from 4 billion up to 20 billion tokens, providing continuous and early-stage feedback on model performance. The key metrics evaluated include validation loss, commonsense reasoning, knowledge-based task performance, and toxicity assessments, as outlined in the paper.
> Initialization hyperparameters. Spectra suite models utilize standard initialization techniques for transformer architectures, including Kaiming initialization for weights.
>
>
> ## Questions:
> > Q1: Could the authors clarify what the "validation loss" in Figure.7 means? Is it "Log Perplexity" just as in the y-axis in Figure.19 in Appendix D.4? I am curious about the model's generation performance on commonly used datasets such as WikiText-2, C4, PTB.
>
> To clarify, the validation loss in Figure 7 refers to the final log perplexity evaluated on an unseen validation subset of Slim Pajama, as mentioned in the paper. While it is conceptually the same as the “log perplexity” shown in Figure 19, the datasets used for evaluation differ. The results in Figure 19 are based on separate datasets, which may or may not overlap with Slim Pajama.
>
>
>
>
>
> > Q2: Some fonts in the line chart are difficult to recognize, such as legends in Figure 6(a). In addition, many figures seem incorrectly scaled, making the label of the x/y-axis twisted. Authors should check all figures to improve the clarity.
>
> Thank you for pointing out this issues. We acknowledge that some figures, particularly Figure 6(a), have smaller font sizes that can impact legibility. Additionally, we understand that figure scaling issues have caused the x/y-axis labels to appear distorted.
>
>  In response to your feedback, we will make the following updates in the upcoming revised version of the paper:
> Figure 6 (a) and (b): We will increase the font size of the legends and axis labels and enhance the color scheme to improve differentiation.
> Figure 3 (a) and (b): We will adjust the font size of the figures and improve the color schemes.
> We will also fix the scaling issue with the other figures in the paper.
>
>
> > Q3.  This paper only provides the maximal speed-up compared with FP16. Authors are recommended to benchmark the end-to-end inference performance, such as throughput, first token time, and average latency.
>
> Thank you for highlighting the missing throughput and latency. We have included the results in the [general comments for all reviewers](https://openreview.net/forum?id=TJo6aQb7mK&noteId=9Lym0unZ9V) and will update them in the appendix. Additionally, we will address the GPU kernels in the “Future Work” section, as further optimization of the GPU and CPU kernel implementation remains an ongoing area of development.

---

> ### Author Response · Authors · 2024-11-29
> **Response to Reviewer gRaf**
>
> Dear Reviewer gRaf,
>
> We sincerely thank you for your time and positive assessment of our work.
>
> As the discussion period nears its conclusion, we kindly request that you review our [updated manuscript](https://openreview.net/forum?id=TJo6aQb7mK&noteId=qNrtCMY68w) to determine if all initial concerns have been adequately addressed.
> We have made the following changes to address the reviewers’ concerns:
>
> - Figure 4 has been added, along with the forward pass equation for all models, to enhance clarity.
> - Additional sections have been added to Appendix A: ‘Details on the Selection of Hyperparameters.
> - Figures 2, 3, 5, 6, and 12 have been updated to correct scaling issues and ensure consistent color usage.
> - Appendix H has been added, presenting inference benchmarks titled “Latency and Throughput Analysis of Ternary Quantization Formats”.
>
> During the review period, we have added new experiments and sections suggested by the other reviewers, as well as improved the presentation, writing, and overall structure of the paper. If you find the revisions satisfactory, we would greatly appreciate it if you could consider increasing your score accordingly. Your feedback is invaluable to us.
>
> Best regards,
> The Authors

---

> ### Author Response · Authors · 2024-12-01
> **Response to Reviewer gRaf**
>
> We thank the reviewer for their very positive assessment of our work and constructive feedback. As the discussion period is drawing to a close, we would be truly grateful if you could kindly review the revisions made in response to the initial feedback and let us know if you are satisfied with the revised manuscript and whether any additional changes are required. Your feedback is highly valuable to us.

---

### Author Response · Authors · 2024-11-19
**General comments on inference efficiency for all the reviewers**

We gratefully acknowledge the constructive feedback provided by reviewers gRaf02 and AzmB. In response, we present additional details regarding the inference speedup achieved by our ternary models through the use of llama.cpp kernels [1], an open-source project specifically optimized for efficient inference.

## Packing Schemes for Ternary Weights
Below, we provide details on the packing of our weights.

### TQ1 Packing: 1.6875 Bits per Weight

The packing structure used in TQ1 is as follows:
- Packing: The first 240 elements are packed with 5 elements per byte, and the remaining 16 elements with 4 elements per byte.
- Block structure: Each block of 256 ternary values is packed into 52 bytes, consisting of 48 bytes for 240 elements and 4 bytes for the remaining 16 elements, along with a float16 scale per block. This results in a 1.6875 bpw representation.
- Ternary value mapping: The ternary values are stored as unsigned integers, where the values {-1, 0, 1} are mapped to {0, 1, 2}, respectively.

### TQ2 Packing : 2.0625 Bits per Weight


The TQ2 format was developed as a faster alternative to TQ1, particularly for compute-bound hardware.
- Packing: Each byte contains 4 ternary values, with 2 bits used to represent each value.
- Block structure: Each block consists of 256 ternary values, which are packed into 64 bytes, with a float16 scale per block. This results in a 2.0625 bpw representation.
- Ternary value mapping: Same as TQ1.

It is crucial to highlight that the choice of 256 elements is deliberate, as all of our models are designed with row sizes that are multiples of 256, ensuring compatibility and optimal performance.


## Experimental Results

The performance of TQ1 and TQ2 was evaluated in terms of throughput and latency. **The results indicate that TQ2 is significantly faster than other quantization formats.**
A comparison of the throughput of various quantization formats is presented in the table below:

| Configuration         | Model Size      | Apple M3 Pro (GB/s)       | Intel Core m3-8100Y (AVX2) (GB/s) |
|-----------------------|-----------|---------------------------|------------------------------------|
| **f16**               | 7.99 GB  | 93.47                     | 30.60                              |
| **q8**              | 4.24 GB  | 131.86                    | 67.03                              |
| **q4**              | 2.42 GB  | 135.33                    | 64.17                              |
| **TQ1**             | 994 MB   | 124.70                    | 70.31                              |
| **TQ2**             | 1.17 GB  | **176.43**                   | **141.83**                             |

A latency comparison on the MacBook M3 Pro across various configurations is shown in the table below:
| **Configuration**  | **Total Time (ms)** | **Tokens Processed** | **Time per Token (ms)** | **Tokens per Second** |
|--------------------|---------------------|----------------------|-------------------------|-----------------------|
| **F16**            | 12574.22           | 260                  | 48.44                   | 20.64                 |
| **q8**           | 4660.77            | 260                  | 17.69                   | 56.53                 |
| **q4**           | 4927.69            | 261                  | 18.88                   | 52.97                 |
| **TQ1**           | 7022.70            | 260                  | 27.01                   | 37.02                 |
| **TQ2**            | **3990.42**           | **260**                  | **15.27**                   | **65.47**                 |


Throughput is measured for the quantized vector operations in llama.cpp, providing a comparison across different quantization types. q8 and q4 refer to the 8-bit and 4-bit quantized formats, respectively. Additionally, it should be noted that the TQ1 kernels are not fully optimized for Metal.

The primary focus of our paper is to explore the scalability and feasibility of pre-training extreme-low bit models, such as binary and ternary LLMs, and to compare their performance with FloatLMs. To demonstrate actual speedup numbers, we will include these results in the appendix. While the CPU benchmarking results are presented here, we acknowledge that there is room for further optimization of both CPU and GPU kernel implementations. This optimization remains an important area for future work.

[1] Gerganov, G. et al (2023). llama.cpp: Port of Facebook’s LLaMA model in C++. Retrieved from https://github.com/ggerganov/llama.cpp.

---

### Author Response · Authors · 2024-11-25
**General Comments on Benchmarking 3-bit Post-Training Quantized Models with OmniQuant**

We provide additional 3-bit quantized models, which were trained using Omniquant for 5 iterations with a group size of 1 row.

# Model Performance Table

| Tasks                | Metric      | 99M     | 190M    | 390M    | 560M    | 1.1B   | 1.5B   | 2.4B   |
|----------------------|-------------|---------|---------|---------|---------|--------|--------|--------|
| arc_challenge        | acc         | 0.19 ± 0.01 | 0.21 ± 0.01 | 0.22 ± 0.01 | 0.22 ± 0.01 | 0.24 ± 0.01 | 0.26 ± 0.01 | 0.27 ± 0.01 |
|                      | acc_norm    | 0.23 ± 0.01 | 0.24 ± 0.01 | 0.26 ± 0.01 | 0.24 ± 0.01 | 0.26 ± 0.01 | 0.29 ± 0.01 | 0.30 ± 0.01 |
| arc_easy             | acc         | 0.37 ± 0.01 | 0.41 ± 0.01 | 0.45 ± 0.01 | 0.48 ± 0.01 | 0.55 ± 0.01 | 0.56 ± 0.01 | 0.60 ± 0.01 |
|                      | acc_norm    | 0.34 ± 0.01 | 0.38 ± 0.01 | 0.41 ± 0.01 | 0.42 ± 0.01 | 0.48 ± 0.01 | 0.51 ± 0.01 | 0.53 ± 0.01 |
| boolq                | acc         | 0.55 ± 0.01 | 0.54 ± 0.01 | 0.62 ± 0.01 | 0.57 ± 0.01 | 0.63 ± 0.01 | 0.59 ± 0.01 | 0.63 ± 0.01 |
| hellaswag            | acc         | 0.28 ± 0.00 | 0.29 ± 0.00 | 0.32 ± 0.00 | 0.34 ± 0.00 | 0.38 ± 0.00 | 0.40 ± 0.00 | 0.44 ± 0.00 |
|                      | acc_norm    | 0.29 ± 0.00 | 0.32 ± 0.00 | 0.39 ± 0.00 | 0.41 ± 0.00 | 0.48 ± 0.01 | 0.51 ± 0.01 | 0.57 ± 0.00 |
| lambada_openai       | acc         | 0.05 ± 0.00 | 0.07 ± 0.00 | 0.23 ± 0.01 | 0.25 ± 0.01 | 0.31 ± 0.01 | 0.35 ± 0.01 | 0.44 ± 0.01 |
| logiqa               | acc         | 0.16 ± 0.01 | 0.20 ± 0.02 | 0.23 ± 0.02 | 0.23 ± 0.02 | 0.21 ± 0.02 | 0.23 ± 0.02 | 0.19 ± 0.02 |
|                      | acc_norm    | 0.23 ± 0.02 | 0.25 ± 0.02 | 0.29 ± 0.02 | 0.28 ± 0.02 | 0.27 ± 0.02 | 0.26 ± 0.02 | 0.30 ± 0.02 |
| piqa                 | acc         | 0.58 ± 0.01 | 0.59 ± 0.01 | 0.63 ± 0.01 | 0.65 ± 0.01 | 0.69 ± 0.01 | 0.69 ± 0.01 | 0.71 ± 0.01 |
|                      | acc_norm    | 0.58 ± 0.01 | 0.59 ± 0.01 | 0.63 ± 0.01 | 0.66 ± 0.01 | 0.69 ± 0.01 | 0.70 ± 0.01 | 0.72 ± 0.01 |
| sciq                 | acc         | 0.55 ± 0.02 | 0.64 ± 0.02 | 0.74 ± 0.01 | 0.77 ± 0.01 | 0.82 ± 0.01 | 0.82 ± 0.01 | 0.86 ± 0.01 |
|                      | acc_norm    | 0.49 ± 0.02 | 0.59 ± 0.02 | 0.61 ± 0.02 | 0.68 ± 0.01 | 0.73 ± 0.01 | 0.76 ± 0.01 | 0.80 ± 0.01 |
| triviaqa             | exact_match | 0.00 ± 0.00 | 0.00 ± 0.00 | 0.01 ± 0.00 | 0.01 ± 0.00 | 0.05 ± 0.00 | 0.07 ± 0.00 | 0.07 ± 0.00 |
| winogrande           | acc         | 0.51 ± 0.01 | 0.50 ± 0.01 | 0.51 ± 0.01 | 0.53 ± 0.01 | 0.56 ± 0.01 | 0.56 ± 0.01 | 0.58 ± 0.01 |
| mmlu (continuation)  | acc         | 0.25 ± 0.00 | 0.25 ± 0.00 | 0.26 ± 0.00 | 0.27 ± 0.00 | 0.29 ± 0.00 | 0.28 ± 0.00 | 0.30 ± 0.00 |
| - humanities         | acc         | 0.24 ± 0.01 | 0.23 ± 0.01 | 0.24 ± 0.01 | 0.25 ± 0.01 | 0.26 ± 0.01 | 0.26 ± 0.01 | 0.27 ± 0.01 |
| - other              | acc         | 0.27 ± 0.01 | 0.27 ± 0.01 | 0.29 ± 0.01 | 0.29 ± 0.01 | 0.33 ± 0.01 | 0.32 ± 0.01 | 0.36 ± 0.01 |
| - social sciences    | acc         | 0.26 ± 0.01 | 0.27 ± 0.01 | 0.29 ± 0.01 | 0.29 ± 0.01 | 0.32 ± 0.01 | 0.30 ± 0.01 | 0.33 ± 0.01 |
| - stem               | acc         | 0.22 ± 0.01 | 0.23 ± 0.01 | 0.25 ± 0.01 | 0.24 ± 0.01 | 0.25 ± 0.01 | 0.25 ± 0.01 | 0.27 ± 0.01 |

It is important to note that the benchmarks are comparable to GPT-Q (values provided as percentages in Tables 6, 7, 9, 14, and 15 of the main paper) and, therefore, do not affect our findings. Furthermore, we followed best practices for GPT-Q [1]

Reference:
[1] Malinovskii, V., Mazur, D., Ilin, I., Kuznedelev, D., Burlachenko, K., Yi, K., Alistarh, D., & Richtárik, P. (2024). PV-Tuning: Beyond Straight-Through Estimation for Extreme LLM Compression. arXiv preprint arXiv:2405.1485216.

---

### Author Response · Authors · 2024-11-26
**Changes to the manuscript**

We have made the following revisions to the manuscript based on the reviewer’s feedback:

- Figure 4 has been added, along with the forward pass equation for all models, to enhance clarity.
- Section 2 has been streamlined for improved conciseness.
- The section on training dynamics and scaling laws has been separated, with the training dynamics section expanded for further detail.
- Two new sections have been added to Appendix A: “Details on the Selection of Hyperparameters” and “Perplexity for Increasing Training Tokens in TriLM Models”.
- Figures 2, 3, 5, 6, and 12 have been updated to correct scaling issues and ensure consistent color usage.
- Appendix F has been added, presenting an analysis of the "learning dynamics during the peak learning rate drop and decay stage".
- Appendix H has been added, presenting inference benchmarks titled “Latency and Throughput Analysis of Ternary Quantization Formats”.
- An additional QuantLM variant using OmniQuant has been added as Appendix D.5.
- Added a table of contents for the appendix section.

---

### Meta-Review · Area_Chair_TXFZ · 2024-12-21

**Metareview:**

TriLMs offers a more efficient alternative to traditional floating-point models and post-training quantized models in terms of scaling behavior and performance at larger model sizes. The paper presents a thorough evaluation of TriLMs across various aspects, including model size scaling, training dynamics, commonsense reasoning, knowledge capacity, and toxicity. The paper is well-organized. The authors provide benchmarks for various quantization formats, including inference speed and memory usage. The paper compares their TriLM models with existing methods like BitNet b1.58. The authors have actively addressed reviewer concerns during the rebuttal, improving the paper's clarity, providing further clarifications.

Reviewer DqjN felt the work was too similar to BitNet b1.58 in terms of architecture, training approach, and some conclusions. They questioned what additional contributions the paper offered beyond this prior work. The claim that Spectra was the "first" to demonstrate the feasibility of pretraining ternary language models was challenged, given the existence of BitNet b1.58. Reviewer AzmB was concerned that the models were not fully converged after training on 300B tokens, especially the larger models. They questioned how the performance of FloatLMs and TriLMs would compare after full convergence. Reviewer AzmB argued that comparing a pretrained TriLM with PTQ models was unfair, as QAT during fine-tuning could significantly improve the performance of 3/4-bit models.

Two reviewers with high scores gave low confidence scores (3), while two reviewers who leans to reject the paper had high confidence scores (5 and 4). Given the mixed reviews and the detailed responses, the paper can be accepted for a poster presentation.

**Additional Comments On Reviewer Discussion:**

Comparisons with more recent PTQ techniques like OmniQuant, BiLLM, or AffineQuant were initially missing, but the authors added OmniQuant results later in the rebuttal.

---

### Decision · Program_Chairs · 2025-01-22

Accept (Spotlight)